# EquiReg: Equivariance Regularized Diffusion for Inverse Problems

## Abstract

Diffusion models represent the state-of-the-art for solving inverse problems such as image restoration tasks. Diffusion-based inverse solvers incorporate a likelihood term to guide prior sampling, generating data consistent with the posterior distribution. However, due to the intractability of the likelihood, most methods rely on isotropic Gaussian approximations, which can push estimates off the data manifold and produce inconsistent, poor reconstructions. We propose *Equivariance Regularized* (EquiReg) diffusion, a general plug-in framework that improves posterior sampling by penalizing trajectories that deviate from the data manifold. EquiReg formalizes manifold-preferential equivariant functions that exhibit low equivariance error for on-manifold samples and high error for off-manifold ones, thereby guiding sampling toward symmetry-preserving regions of the solution space. We highlight that such functions naturally emerge when training non-equivariant models with augmentation or on data with symmetries. EquiReg's largest gains are under reduced sampling and measurement consistency steps, where many methods suffer severe quality degradation. By regularizing trajectories toward the manifold, EquiReg implicitly accelerates convergence and enables high-quality reconstructions. EquiReg consistently improves performance in linear and nonlinear image restoration tasks and solving partial differential equations.

## 1 Introduction

Inverse problems aim to recover an unknown signal $\boldsymbol{x}^* \in \mathbb{R}^d$ from undersampled noisy measurements:

$$\boldsymbol{y} = \mathcal{A}(\boldsymbol{x}^*) + \boldsymbol{\nu} \in \mathbb{R}^m, \tag{1}$$

where $\mathcal{A}$ is a known measurement operator, and $\boldsymbol{\nu}$ is an unknown noise (Groetsch, 1993). Inverse problems are widely studied in science and engineering, including imaging and astrophotography.

Inverse problems are ill-posed, i.e., the inversion process can have many solutions; hence, they require prior information about the desired solution (Kabanikhin, 2008). In the Bayesian formulation, inference targets the posterior distribution $p(\boldsymbol{x}|\boldsymbol{y}) \propto p(\boldsymbol{y}|\boldsymbol{x})p(\boldsymbol{x})$, where $p(\boldsymbol{y}|\boldsymbol{x})$ is the likelihood of the measurements and $p(\boldsymbol{x})$ is a prior describing the signal structure (Stuart, 2010). Examples of handcrafted priors include sparsity (Donoho, 2006) and low-rankness (Candès et al., 2011).

This paper focuses on methods that leverage unconditionally pre-trained score-based generative diffusion models as learned priors (Ho et al., 2020; Song & Ermon, 2019) with applications in image restoration (Chung et al., 2023), medical imaging (Chung et al., 2022a), and solving partial differential equations (PDEs) (Huang et al., 2024; Yao et al., 2025). These methods define a sequential noising process $\boldsymbol{x}_0 \sim p_{\text{data}} \to \boldsymbol{x}_t \to \boldsymbol{x}_T \sim p_T(\boldsymbol{x}) \approx \mathcal{N}(\boldsymbol{0}, \boldsymbol{I})$ and a reverse denoising process parameterized by a neural network score $\nabla_{\boldsymbol{x}_t} \log p_t(\boldsymbol{x}_t)$ (Vincent, 2011). During sampling, these approaches incorporate gradient signals carrying likelihood information to solve inverse problems.

Solving inverse problems with diffusion (Zhang et al., 2025a; Alkhouri et al., 2025) requires computing the conditional score $\nabla_{\boldsymbol{x}_t} \log p_t(\boldsymbol{x}_t|\boldsymbol{y})$, decomposed into $\nabla_{\boldsymbol{x}_t} \log p_t(\boldsymbol{x}_t) + \nabla_{\boldsymbol{x}_t} \log p_t(\boldsymbol{y}|\boldsymbol{x}_t)$. This introduces challenges, as the likelihood score $\nabla_{\boldsymbol{x}_t} \log p_t(\boldsymbol{y}|\boldsymbol{x}_t) = \nabla_{\boldsymbol{x}_t} \log \int p(\boldsymbol{y}|\boldsymbol{x}_0)p_t(\boldsymbol{x}_0|\boldsymbol{x}_t)\mathrm{d}\boldsymbol{x}_0$ is only computationally

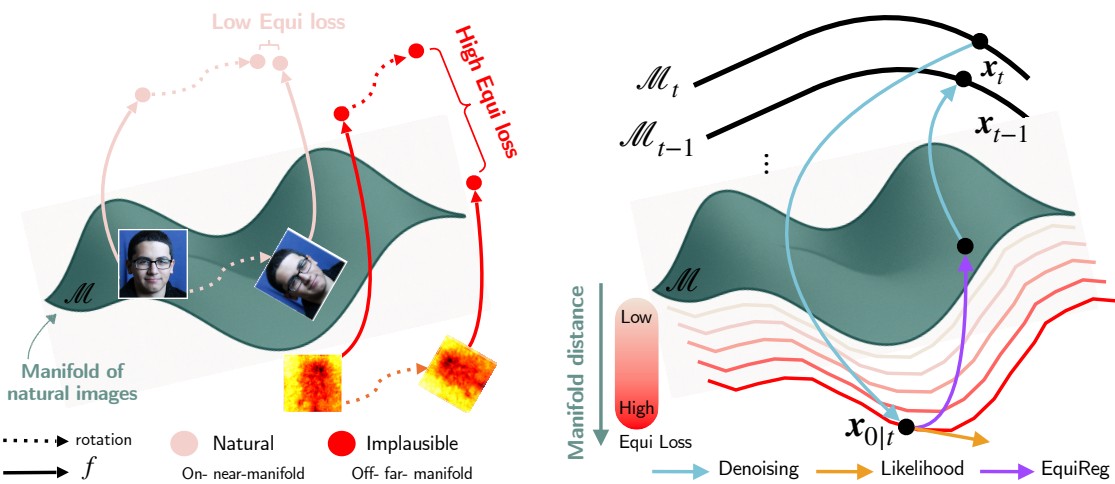

Figure 1: **Equivariance Regularized (EquiReg) diffusion for inverse problems.** (left) Manifold preferential equivariance (MPE) functions whose equivariance error is lower for on-manifold and higher for off-manifold data. (right) EquiReg regularizes the posterior sampling trajectory for improved performance. It penalizes off-manifold trajectories via MPE-based regularization. This is a schematic illustrating the motivation of our method; the depicted EquiLoss color gradient is illustrative rather than a direct measurement of manifold distance, with empirical equivariance error shown in Figure 4a. $\mathcal{M}_t$ refers to the noisy data manifold at diffusion time $t$ (i.e., the support of $p_t$), with $\mathcal{M}_0 = \mathcal{M}$ recovering the clean data manifold.

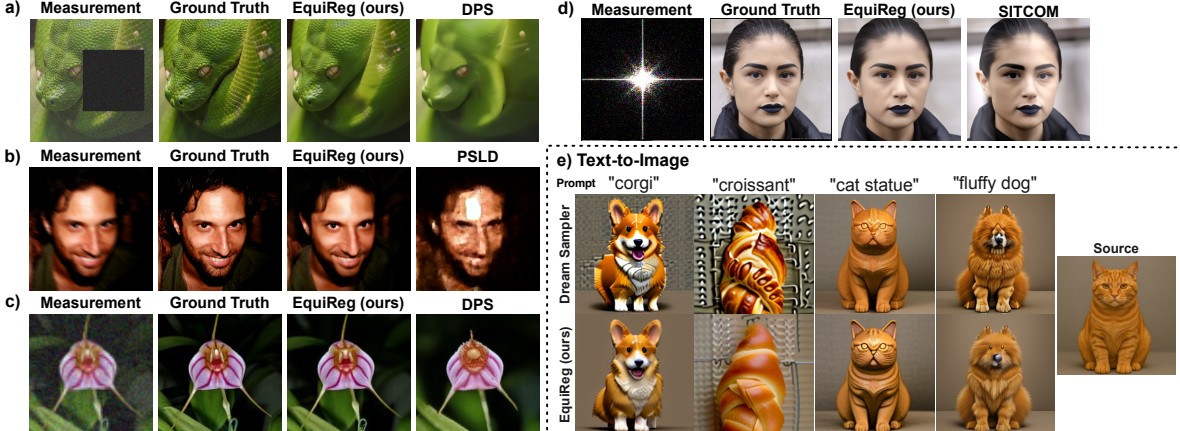

Figure 2: **EquiReg's broad applicability.** a-d) image restoration inverse problems and e) text-guided image generation, resulting in artifact reduction and more realistic generation. Here, EquiReg refers to our regularization being applied to the diffusion sampling method on the same row.

tractable when $t = 0$. To handle the likelihood for $t > 0$, many methods approximate the posterior $p_t(\boldsymbol{x}_0|\boldsymbol{x}_t)$ with the isotropic Gaussian distribution (Zhang et al., 2025a), where the distribution expectation is computed using the optimal denoising score (Robbins, 1956). The Gaussian approximation can be inaccurate for complex distributions (Figure 3), leading to errors in likelihood computation, especially with point estimations (Chung et al., 2023). Since the posterior expectation is a conditional expectation, a linear combination of all possible $\boldsymbol{x}_0$, it may lie off the data manifold even when individual samples remain on it. These issues are further amplified in latent diffusion models (LDMs), introducing artifacts (Rout et al., 2023).

Prior work has attempted to address this challenge via projection-based (He et al., 2024; Zirvi et al., 2025) or decoupled optimization strategies (Zhang et al., 2025a), aimed at reducing the propagation of measurement errors during sampling. However, they rely on the isotropic Gaussian assumption, which can lead to failures on difficult tasks or at reduced sampling steps. While higher-order statistics reduces errors (Boys et al., 2024), most approaches employ such approximation for its efficiency, scalability, and simplicity (Alkhouri et al.,

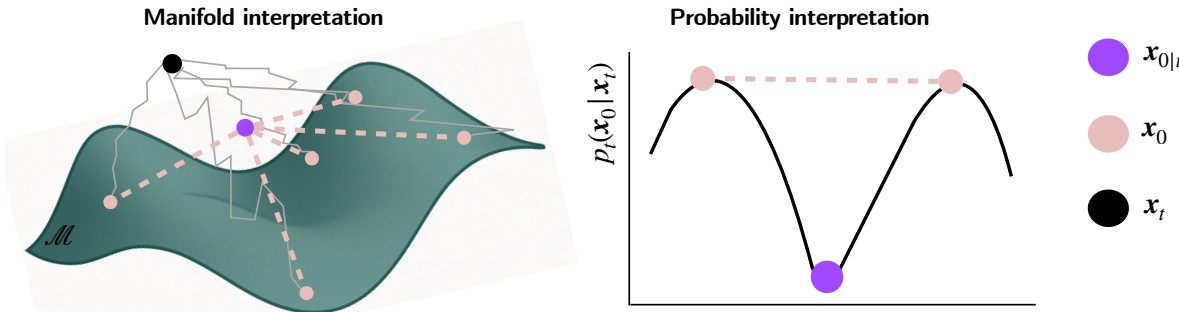

Figure 3: **Off-manifold posterior expectation.** This impacts the likelihood score $p_t(\boldsymbol{y}|\boldsymbol{x}_t) = \int p(\boldsymbol{y}|\boldsymbol{x}_0)p_t(\boldsymbol{x}_0|\boldsymbol{x}_t)\mathrm{d}\boldsymbol{x}_0$ computation achieved via isotropic Gaussian modelling of $p_t(\boldsymbol{x}_0|\boldsymbol{x}_t)$. The right panel is a schematic 1D illustration of $p_t(\boldsymbol{x}_0|\boldsymbol{x}_t)$ for a fixed $\boldsymbol{x}_t$, with the curve being the (bimodal) posterior density. The Tweedie posterior mean $\boldsymbol{x}_{0|t}$ is a value on the $\boldsymbol{x}_0$-axis, and the pink dotted segment connects the two modes to emphasize that $\boldsymbol{x}_{0|t}$ falls between them in a low-density region.

2025), often coupled with large-scale LDMs (Peebles & Xie, 2023). This raises a key question: how can we ensure that conditional diffusion samples remain on the data manifold under this approximation?

Equivariance offers a natural mechanism to keep sampling trajectories close to the data manifold. We address this challenge with a regularization scheme that leverages equivariance to improve posterior sampling by guiding trajectories toward symmetry-preserving solution spaces. Prior work has enforced equivariance directly within generation or denoising processes (Chen et al., 2023a; Terris et al., 2024), with extensions to probabilistic symmetries (Bloem-Reddy et al., 2020) enabling sample efficiency (Wang et al., 2024).

Our approach differs as follows: rather than strictly enforcing equivariance within denoising architectures, which can hinder tasks requiring symmetry breaking (Lawrence et al., 2025), we employ equivariance as a plug-in regularizer to guide diffusion trajectories toward the data manifold.

**Our contributions.** We propose *Equivariance Regularized* (EquiReg) diffusion, an equivariance-based regularization framework for solving inverse problems with diffusion models (Figure 1). EquiReg leverages equivariance to *regularize* likelihood-induced errors during posterior sampling, guiding diffusion trajectories toward more consistent, on-manifold solutions. Crucially, it employs *Manifold-Preferential Equivariant* (MPE) functions, which discriminate on-manifold from off-manifold data by exhibiting low equivariance error in-distribution and higher error out-of-distribution. We formalize that a regularizer for diffusion posterior sampling should capture such a global property, and MPE functions provide a principled way to direct sampling toward plausible solutions. This design makes EquiReg architecture-agnostic: the regularizer operates independently of the diffusion model itself. With a suitable MPE function, EquiReg improves performance across models, including those with equivariant scores, where likelihood guidance may otherwise push trajectories off the manifold.

We observe that many trained neural networks behave as MPEs: their equivariance error is small on the training or data manifold but grows off-manifold. This behavior arises in learned models trained with data augmentation, as well as in data with inherent symmetries such as those from physical systems. Rather than treating the degradation off-manifold as a limitation, we exploit it as a signal: equivariance error serves as a natural discriminator for identifying undesirable states during diffusion sampling. Building on this idea, we construct pre-trained MPEs as the foundation of our EquiReg loss. The choice of this function is independent of the denoiser in diffusion models and can be derived separately. For instance, if the diffusion architecture is itself equivariant to a set of group actions, it cannot be regularized via EquiReg using these same equivariances, as the MPE cannot discriminate between on- and off-manifold samples (the equivariance loss would remain low). In this case, a separate MPE with different equivariance properties can be trained. We explore several constructed and pre-trained MPE functions and show that EquiReg's performance is stable under different choices of MPE.

Beyond architectural flexibility, we systematically analyze the diversity properties of EquiReg-guided posterior sampling. We demonstrate that EquiReg achieves improved fidelity without posterior collapse, and as inverse problems become more ill-posed, our diversity metrics (intra-LPIPS and pixel-std) grow linearly with

task difficulty (Figure 14), indicating that EquiReg naturally expands posterior exploration with growing uncertainty rather than collapsing to a single mode.

We further demonstrate that MPE functions are not rare or specialized constructs, but emerge broadly across widely-used pre-trained networks. We empirically analyze multiple architectures, including the latent diffusion encoder (Rombach et al., 2022), CNN autoencoders trained with symmetry-based augmentations, pre-trained ResNet-50 (He et al., 2016), and CLIP encoders (Radford et al., 2021), and show that their equivariance error increases systematically as Gaussian noise pushes samples off the data manifold. We further show that using these MPE functions for EquiReg guidance improves solutions to inverse problems. Overall, this study shows that MPEs are widespread, and EquiReg uses this property as the basis for the regularization loss.

We validate the efficacy of EquiReg through extensive experiments across diverse diffusion models, inverse problems, and datasets. We demonstrate that EquiReg improves perceptual image quality and remains effective in cases where baselines fail. We show that EquiReg improves the performance of SITCOM (Alkhouri et al., 2025) and DPS (Chung et al., 2023) when the number of measurement consistency and sampling steps are reduced, thus moving toward more efficient diffusion-based solvers. The improvement is largest when applied to LDMs. EquiReg reduces failure cases, and consistently improves PSLD (Rout et al., 2023), ReSample (Song et al., 2023a), and DPS (Chung et al., 2023) on linear and nonlinear image restoration tasks. For example, EquiReg significantly improves the LPIPS (Song et al., 2023a) of ReSample by 51% for motion deblur and the FID of DPS (Chung et al., 2023) by 59% on super-resolution. We also include diversity analyses, demonstrating that EquiReg maintains diversity without collapse of single mode reconstruction.

We extend EquiReg's applicability to function-space diffusion models and demonstrate its added benefit for solving PDEs. EquiReg achieves a 7.3% relative reduction in the $\ell_2$ error of FunDPS (Mammadov et al., 2024a; Yao et al., 2025) on the Helmholtz equation and a 7.5% relative reduction on the Navier-Stokes equation. Lastly, we include preliminary experiments on EquiReg improving the realism and plausibility of text-guided image generation, emphasizing that the benefits of EquiReg extend beyond image restorations. Overall, the flexibility of EquiReg as a plug-in regularization framework suggests that its utility will extend well beyond the specific methods studied in this paper.

## 2 Preliminaries and Related Works

**Diffusion models.** Diffusion generative models (Ho et al., 2020; Song & Ermon, 2019; Sohl-Dickstein et al., 2015; Kadkhodaie & Simoncelli, 2021) are state-of-the-art in computer vision for image (Esser et al., 2024) and video generation (Brooks et al., 2024; Zhang et al., 2025b), with score-based methods (Song et al., 2021) being among the most widely used. Diffusion models generate data via a reverse noising process. The forward noising process transforms the data sample $\boldsymbol{x}_0 \sim p_{\text{data}}$ via a series of additive noise into an approximately Gaussian distribution ($p_{\text{data}} \to p_t \to \mathcal{N}(0, I)$ as $t \to \infty$), described by the stochastic differential equation (SDE) $d\boldsymbol{x} = -\frac{\beta_t}{2}\boldsymbol{x}dt + \sqrt{\beta_t}d\boldsymbol{w}$, where $\boldsymbol{w}$ is a standard Wiener process, and the drift and diffusion coefficients are parameterized by a monotonically increasing noise scheduler $\beta_t \in (0, 1)$ in time $t$ (Ho et al., 2020). Reversing the forward diffusion process is described by (Anderson, 1982)

$$d\boldsymbol{x} = [-\tfrac{\beta_t}{2}\boldsymbol{x} - \beta_t \nabla_{\boldsymbol{x}_t} \log p_t(\boldsymbol{x}_t)] \, dt + \sqrt{\beta_t}d\bar{\boldsymbol{w}} \tag{2}$$

with $dt$ moving backward in time or in discrete steps from $T$ to 0. This reverse SDE is used to sample data from the distribution $p_{\text{data}}$, where the unknown gradient $\nabla_{\boldsymbol{x}_t} \log p_t(\boldsymbol{x}_t)$ is approximated by a scoring function $s_\theta(\boldsymbol{x}_t, t)$, parameterized by a neural network and learned via denoising score matching methods (Hyvärinen & Dayan, 2005; Vincent, 2011). Solving inverse problems is described as a conditional generation where the data is sampled from the posterior $p(\boldsymbol{x}|\boldsymbol{y})$:

$$d\boldsymbol{x} = [-\tfrac{\beta_t}{2}\boldsymbol{x}dt - \beta_t(\nabla_{\boldsymbol{x}_t} \log p_t(\boldsymbol{x}_t) + \nabla_{\boldsymbol{x}_t} \log p_t(\boldsymbol{y}|\boldsymbol{x}_t))]dt + \sqrt{\beta_t}d\bar{\boldsymbol{w}} \tag{3}$$

For solving general inverse problems where the diffusion is *pre-trained* unconditionally, the prior score $\nabla_{\boldsymbol{x}_t} \log p_t(\boldsymbol{x}_t)$ can be estimated using $s_\theta(\boldsymbol{x}_t, t)$. However, the likelihood score $\nabla_{\boldsymbol{x}_t} \log p_t(\boldsymbol{y}|\boldsymbol{x}_t)$ is only known at $t = 0$, otherwise it is computationally intractable.

**Diffusion models for inverse problems.** Solving inverse problems with pre-trained diffusion models requires approximating the intractable likelihood score $\nabla_{\boldsymbol{x}_t} \log p_t(\boldsymbol{y}|\boldsymbol{x}_t)$. Training-free solvers dif-

fer in how they approximate $p_t(\boldsymbol{y}|\boldsymbol{x}_t)$ and combine it with the prior $p_t(\boldsymbol{x}_t)$ (Peng et al., 2024). Since $p_t(\boldsymbol{y}|\boldsymbol{x}_t) = \int p(\boldsymbol{y}|\boldsymbol{x}_0)p_t(\boldsymbol{x}_0|\boldsymbol{x}_t)\mathrm{d}\boldsymbol{x}_0$, the common choice is to approximate $p_t(\boldsymbol{x}_0|\boldsymbol{x}_t)$ by an isotropic Gaussian $\mathcal{N}(\boldsymbol{x}_{0|t}, r_t^2 \boldsymbol{I})$ (Chung et al., 2023; Song et al., 2023b; Zhu et al., 2023; Zhang et al., 2025a). With an optimal denoising score $s_\theta(\boldsymbol{x}_t, t)$, the posterior mean $\boldsymbol{x}_{0|t} := \mathbb{E}[\boldsymbol{x}_0|\boldsymbol{x}_t]$ follows from Tweedie's formula (Robbins, 1956; Miyasawa et al., 1961; Efron, 2011). While an MMSE estimate, $p_t(\boldsymbol{x}_0|\boldsymbol{x}_t)$ may not be concentrated around its mean for complex or multimodal distributions, leading to off-manifold solutions (see Figure 3).

**Equivariance.** Equivariance describes how functions transform predictably under group actions and provides a principled way to incorporate symmetry into deep learning (Bronstein et al., 2021). It has been applied to graphs (Satorras et al., 2021), convolutional networks (Cohen & Welling, 2016; Romero & Lohit, 2022), Lie groups for dynamical systems (Finzi et al., 2020), and diffusion models (Wang et al., 2024), with applications spanning molecular generation (Hoogeboom et al., 2022; Cornet et al., 2024), autonomous driving (Chen et al., 2023b), robotics (Brehmer et al., 2023), crystal structure prediction (Jiao et al., 2023), and audio inverse problems (Moliner et al., 2023). Equivariance guidance has also improved temporal consistency in video generation (Daras et al., 2024), and its role as a prior in inverse problems is theoretically supported in compressed sensing (Tachella et al., 2023).

**Definition 2.1** (Equivariance). *Let $G$ act on $\mathcal{Z}$ via $T_g : \mathcal{Z} \to \mathcal{Z}$ and on $\mathcal{X}$ via $S_g : \mathcal{X} \to \mathcal{X}$. A function $f : \mathcal{Z} \to \mathcal{X}$ is equivariant if for all $g \in G$ and $\boldsymbol{z} \in \mathcal{Z}$, $f(T_g(\boldsymbol{z})) = S_g(f(\boldsymbol{z}))$.*

An equivariant function preserves structure under group transformations (Definition 2.1). While prior work leverages exact equivariance to encode symmetries directly into neural networks, recent studies investigate approximate equivariance to relax strict symmetry assumptions that may not hold in real-world data, aiming to improve performance (Wang et al., 2022). These works introduce a formal definition of approximate equivariance (Definition 2.2) and an equivariance error to quantify deviations from perfect symmetry.

**Definition 2.2** (Approximate Equivariant Functions). *Let $G$ act on $\mathcal{Z}$ via $T_g : \mathcal{Z} \to \mathcal{Z}$ and on $\mathcal{X}$ via $S_g : \mathcal{X} \to \mathcal{X}$. A function $f : \mathcal{Z} \to \mathcal{X}$ is $\epsilon$-approximate equivariant if for all $g \in G$ and $\boldsymbol{z} \in \mathcal{Z}$, $\|S_g(f(\boldsymbol{z})) - f(T_g(\boldsymbol{z}))\| \le \epsilon$. The equivariance error of the function $f : \mathcal{Z} \to \mathcal{X}$ is defined as $\sup_{\boldsymbol{z},g} \|S_g(f(\boldsymbol{z})) - f(T_g(\boldsymbol{z}))\|$. Hence, $f$ is $\epsilon$-approximate equivariant iff its error $\le \epsilon$.*

Several recent works incorporate equivariance into image restoration and posterior sampling, and EquiReg differs from each in a specific way. Terris et al. (2024) use equivariance to symmetrize the denoiser inside a plug-and-play fixed-point iteration, producing a deterministic MAP-style estimate. Hoogeboom et al. (2022) build equivariance directly into the architecture of the diffusion model, producing denoisers that are *exactly* equivariant by construction (e.g., E(n)-equivariant networks for 3D molecule generation); a broader theoretical line of work studies probabilistic symmetries and invariant neural networks (Bloem-Reddy et al., 2020). Daras et al. (2024) use equivariance to enforce temporal consistency across video frames in image diffusion models. EquiReg sits in a different part of this design space. Equivariance enters as a *posterior-sampling regularizer*, applied through an MPE function that is *separate* from the diffusion denoiser and only *approximately* equivariant, and the equivariance error acts as a manifold-preference signal rather than as a hard constraint. This decoupling lets EquiReg be applied on top of any existing diffusion-based inverse solver without modifying the underlying denoiser, and accommodates symmetry-breaking tasks where architectural equivariance would be inappropriate.

We use the term manifold which refers to the data manifold hypothesis (see Assumption H.1 ) (Cayton et al., 2005) that assumes data is sampled from a low-dimensional manifold embedded in a high-dimensional space. This hypothesis is popular in machine learning (Bordt et al., 2023) and diffusion-based solvers (He et al., 2024; Chung et al., 2022b; 2023), supported by empirical evidence for imaging (Weinberger & Saul, 2006).

# 3 EquiReg: Equivariance Regularized Diffusion

**Regularizing diffusion models.** We begin by presenting a generalized regularization framework for improving diffusion-based inverse solvers. We focus on the property of *equivariance* and introduce a new class of functions whose equivariance errors are distribution-dependent (low for on- or near-manifold samples and high for off-manifold samples). We leverage these functions to regularize diffusion models, guiding sampling trajectories toward better inverse solutions.

This paper addresses the propagation error introduced by the approximation of posterior $p_t(\boldsymbol{x}_0|\boldsymbol{x}_t)$ by incorporating an explicit regularization term. The proposed framework is general and can be applied as plug-in on a wide range of pixel and latent-space diffusion models. Given $p_t(\boldsymbol{y}|\boldsymbol{x}_t) = \int p(\boldsymbol{y}|\boldsymbol{x}_0)p_t(\boldsymbol{x}_0|\boldsymbol{x}_t)\mathrm{d}\boldsymbol{x}_0$, let $\tilde{p}_t(\boldsymbol{x}_0|\boldsymbol{x}_t)$ denote an approximation of the posterior to make the likelihood tractable. We formulate the regularized reverse diffusion dynamics as

$$\mathrm{d}\boldsymbol{x} = [-\tfrac{\beta_t}{2}\boldsymbol{x}\mathrm{d}t - \beta_t \nabla_{\boldsymbol{x}_t}(\log p_t(\boldsymbol{x}_t) + \log \textstyle\int p(\boldsymbol{y}|\boldsymbol{x}_0)\tilde{p}_t(\boldsymbol{x}_0|\boldsymbol{x}_t)\mathrm{d}\boldsymbol{x}_0 - \mathcal{R}(\boldsymbol{x}_t))]\mathrm{d}t + \sqrt{\beta_t}\mathrm{d}\bar{\boldsymbol{w}}, \tag{4}$$

where $\mathcal{R}(\boldsymbol{x}_t)$ is the regularizer. Applying this to DPS (Chung et al., 2023) takes the form in Algorithm 1). This formulation brings us to the key contribution: how to design the regularizer to improve posterior sampling of diffusion models. Regularization in optimization aims to penalize undesired solutions, moving the algorithm towards solutions with low regularizer value. In the context of diffusion models for sampling from data distributions, we interpret an ideal regularizer as follows: an ideal regularizer should yield low values for on-manifold and high values for off-manifold samples, enabling accurate posterior sampling even when the likelihood score is approximated.

---

**Algorithm 1** Equi-DPS for Inverse Problems.

**Require:** $T, \boldsymbol{y}, \{\zeta_t\}_{t=1}^T, \{\tilde{\sigma}_t\}_{t=1}^T, \boldsymbol{s}_\theta, \mathcal{R}(\cdot), \{\lambda_t\}_{t=1}^T$
1: $\boldsymbol{x}_T \sim \mathcal{N}(\boldsymbol{0}, \boldsymbol{I})$
2: **for** $t = T - 1$ **to** $0$ **do**
3: $\quad \hat{\boldsymbol{s}} \leftarrow \boldsymbol{s}_\theta(\boldsymbol{x}_t, t)$
4: $\quad \boldsymbol{x}_{0|t} \leftarrow \frac{1}{\sqrt{\bar{\alpha}_t}}(\boldsymbol{x}_t + (1 - \bar{\alpha}_t)\hat{\boldsymbol{s}})$
5: $\quad \boldsymbol{\epsilon} \sim \mathcal{N}(\boldsymbol{0}, \boldsymbol{I})$
6: $\quad \boldsymbol{x}'_{t-1} \leftarrow \frac{\sqrt{\alpha_t}(1-\bar{\alpha}_{t-1})}{1-\bar{\alpha}_t}\boldsymbol{x}_t + \frac{\sqrt{\bar{\alpha}_{t-1}}\beta_t}{1-\bar{\alpha}_t}\boldsymbol{x}_{0|t} + \tilde{\sigma}_t\boldsymbol{\epsilon}$
7: $\quad \boldsymbol{x}_{t-1} \leftarrow \boldsymbol{x}'_{t-1} - \zeta_t(\nabla_{\boldsymbol{x}_t}\|\boldsymbol{y} - \mathcal{A}(\boldsymbol{x}_{0|t})\|_2^2 + \lambda_t \nabla_{\boldsymbol{x}_t}\mathcal{R}(\boldsymbol{x}_t))$
8: **end for**
9: **return** $\boldsymbol{x}_0$

---

In terms of sampling dynamics, i.e., when applied at each reverse-diffusion step, the regularizer should effectively penalize trajectories leaving the data manifold and reinforce those aligned with high-probability regions. This motivates designing a regularizer that applies a global correction to the entire functional, in contrast to prior works that focus only on locally reducing likelihood error. The ideal property of a regularizer would be to produce high error on undesirable samples and low error on desirable samples.

We offer a Wasserstein-flow perspective on the role of regularization in conditional diffusion sampling. Reinterpreting the reverse conditional diffusion as a time-inhomogeneous Wasserstein gradient flow (Ferreira & Valencia-Guevara, 2018) identifies a functional whose minimization the ideal dynamics correspond to (Proposition G.1 in Appendix). Hence, our framework can be viewed as a practical realization that approximates the Wasserstein gradient flow associated with a reweighted version of this functional, which down-weights off-manifold regions (Proposition G.2 in Appendix). We provide this perspective as motivation rather than a rigorous derivation. Given the regularized framework, one can use any regularizer that down-weights off-manifold trajectories to improve solutions to diffusion-based inverse problems. Next, we instantiate one such regularizer based on notion of equivariance.

**Equivariance-based regularizer.** We use equivariance, a global property that enforces geometric symmetries to instantiate a regularizer to guide the diffusion process toward the data manifold. To realize this idea, we seek functions that exhibit approximate equivariance and discriminate on- from off-manifold samples.

We propose to quantify the equivariance of a function relative to a data distribution. Specifically, while the literature has primarily studied the equivariance properties of functions for general inputs, we propose a new definition for functions in which their equivariance error is distribution-dependent and defined under the support of an input data distribution (Definition 3.1).

**Definition 3.1** (Distribution-Dependent Equivariant Functions)**.** *Let $G$ act on $\mathcal{Z}$ via $T_g : \mathcal{Z} \to \mathcal{Z}$ and on $\mathcal{X}$ via $S_g : \mathcal{X} \to \mathcal{X}$. The equivariance error of the function $f : \mathcal{Z} \to \mathcal{X}$ under the distribution $p$ is defined as* $\sup_g \mathbb{E}_{\boldsymbol{z} \sim p}[\|S_g(f(\boldsymbol{z})) - f(T_g(\boldsymbol{z}))\|]$.

The equivariance error (Definition 3.1) is a *pointwise* property of a learned function $f$ at a sample $\boldsymbol{z}$, distinct from *distributional* invariance (whether the distribution itself is preserved under the group action). For instance, the distribution of i.i.d. Gaussian noise is invariant under arbitrary pixel permutations, yet a network trained on natural images is not pointwise equivariant on noise samples; different noise draws produce wildly different outputs. This definition enables us to define functions whose equivariance error can differentiate on-manifold samples from off-manifold ones. We aim to find functions whose equivariance error is low for on-manifold data and high elsewhere. The equivariance error is non-local, defined at the distribution level.

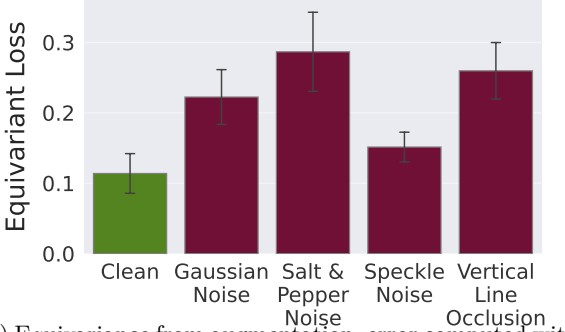
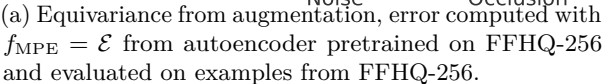
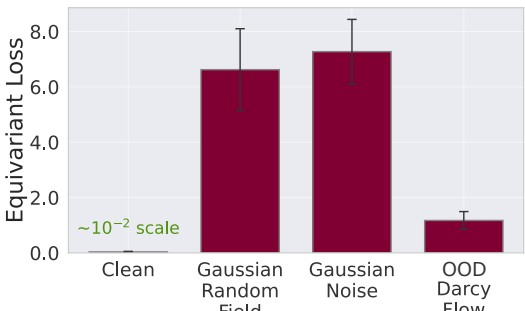

(a) Equivariance from augmentation, error computed with $f_{\text{MPE}} = \mathcal{E}$ from autoencoder pretrained on FFHQ-256 and evaluated on examples from FFHQ-256.

(b) Equivariance arising from data symmetries. Error computed with $f_{\text{MPE}} = \text{FNO}$ and evaluated on Navier-Stokes data.

Figure 4: **Equivariance error is consistently lower for clean vs perturbed examples.** In both subfigures the $y$-axis is the per-sample equivariance loss, averaged over images and over the chosen group actions. Perturbations are common image corruptions in (a) (Gaussian noise, salt-and-pepper, speckle, vertical occlusion) and physics inputs in (b) including an out-of-distribution Darcy-flow example.

When used to regularize the reverse conditional diffusion process, it is computed via local evaluations over the sampled data.

To define our method, we term a class of *manifold-preferential equivariant (MPE)* functions, whose equivariance error is lower for samples on the data manifold than for off-manifold samples. EquiReg is a regularization framework, not a manifold projection method. EquiReg penalizes states that deviate from symmetry-preserving regions; when an MPE function is used, these regions align with the data manifold. In practice, MPE functions can emerge from training with symmetry-preserving augmentation and from inherent data symmetries; we illustrate both with concrete examples below. MPE can emerge when functions are trained with symmetry-preserving mechanisms such as data augmentation. Prior work has studied equivariant properties of learned representations in deep networks (Lenc & Vedaldi, 2015), showing that data augmentations (Krizhevsky et al., 2012) and representation compression via reduced model capacity (Bruintjes et al., 2023) promote equivariant features even when equivariance is not explicitly built into the architecture. Importantly, the trained network is only approximately equivariant, and prior studies have noted that symmetry-preserving properties degrade for inputs deviating from in-distribution data (Azulay & Weiss, 2019). A few studies have leveraged this emergent MPE in trained networks for out-of-distribution detection (Zhou, 2022; Kaur et al., 2022; 2023).

To demonstrate the widespread MPE property of learned mappings, we have considered additional pre-trained models and quantified their equivariance loss for several datasets, i.e., natural images and corrupted ones (see Section I of Appendix.) Figure 4a illustrates the MPE property, emergent via training with augmentations, of $\mathcal{E}$-$\mathcal{D}$ of a pre-trained autoencoder, currently used in LDMs. Specifically, it shows that the equivariance error is lower for natural images and increases when images deviate from the clean data distribution. Based on Definition 3.1, we propose *Equi* loss using an MPE function $f$ for diffusion-based inverse solvers, defined as

$$\mathcal{R}_{f_{\text{MPE}}}(\boldsymbol{x}_t) = \|S_g(f_{\text{MPE}}(\boldsymbol{x}_{0|t})) - f_{\text{MPE}}(T_g(\boldsymbol{x}_{0|t}))\|_2^2 \tag{5}$$

where $\boldsymbol{x}_{0|t}$ and $\boldsymbol{z}_{0|t}$ are functions of $\boldsymbol{x}_t$ and $\boldsymbol{z}_t$, respectively.

We refer to the algorithm instantiated with this loss as *Equi-X*, where X is the underlying solver; *EquiCon-X* denotes the manifold-constrained variant that uses the equivariance error defined in Section H. The EquiReg gradient update can be applied every $P$ DDIM steps (period $P$, default $P = 1$); Table 13a characterizes performance as a function of $P$. The actions $T_g$ on the input and $S_g$ on the output are determined by a chosen symmetry group $G$. In practice, $G$ is selected to match the symmetries that the pre-trained $f_{\text{MPE}}$ approximately respects. For example, an autoencoder trained with horizontal-flip augmentation is approximately equivariant under that flip, and a neural operator trained on rotation-symmetric physics data is approximately equivariant under discrete rotations. The specific $f_{\text{MPE}}$, autoencoder source, and group used in our experiments are listed in Section 4, with broader guidelines on choosing $G$ in Section H.

MPE can also emerge due to symmetries present in the data itself during training. This often occurs in physics systems where coefficient functions, boundary values, and solution functions of PDEs remain valid under invertible coordinate transformations. Formally, let $\mathcal{G}(a) \mapsto u$ be a PDE operator mapping initial condition $a$

to solution $u$, and let $T_g$ and $S_g$ be invertible transformations that preserve PDE structure and boundary conditions. Then, $S_g(\mathcal{G}(a)) = \mathcal{G}(T_g(a))$. Neural operators (Kovachki et al., 2021), popular architectures for modelling physics, trained on PDEs with such inherent symmetries can learn equivariance properties. Figure 4b shows that an MPE function can be constructed using Fourier Neural Operators (FNOs (Li et al., 2021)) trained on non-augmented Navier–Stokes physics data. When a pre-trained FNO is used as the MPE function, the equivariance loss in equation 5 is lower for in-distribution data than for out-of-distribution data under reflection as the group action.

The key message from our MPE examples is that MPE behaviour naturally emerges when a function (e.g., a neural network) is trained with appropriate augmentations or when the data exhibit inherent symmetries. Our results extend this analysis and demonstrate that MPE functions are widely present and can be integrated into the EquiReg framework to improve posterior sampling (Table 7). We leverage this property to distinguish on-manifold from off-manifold samples and to regularize the posterior sampling trajectory toward high-probability regions. Finally, we note that the choice of symmetry group may often be a challenge depending on application domain, a shared challenge in the broader equivariance literature. We provide guidelines on how to choose symmetry groups in Section H with reference on automatic symmetry discovery from data (Zhou et al., 2021; Quessard et al., 2020; Dehmamy et al., 2021; Mohapatra et al., 2025).

## 4 Results

This section provides experimental results evaluating the performance of EquiReg on inverse problems, including linear and nonlinear image restorations and PDE solving. To fairly assess the impact of EquiReg, we adopt a paired comparison setting (e.g., PSLD vs. Equi-PSLD) across experiments, keeping all other factors such as architecture, training, and sampling fixed. This design ensures that any observed improvements can be attributed specifically to EquiReg rather than to differences in the underlying model or inference procedure. For fair comparison, we set a fixed seed for the initial noise and evaluate the impact of EquiReg under reduced measurement consistency and sampling steps, providing a path toward faster diffusion-based inversion. Results emphasize EquiReg's usefulness when the baseline performance deteriorates. We analyze the diversity trade-offs, and demonstrate the broad emergence of MPEs and their usefulness within EquiReg. Lastly, we provide preliminary analysis on EquiReg improving the realism of text-guided image generation.

**Image restoration tasks.** We evaluate EquiReg when applied to: SITCOM (Alkhouri et al., 2025), PSLD (Rout et al., 2023), ReSample (Song et al., 2023a), and DPS (Chung et al., 2023). We compare against manifold-preserving or geometry-constraint approaches including MCG (Chung et al., 2022b), MPGD-AE (He et al., 2024), and DiffStateGrad (Zirvi et al., 2025). We measure performance via perceptual similarity (LPIPS), distribution alignment (FID), pixel-wise fidelity (PSNR), and structural consistency (SSIM). We test EquiReg on a) the FFHQ $256 \times 256$ (Karras et al., 2021) and b) ImageNet $256 \times 256$ validation set (Deng et al., 2009). For pixel-based experiments, we use i) the pre-trained model from (Chung et al., 2023) on FFHQ, and ii) the pre-trained model from (Dhariwal & Nichol, 2021) on ImageNet. For latent diffusion experiments, we use i) the unconditional LDM-VQ-4 model (Rombach et al., 2022) on FFHQ, and ii) the Stable Diffusion v1.5 (Rombach et al., 2022) model on ImageNet.

Table 1: **EquiReg improves SITCOM under reduced measurement consistency steps ($K_{\text{meas}}$).** We reduce $K_{\text{meas}}$ and add an equal amount of EquiReg steps ($K_{\text{EquiReg}}$). Evaluated with 50 DDIM steps on motion deblur for FFHQ.

| $K_{\text{meas.}}$ | $K_{\text{EquiReg}}$ | PSNR↑ | SSIM↑ | Runtime (s) |
|---|---|---|---|---|
| 10 | N/A | 28.06 | 0.81 | 21.57 |
| 5 | 5 | **29.26** | **0.83** | **11.09** |
| 20 | N/A | 27.04 | 0.79 | 38.85 |
| 10 | 10 | **28.93** | **0.82** | **20.92** |
| 30 | N/A | 27.79 | 0.80 | 58.84 |
| 15 | 15 | **29.63** | **0.84** | **30.19** |
| 40 | N/A | **30.40** | **0.85** | 78.08 |
| 20 | 20 | 29.50 | 0.83 | **41.02** |
| 60 | N/A | 28.35 | 0.81 | 108.57 |
| 30 | 30 | **31.36** | **0.87** | **59.38** |

We evaluate EquiReg on linear and nonlinear restoration tasks for natural images (see Section E for task details). We adopt the publicly-released LDM autoencoder of Rombach et al. (2022) (LDM-VQ-4 for FFHQ, Stable Diffusion v1.5 for ImageNet) as our MPE function. For latent-space solvers (Equi-PSLD, Equi-ReSample), $f_{\text{MPE}}$ is the decoder $\mathcal{D}$ applied to $z_{0|t}$, so the equivariance loss in equation 5 becomes $\|S_g(\mathcal{D}(z_{0|t})) - \mathcal{D}(T_g(z_{0|t}))\|_2^2$; this $\mathcal{D}$ is the same one those solvers already use internally for sampling.

Table 3: **EquiReg for diffusion models on FFHQ**. $256 \times 256$ with $\sigma_{\boldsymbol{y}} = 0.05$.

| Method | Gaussian deblur | | | Motion deblur | | | Super-resolution ($\times 4$) | | | Box inpainting | | | Random inpainting | | |
|---|---|---|---|---|---|---|---|---|---|---|---|---|---|---|---|
| | LPIPS↓ | FID↓ | PSNR↑ | LPIPS↓ | FID↓ | PSNR↑ | LPIPS↓ | FID↓ | PSNR↑ | LPIPS↓ | FID↓ | PSNR↑ | LPIPS↓ | FID↓ | PSNR↑ |
| PSLD | 0.357 | 106.2 | 22.87 | **0.322** | **84.62** | 24.25 | 0.313 | 89.72 | 24.51 | 0.158 | 43.02 | 24.22 | 0.246 | 49.77 | 29.05 |
| Equi-PSLD | 0.344 | 94.09 | **24.42** | 0.338 | 99.14 | 24.83 | 0.289 | 90.88 | **26.32** | 0.098 | **31.54** | 24.19 | **0.188** | 41.61 | **30.43** |
| EquiCon-PSLD | **0.320** | **83.18** | 24.38 | **0.322** | 89.87 | **25.14** | **0.277** | **79.39** | 26.14 | **0.092** | 35.07 | **24.26** | 0.204 | **40.75** | 29.99 |

(a) Latent diffusion.

| Method | Gaussian deblur | | | Motion deblur | | | Super-resolution ($\times 4$) | | | Box inpainting | | | Random inpainting | | |
|---|---|---|---|---|---|---|---|---|---|---|---|---|---|---|---|
| | LPIPS↓ | FID↓ | PSNR↑ | LPIPS↓ | FID↓ | PSNR↑ | LPIPS↓ | FID↓ | PSNR↑ | LPIPS↓ | FID↓ | PSNR↑ | LPIPS↓ | FID↓ | PSNR↑ |
| DPS | 0.119 | 61.76 | 26.05 | 0.100 | 54.71 | 27.77 | 0.126 | 65.15 | 26.72 | 0.108 | 54.14 | 22.84 | 0.073 | 49.60 | 31.49 |
| Equi-DPS | **0.114** | **48.76** | **26.32** | **0.094** | **41.71** | **28.23** | **0.120** | **51.00** | **27.15** | 0.099 | 40.47 | 23.39 | **0.068** | 33.65 | **32.16** |
| DiffStateGrad-DPS | 0.128 | 52.73 | 26.29 | 0.118 | 50.14 | 27.61 | 0.186 | 73.02 | 24.65 | 0.114 | 47.53 | **24.10** | 0.107 | 49.42 | 30.15 |
| MCG | 0.340 | 101.2 | 6.72 | 0.702 | 310.5 | 6.72 | 0.520 | 87.64 | 20.05 | 0.309 | **40.11** | 19.97 | 0.286 | **29.26** | 21.57 |
| MPGD-AE | 0.150 | 114.9 | 24.42 | 0.120 | 104.5 | 25.72 | 0.168 | 137.7 | 24.01 | 0.138 | 248.7 | 21.59 | 0.172 | 339.0 | 25.22 |

(b) Pixel-based diffusion.

Table 4: **EquiReg for latent diffusion models on ImageNet**. $256 \times 256$ with $\sigma_{\boldsymbol{y}} = 0.05$.

| Method | Gaussian deblur | | Motion deblur | | Super-resolution (x4) | | Box inpainting | | Random inpainting | |
|---|---|---|---|---|---|---|---|---|---|---|
| | FID↓ | PSNR↑ | FID↓ | PSNR↑ | FID↓ | PSNR↑ | FID↓ | PSNR↑ | FID↓ | PSNR↑ |
| PSLD | 263.9 | 20.70 | 252.1 | 21.26 | 224.3 | 22.29 | 151.4 | 16.28 | 83.22 | 26.56 |
| EquiCon-PSLD | **214.5** | **22.01** | **196.3** | **22.69** | **198.5** | **22.34** | **137.6** | **19.25** | **65.14** | **27.03** |

For pixel-space solvers (Equi-DPS, Equi-SITCOM), $f_{\mathrm{MPE}}$ is the encoder $\mathcal{E}$ from the same autoencoder applied to $\boldsymbol{x}_{0|t}$, giving $\|S_g(\mathcal{E}(\boldsymbol{x}_{0|t})) - \mathcal{E}(T_g(\boldsymbol{x}_{0|t}))\|_2^2$. The EquiCon variants instead use the manifold-constrained error from Section H, which involves both $\mathcal{E}$ and $\mathcal{D}$. No additional pre-training is performed. For FFHQ, we use vertical reflection as the symmetry group, which preserves upright facial orientation. For ImageNet, we define a rotation group $G = \{0, \pi/2, \pi, 3\pi/2\}$, and uniformly at random select the group action for each sample. Finally, the loss functions given in Equation (5) are used to regularize. While our main experiment explore the reflection and rotation groups with small cardinality, EquiReg does not rely on full group coverage. Sampling even a sparse or randomly chosen subset of group actions is sufficient, as long as the function used for regularization exhibits the MPE property across the group (Table 9).

We characterize EquiReg under change of its period and $\lambda_t$. EquiReg can be applied efficiently with longer period' it preserves performance when applied with lower frequency (Table 13a); and it is robust to the choice of its hyperparameter $\lambda_t$, demonstrated on PSLD (Table 13b; see also Figure 20 for performance as a function of $\lambda$ and $\zeta$). Another advantage of EquiReg is that it allows the user to reduce the number of measurement consistency optimization steps, thereby introducing an implicit acceleration in solving the inverse problem. Table 1 shows that EquiReg regularization consistently enables SITCOM to achieve superior performance with significantly reduced runtime by using fewer measurement consistency steps.

Table 2: **EquiReg for ReSample on linear and nonlinear tasks.** FFHQ $256 \times 256$ with $\sigma_{\boldsymbol{y}} = 0.01$.

| Task | Method | LPIPS↓ | FID↓ | PSNR↑ | SSIM↑ |
|---|---|---|---|---|---|
| *Linear* | | | | | |
| Gaussian deblur | ReSample | 0.253 | 55.65 | 27.78 | 0.757 |
| | Equi-ReSample | 0.197 | 64.86 | **29.08** | **0.825** |
| | EquiCon-ReSample | **0.156** | **54.72** | 28.18 | 0.777 |
| Motion deblur | ReSample | 0.160 | 40.14 | 30.55 | 0.854 |
| | Equi-ReSample | 0.120 | 46.28 | **30.92** | **0.870** |
| | EquiCon-ReSample | **0.078** | **37.61** | 30.73 | 0.860 |
| Super-res. ($\times 4$) | ReSample | 0.204 | 40.46 | 28.02 | 0.790 |
| | Equi-ReSample | **0.098** | 43.56 | **29.74** | **0.849** |
| | EquiCon-ReSample | 0.112 | **40.38** | 28.27 | 0.801 |
| Box inpainting | ReSample | 0.198 | 108.30 | 19.91 | 0.807 |
| | Equi-ReSample | **0.150** | **59.69** | **22.56** | **0.832** |
| | EquiCon-ReSample | 0.171 | 110.70 | 21.04 | 0.815 |
| Random inpainting | ReSample | 0.115 | 36.12 | 31.27 | 0.892 |
| | Equi-ReSample | **0.047** | 29.88 | **31.47** | **0.908** |
| | EquiCon-ReSample | **0.047** | **28.81** | 31.21 | 0.904 |
| *Nonlinear* | | | | | |
| HDR | ReSample | 0.190 | **49.06** | **24.88** | **0.819** |
| | Equi-ReSample | **0.133** | 49.52 | 24.71 | 0.815 |
| | EquiCon-ReSample | 0.135 | 49.98 | 24.67 | 0.817 |
| Phase retrieval | ReSample | 0.237 | 97.86 | 27.61 | 0.750 |
| | Equi-ReSample | **0.155** | **85.22** | **28.16** | 0.770 |
| | EquiCon-ReSample | 0.159 | 88.75 | 28.11 | **0.774** |
| Nonlinear deblur | ReSample | 0.188 | 56.06 | 29.54 | 0.842 |
| | Equi-ReSample | 0.128 | 55.09 | 29.45 | 0.840 |
| | EquiCon-ReSample | **0.125** | **54.62** | **29.55** | **0.843** |

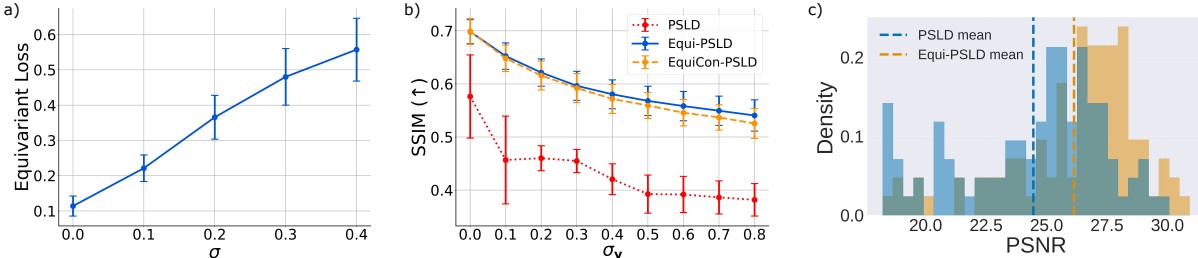

Figure 5: **EquiReg effectiveness and robustness across a range of measurement noise levels and regularization parameter.** (a) Equivariance error of the pre-trained LDM decoder $\mathcal{D}$ evaluated on FFHQ images with Gaussian noise added at varying standard deviations $\sigma$ (averaged over 100 images and the chosen symmetry group). (b) Reconstruction quality (SSIM) as a function of measurement noise level $\sigma_{\boldsymbol{y}}$, on super-resolution ($4\times$) with PSLD vs Equi-PSLD vs EquiCon-PSLD on FFHQ. (c) Overlaid distributions of per-image PSNR for DPS vs Equi-DPS on super-resolution on FFHQ; dashed lines mark the means. The shift in the Equi-DPS distribution toward higher PSNR illustrates the reduction of low-quality failure cases.

Table 3a, Table 4, and Table 2 highlights the benefits of EquiReg for latent models by consistently improving the performance of ReSample and PSLD across several tasks on FFHQ and ImageNet. We attribute this improvement in part to the reduction of failure cases (Figure 5d). EquiReg also significantly improves the performance of pixel-based methods (see Equi-DPS vs. DPS, Table 3d).

We observe that EquiReg achieves its largest improvements on perceptual metrics (FID and LPIPS), suggesting it generates more realistic images that lie closer to the data manifold (see Appendix E for supporting qualitative results). EquiReg improves performance under high measurement noise (Figure 5b). This result aligns with Figure 5a, which shows the equivariance error is lower on clean images than noisy ones, indicating that EquiReg enforces denoising. Lastly, we note that EquiReg is robust to regularizing hyperparameter $\lambda_t$ (Figure 5c, see Section C for details). For qualitative performance of EquiReg, see Figure 13.

**Solving PDEs from sparse observations.** EquiReg is evaluated on two important problems: the Helmholtz and Navier-Stokes equations (see Section F). The objective is to solve both forward and inverse problems in sparse sensor settings. The forward problem involves predicting the solution function or the final state using measurements from 3% of the coefficient field or the initial state. The inverse problem, conversely, aims to predict the input conditions from observations of 3% of the system's output. This task is challenging due to the nonlinearity of the equations, the complex structure of Gaussian random fields, and the sparsity of observations. Recent studies (Huang et al., 2024; Mammadov et al., 2024a; Yao et al., 2025) have demonstrated the superiority of diffusion models over deterministic single-forward methods for solving PDEs. DiffusionPDE (Huang et al., 2024) decomposes the conditional log-likelihood into a learned diffusion prior and a measurement score. FunDPS (Yao et al., 2025) extends the sampling process to a more natural infinite-dimensional spaces, achieving better accuracy and speed via function space models.

Table 5: **Solving PDEs from sparse observations.** Relative $\ell_2$ error (%, lower is better) on the Helmholtz and Navier-Stokes benchmarks, averaged over 100 test samples; bold marks the best result in each column.

| | Steps ($N$) | Helmholtz | | Navier-Stokes | |
|---|---|---|---|---|---|
| | | Forward | Inverse | Forward | Inverse |
| DiffusionPDE | 2000 | 12.64% | 19.07% | 3.78% | 9.63% |
| FunDPS | 500 | 2.13% | 17.16% | 3.32% | 8.48% |
| Equi-FunDPS | 500 | **2.12%** | **15.91%** | **3.06%** | **7.84%** |

We integrate EquiReg into the state-of-the-art FunDPS framework (Mammadov et al., 2024a; Yao et al., 2025), where the equivariance loss is computed using an FNO trained on the corresponding inverse problem. We employ reflection symmetry (i.e., flipping along the $y = x$ axis) and observe no significant performance differences when using alternative transformations such as rotations or alternating flips. Equi-FunDPS improves performance (Table 5), measured by relative $\ell_2$ error, across multiple tasks, particularly in inverse problems where a strong data prior is essential. Importantly, these results highlight that EquiReg is not limited to conventional neural networks and is naturally applicable to neural operators, enabling regularization in both finite-dimensional and function-space diffusion models.

Table 6: **Fidelity and diversity across inverse problems.** EquiReg preserves sampling diversity (20 test images with $K = 10$ samples per image.)

| Task | Method | Fidelity Metrics | | Diversity Metrics | |
|---|---|---|---|---|---|
| | | LPIPS↓ | FID↓ | Intra-LPIPS↑ | Pixel-Std↑ |
| Box inpainting | Equi-DPS | **0.112** | **59.70** | **0.118** | **10.59** |
| Gaussian deblur | Equi-DPS | **0.120** | **63.02** | 0.092 | 5.669 |
| Super-resolution (×4) | Equi-DPS | **0.703** | **87.52** | **0.187** | **23.52** |

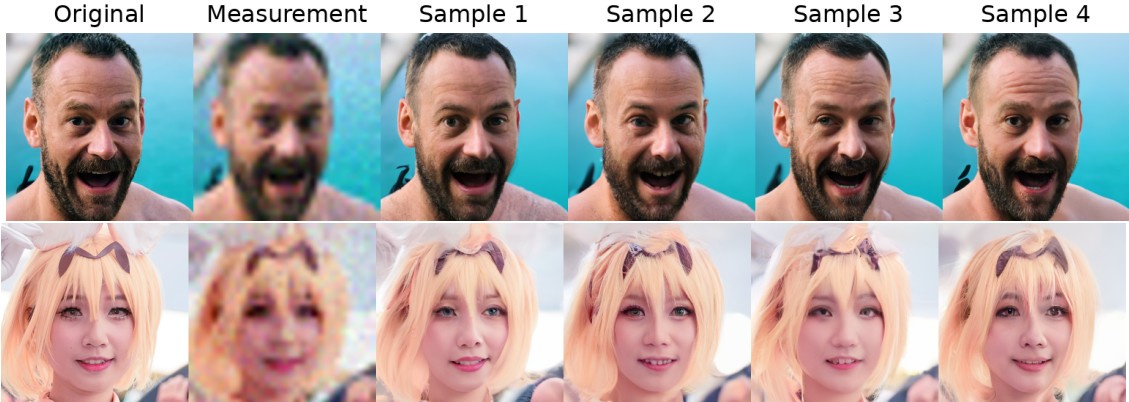

Figure 6: **Qualitative diversity examples for super-resolution.** We show $K = 4$ posterior samples for two test images. Samples differ in facial features (i.e., teeth in the first test image, eye color and eyelashes in the second test image) while maintaining high fidelity to the ground truth, demonstrating EquiReg generates diverse plausible reconstructions rather than collapsing to a single mode.

**Diversity analysis.**   To study posterior sampling diversity of EquiReg, we generated $K = 10$ posterior samples for 20 test images across three inverse problems of box inpainting, Gaussian deblurring, 4× super-resolution, and measured diversity using two complementary metrics: Intra-LPIPS for perceptual diversity and Pixel-Std for spatial diversity. Table 6 demonstrates that Equi-DPS does not show posterior collapse. We further investigated diversity scaling by varying box inpainting mask size from $128 \times 128$ to $192 \times 192$ pixels (Figure 14). Results show that diversity metrics increase linearly with task difficulty, demonstrating that Equi-DPS expands sampling as problems become more ill-posed rather than artificially constraining solutions. This linear relationship indicates consistent, non-collapsed posterior sampling across the difficulty spectrum. Lastly, Figures 6 and 15 provides qualitative confirmation through visual examples showing four posterior samples per image. Observable variations in facial features, expressions, and eye gaze validate our quantitative measurements, confirming EquiReg can generate genuinely diverse reconstructions rather than collapsing to a single solution (for additional discussion and analysis, see Section D).

**MPE behavior emerges across neural networks and improves EquiReg's performance.**   We empirically evaluate whether MPE behaviour naturally emerges across commonly used neural networks. We examine i) the emergence of MPE properties in different functions (neural networks) and ii) the effect of using these functions within EquiReg on identical inverse problem settings. For FFHQ and ImageNet, we analyze four function classes: the LDM encoder (Rombach et al., 2022), a CNN autoencoder trained with symmetry augmentations (flip for FFHQ and rotation for ImageNet), pre-trained ResNet-50 (He et al., 2016), and CLIP (Radford et al., 2021). Across all architectures, equivariance error increases systematically as Gaussian noise pushes samples off the data manifold, confirming clear MPE behaviour. The strength of this effect varies: the CNN autoencoder exhibits the strongest MPE signal, while the LDM encoder shows the weakest; ResNet-50 and CLIP lie between these extremes (see Figures 12 and 16). Notably, the strongest MPE behaviour of the CNN autoencoder is in line with our systematic guidelines for constructing MPE functions; this is precisely the regime where the train distribution matches the test.

Table 7: **MPE behaviour emerges across diverse network architectures and consistently improves diffusion-based inverse problems with EquiReg.** SITCOM motion deblurring on FFHQ 256 with $\lambda = 0.05$. Results are reported as mean (standard deviation).

| MPE function | PSNR | SSIM | LPIPS |
|---|---|---|---|
| None | 27.670 (1.343) | 0.790 (0.031) | 0.221 (0.040) |
| LDM Encoder (FFHQ) | 28.357 (1.379) | 0.806 (0.031) | 0.200 (0.036) |
| CNN Autoencoder (FFHQ) | **28.852 (1.376)** | **0.819 (0.044)** | **0.193 (0.033)** |
| Pretrained ResNet50 | 28.682 (1.388) | 0.811 (0.036) | 0.198 (0.036) |

We then integrate each function into EquiReg under identical inverse problem settings (FFHQ/ImageNet, DPS/SITCOM, super-resolution/motion deblurring). In all cases, EquiReg improves reconstruction quality over the corresponding baseline without regularization (Tables 7 and 12). These results demonstrate that i) MPE behaviour naturally arises in widely used pre-trained networks, and ii) EquiReg is robust to the choice of MPE function, even when the MPE signal is relatively weak. Notably, our main experiments use the weakest MPE (the LDM encoder), suggesting further gains are possible with stronger MPE constructions.

**Architectural and training factors behind MPE effectiveness.** The ranking observed in Tables 7 and 12 and Figure 16 reflects several architectural and training factors. The dominant factor is the match between the MPE function's training distribution and the inverse-problem distribution. The CNN autoencoder, trained directly on FFHQ with flip augmentation, produces the largest on-/off-manifold separation when evaluated on FFHQ-derived inverse problems, while the LDM encoder, trained on a comparable distribution but with milder augmentation and an additional latent-compression objective, exhibits a noticeably smaller separation. A secondary contribution comes from architectural inductive bias (Figure 21). The convolutional and ResNet autoencoders both show the MPE property; however, this property may not emerge in all architectures. Finally, it is important to distinguish equivariant from MPE functions. If a network is explicitly designed to be equivariant to a particular group transformation, then its equivariance behavior is independent of the input distribution. In such cases, the equivariance error will not distinguish between on- and off-manifold samples under that group, making it not suitable as an MPE function for that transformation (Figure 22).

## 5 Conclusion

We introduce *Equivariance Regularized* (EquiReg) diffusion for inverse problems. EquiReg regularizes sampling trajectories to stay closer to the data manifold, leveraging manifold-preferential equivariance (MPE): functions with low equivariance error on-manifold and high error off-manifold. Such functions can emerge in trained networks and can serve as plug-in regularizers without modifying the diffusion denoiser. EquiReg is agnostic across pixel- and latent-space diffusion models and maintains performance under reduced sampling, accelerating convergence. Across diverse inverse problems, it consistently improves perceptual and reconstruction metrics while reducing failure cases, highlighting its generality and efficiency.

**Beyond inverse problems.** EquiReg also applies to text-to-image guidance using DreamSampler (Kim et al., 2024), where it improves perceptual quality and reduces artifacts on FFHQ (see Section A.

EquiReg operates as a plug-in regularization framework and therefore builds upon the quality of the underlying diffusion solver. It does not modify the diffusion architecture itself, but instead improves sampling by guiding trajectories toward more consistent, on-manifold solutions. As a regularization mechanism, its impact is most pronounced in challenging regimes where likelihood guidance alone may degrade or become unstable. Applying EquiReg requires selecting appropriate symmetry groups and constructing suitable MPE functions for the task at hand. While we provide systematic guidelines for imaging and PDE settings, extending these constructions to new domains remains an important direction for future work; we have provided reference to prior work on how symmetries for various application can be learned and set. Finally, a deeper theoretical understanding of when MPE behaviour emerges in trained networks and how it may interact with joint training of diffusion models, remains an interesting avenue for further study.

**Broader Impact Statement.** On the positive side, high-fidelity image restoration can improve downstream tasks in medical imaging, remote-sensing and environmental monitoring (e.g., denoising satellite observations to track pollution or deforestation). Likewise, accelerated PDE-solving via learned diffusion priors may enable faster, more accurate simulations for climate modeling, fluid-dynamics research, and engineering design. On the other hand, robust reconstruction methods could be misappropriated for privacy-invasive surveillance or to create deceptive imagery. We emphasize that our method does not amplify these existing risks.

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

# Appendices for "EquiReg: Equivariance Regularized Diffusion for Inverse Problems"

We provide our source code when EquiReg. We will provide a publicly available source code upon acceptance. This supplementary materials contain the following:

- Section A includes additional experiments on text-to-image guidance. We regularize DreamSampler (Kim et al., 2024) with EquiReg for an improved performance (see Figures 7 to 11).

- Section B includes additional experiments on robustness including robustness to $\lambda_t$ and reduced number of measurent consistency steps.

- Section C includes qualitative analysis on the performance of methods with and without EquiReg. Results show a reduction of artifacts and an improved perceptual quality of the solution. This section also includes the equivariance error of a pre-trained encoder used in EquiReg (Figure 12a).

- Section D includes diversity experiments. Results show that EquiReg achieves improved fidelity without posterior collapse (Table 6, Figure 14, and Figure 15).

- Section E demonstrates EquiReg experimental setup and implementation for PSLD, ReSample, and DPS (Algorithms 2 to 6). It also contains information about the EquiReg hyperparameters for image restoration tasks.

- Section F contains information on the PDE reconstruction experiment. It discusses the equations along with implementation details and hyperparameters.

- Section G provides a Wasserstein-flow perspective on posterior sampling and the proofs of Proposition G.1 and Proposition G.2.

- Section H contains additional background information on solving inverse problems, vanishing-error autoencoders, and equivariance.

- Section I provides additional figures and tables.

- Section J discloses computing resources used to conduct the experiments.

- Section K credits code assets used for our experiments.

- Section L concludes the appendix with a "responsible release" statement.

The authors acknowledge the usage of LLMs on proofreading and improving the coherency of the manuscript. The authors have not used LLMs for content generation.

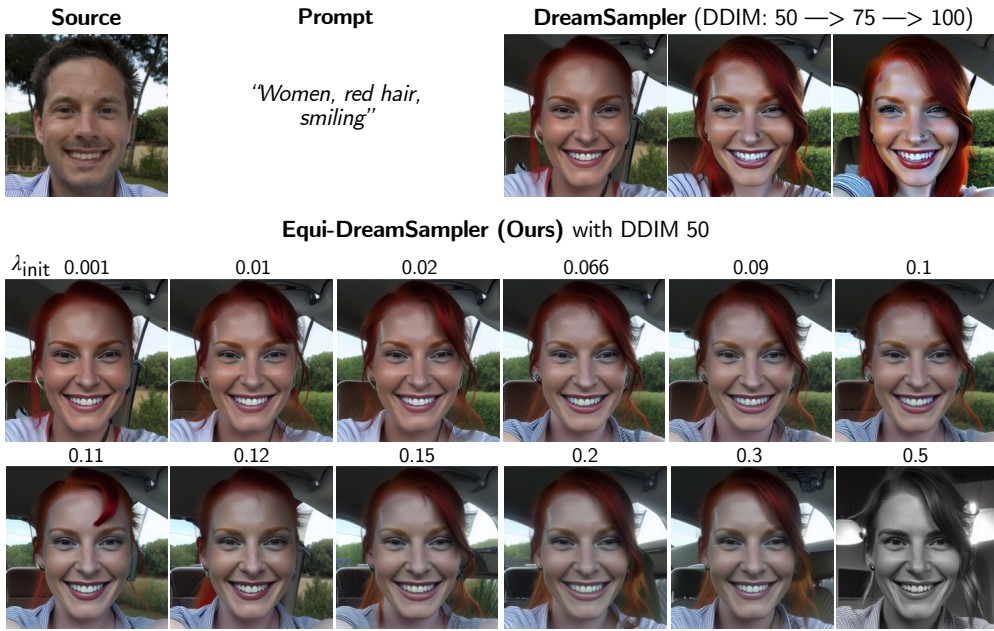

Figure 7: **Impact of EquiReg parameter $\lambda_t$, implicit acceleration, and introduction of more image details on FFHQ 512×512.** Women, red hair, smiling.

## A   EquiReg for Text-to-Image Guidance

Given a "source" image, DreamSampler (Kim et al., 2024) transforms it according to a text prompt. When applying EquiReg to DreamSampler, perceptual quality and reduced artifacts are improved in the generated images. Figure 2 shows a source "cat" transformed into a "corgi", where Equi-DreamSampler produces more realistic results and resolves anatomical inconsistencies (e.g., correcting a three-front-legged corgi to two front legs). We also observe an implicit acceleration effect when EquiReg is imposed (Figure 7). Equi-DreamSampler with 50 DDIM steps produces images comparable to DreamSampler with substantially more steps. Increasing the regularization strength $\lambda_t$ at fixed 50 DDIM steps results in effects similar to increasing DDIM steps in DreamSampler (from 50 to 75 to 100), suggesting that EquiReg guides sampling trajectories closer to the data manifold. These experiments illustrate that EquiReg is readily applicable beyond inverse problems, including text-to-image guidance, and are offered as an illustrative extension rather than a primary contribution.

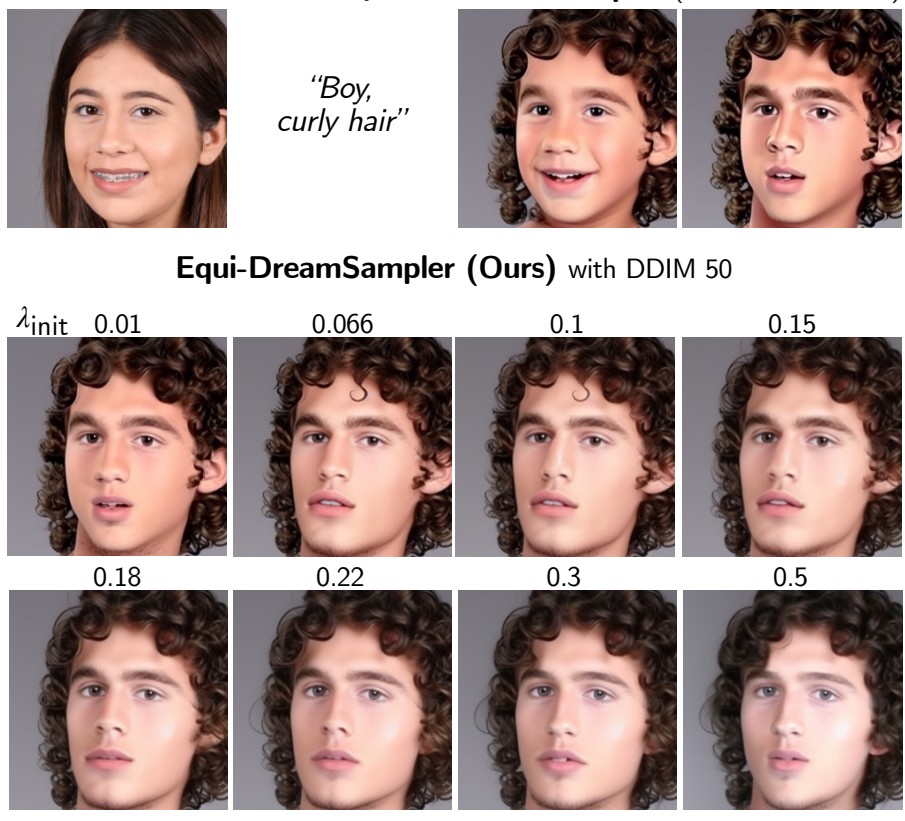

Figure 8: **Adding EquiReg into the text-to-image guidance method DreamSampler for improved performance on FFHQ 512×512.** Boy, curly hair.

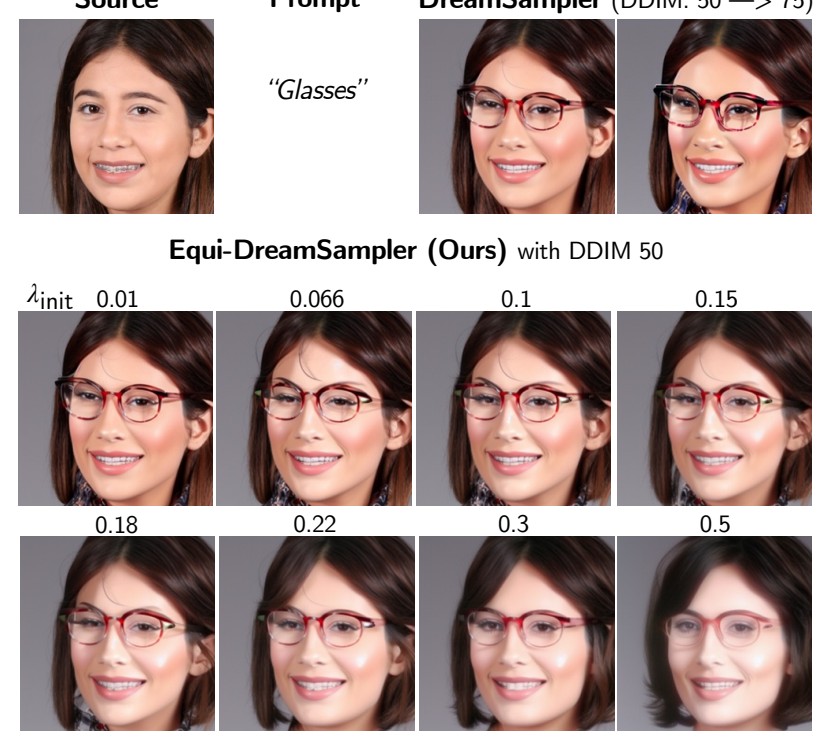

Figure 9: **Adding EquiReg into the text-to-image guidance method DreamSampler for improved performance.** Glasses.

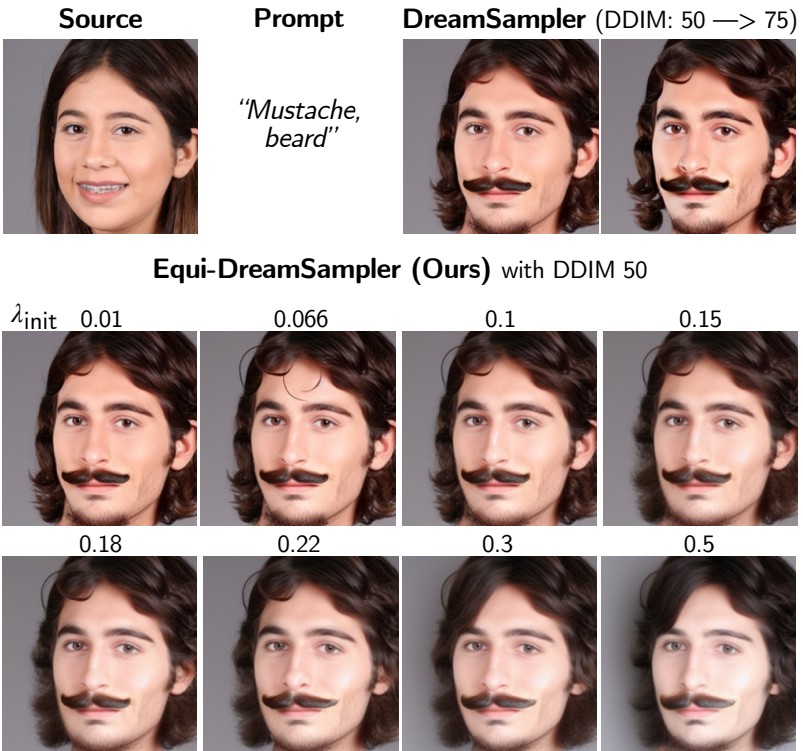

Figure 10: **Adding EquiReg into the text-to-image guidance method DreamSampler for improved performance.** Mustache, beard.

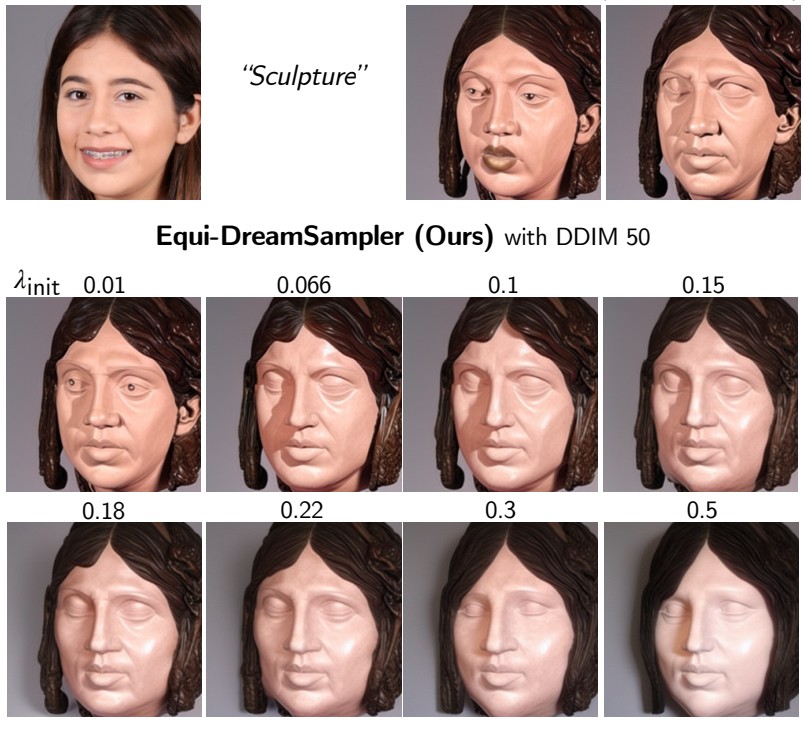

Figure 11: **Adding EquiReg into the text-to-image guidance method DreamSampler for improved performance.** Sculpture.

# B  Additional Experiments on Robustness

Table 8: **EquiReg improves SITCOM under reduced measurement consistency steps ($K_{\mathbf{meas}}$).** Motion deblur on FFHQ sampled with 50 DDIM steps.

| $K_{\text{meas.}}$ | $K_{\text{EquiReg}}$ | PSNR↑ | SSIM↑ | Runtime (s) |
|---|---|---|---|---|
| 10 | N/A | 28.06 | 0.81 | 21.57 |
| 10 | 1 | 28.71 | 0.82 | 21.07 |
| 5 | 5 | **29.26** | **0.83** | **11.09** |
| 20 | N/A | 27.04 | 0.79 | 38.85 |
| 20 | 1 | 28.54 | **0.82** | 37.74 |
| 10 | 10 | **28.93** | 0.82 | **20.92** |
| 30 | N/A | 27.79 | 0.80 | 58.84 |
| 30 | 1 | 28.35 | 0.81 | 55.51 |
| 15 | 15 | **29.63** | **0.84** | **30.19** |
| 40 | N/A | 30.40 | **0.85** | 78.08 |
| 40 | 1 | **30.58** | **0.85** | 69.83 |
| 20 | 20 | 29.50 | 0.83 | **41.02** |
| 60 | N/A | 28.35 | 0.81 | 108.57 |
| 60 | 1 | 27.02 | 0.78 | 95.62 |
| 30 | 30 | **31.36** | **0.87** | **59.38** |

Table 9: **EquiReg Effectiveness with Subset of Group Actions.**

| PSLD | | Equi-PSLD (90, 270 deg) | |
|---|---|---|---|
| PSNR↑ | SSIM↑ | PSNR↑ | SSIM↑ |
| 15.86 (1.19) | 0.77 (0.03) | 17.60 (1.60) | 0.79 (0.03) |

## C Visualizations for Image Restoration Experiments

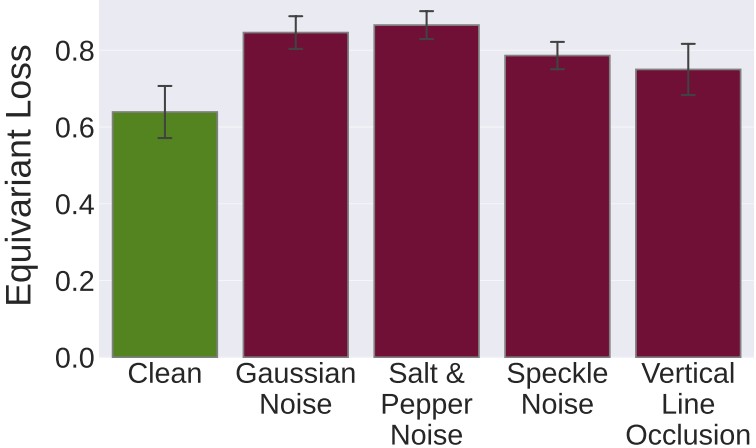

(a) The equivariance error of the encoder is lower on clean, natural images than corrupted ones.

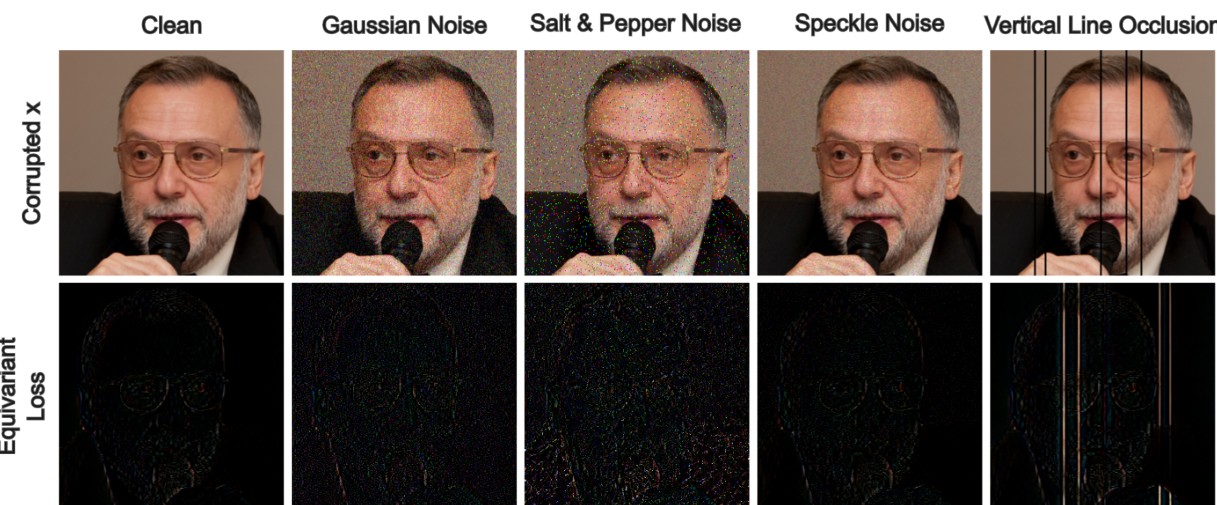

(b) Example visualizations of used images and corresponding equivariance error computed using the decoder (see Figure 4a).

Figure 12: **Training induced equivariance for a pre-trained function.**

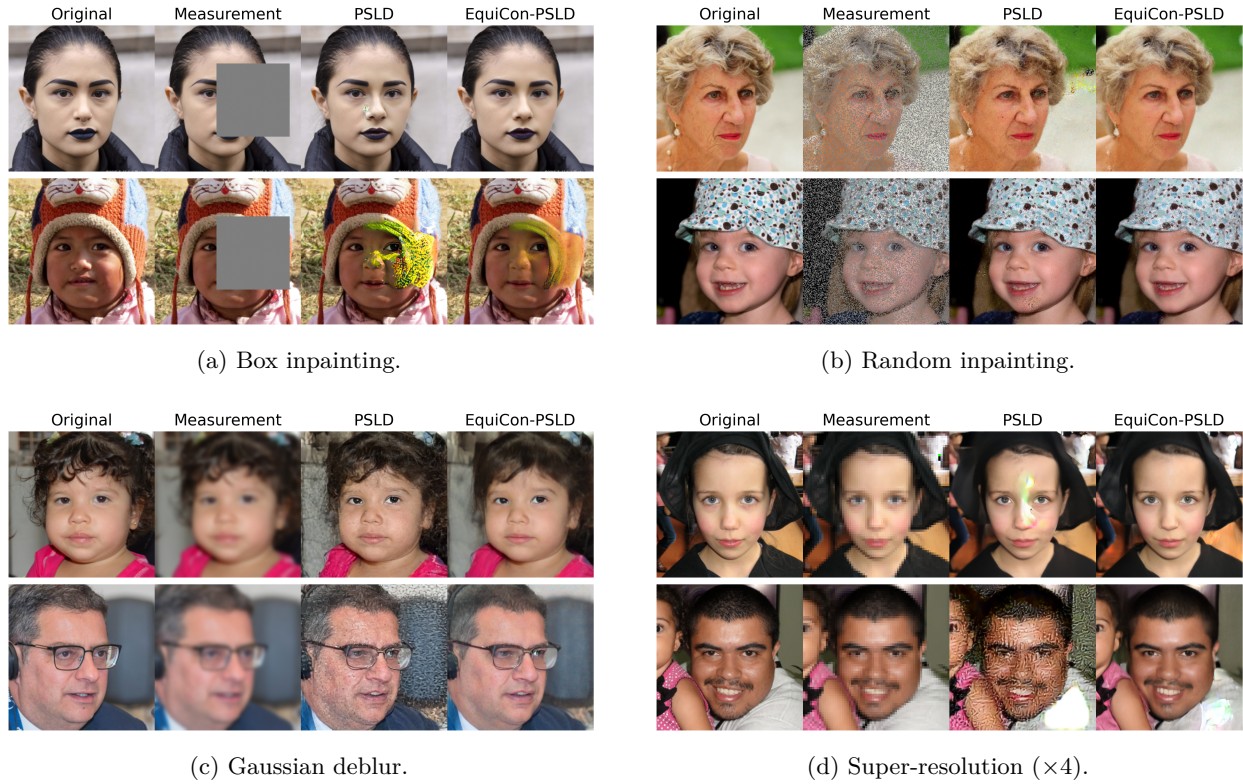

(a) Box inpainting.

(b) Random inpainting.

(c) Gaussian deblur.

(d) Super-resolution ($\times 4$).

Figure 13: **Qualitative comparison of EquiCon-PSLD and PSLD on FFHQ $256 \times 256$.**

## D  Diversity Analysis

In the Bayesian setting, the objective of solving inverse problems with diffusion models is to sample from high-probability regions of the posterior distribution. While the goal is not to maximize "diversity", the true diversity emerges when the posterior admits meaningful variability. In practice, diversity-related concerns in inverse problems arise when a method suffers from mode collapse, i.e., the sampler becomes biased and fails to explore multiple plausible modes of the posterior. Thus, the relevant question is whether a method properly explores the posterior rather than whether it maximizes diversity in an unconstrained sense.

Because closed-form posteriors are unavailable for real image restoration tasks, the standard practice in the diffusion inverse-problem literature is to evaluate diversity through variation among plausible reconstructions consistent with the measurement, without collapsing to a single solution. This is the notion of "diversity" our work adopts.

Given the goal of posterior sampling, EquiReg is not designed to maximize diversity for its own sake. Its objective is to incorporate data-inherent geometric structure (equivariance) to guide sampling toward high-probability regions of the posterior. Hence, diversity arises naturally from the ill-posedness of the inverse problem; it is a consequence of posterior uncertainty, not the goal of the regularizer.

To quantify this effect, in addition to reconstruction quality, we analyzed the diversity of posterior samples produced by EquiReg. We evaluate diversity metrics across multiple tasks and difficulty levels to characterize the sampling behavior of our method.

We emphasize that the intra-LPIPS and pixel-std metrics reported below are diagnostics for posterior collapse, not measures of reconstruction quality. The relevant comparison is between an EquiReg-regularized solver and the same solver without regularization, on the same task. Finding similar or higher diversity under EquiReg while fidelity improves therefore indicates that EquiReg does not artificially shrink posterior coverage, not

that diversity is itself a performance metric. A rigorous calibration analysis against a known posterior (for example, on a toy problem with closed-form posterior) is a worthwhile direction we leave to future work.

### D.1 Experimental Setup

To evaluate diversity, we generate multiple posterior samples and measure variation across these samples. For each of 20 test images, we generate K=10 reconstructions using different random seeds. We evaluate diversity using two complementary metrics: Intra-LPIPS, which measures perceptual diversity by computing the average LPIPS distance between all pairs of samples, and Pixel-Std, which measures spatial diversity through pixel-wise standard deviation across samples. Higher values for both metrics indicate greater diversity. For Intra-LPIPS, we compute distances for all $\binom{K}{2} = 45$ pairs per image and average across all test images. For Pixel-Std, we compute the standard deviation at each pixel location across the K samples, then average across all pixels and test images. We evaluate diversity across three inverse problems (box inpainting, Gaussian deblurring, and 4× super-resolution) comparing EquiReg against DPS (Chung et al., 2023) without equivariance regularization. To investigate how diversity scales with task difficulty, we additionally vary the inpainting mask size from $128 \times 128$ (standard) to $160 \times 160$ to $192 \times 192$ pixels.

### D.2 Results and Discussion

Table 6 shows that Equi-DPS achieves improved fidelity without posterior collapse across three inverse problems. For box inpainting and super-resolution, equivariance regularization improves both fidelity and diversity simultaneously. For Gaussian deblurring, Equi-DPS achieves 15-20% better fidelity while retaining 80-85% of baseline diversity, representing a modest but justified trade-off. These results demonstrate that equivariance constraints do not inherently suppress diversity; rather, they can guide sampling toward regions of higher data fidelity while maintaining posterior exploration.

Figure 14 reveals linear diversity scaling with task difficulty. Diversity metrics grow proportionally with task difficulty, indicating Equi-DPS naturally expands sampling as problems become more ill-posed. This linear relationship demonstrates stable, predictable behavior across difficulty levels without artificial diversity suppression. Figures 6 and 15 provide qualitative results.

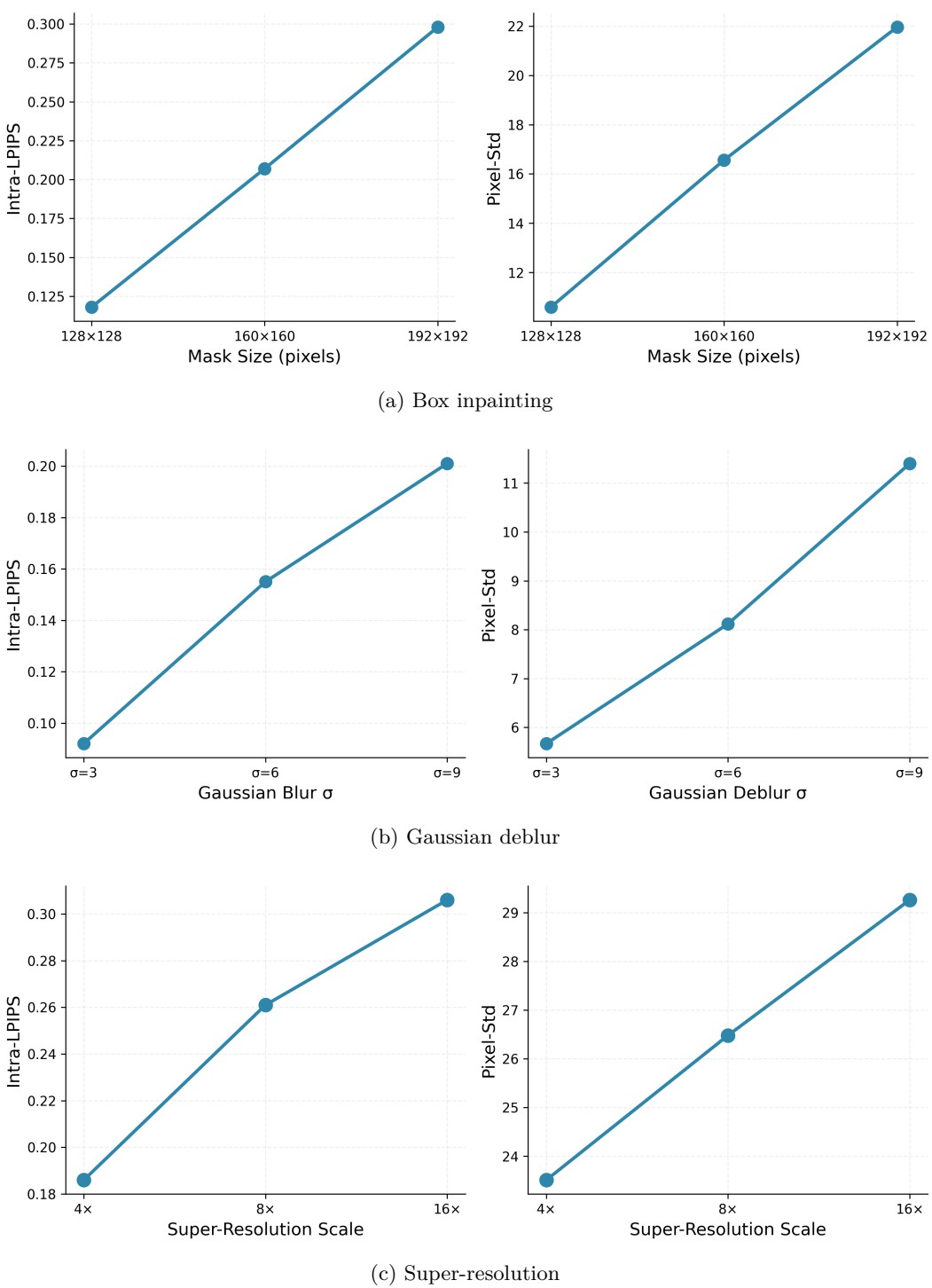

(a) Box inpainting

(b) Gaussian deblur

(c) Super-resolution

Figure 14: **Diversity vs task difficulty across three inverse problems.** As task difficulty increases (larger inpainting mask, stronger blur, higher SR scale), both diversity metrics increase proportionally, demonstrating that Equi-DPS maintains healthy posterior sampling behavior across a wide difficulty spectrum.

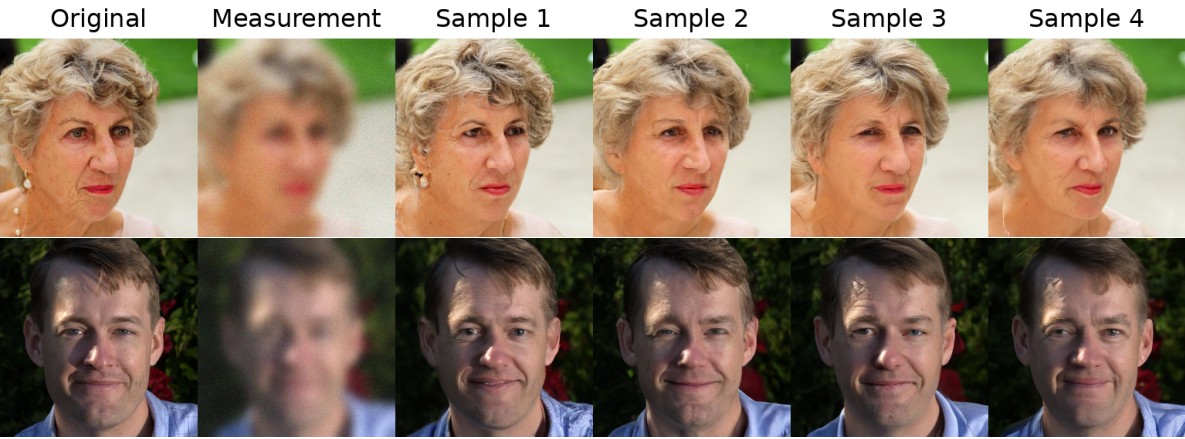

(a) Gaussian deblur

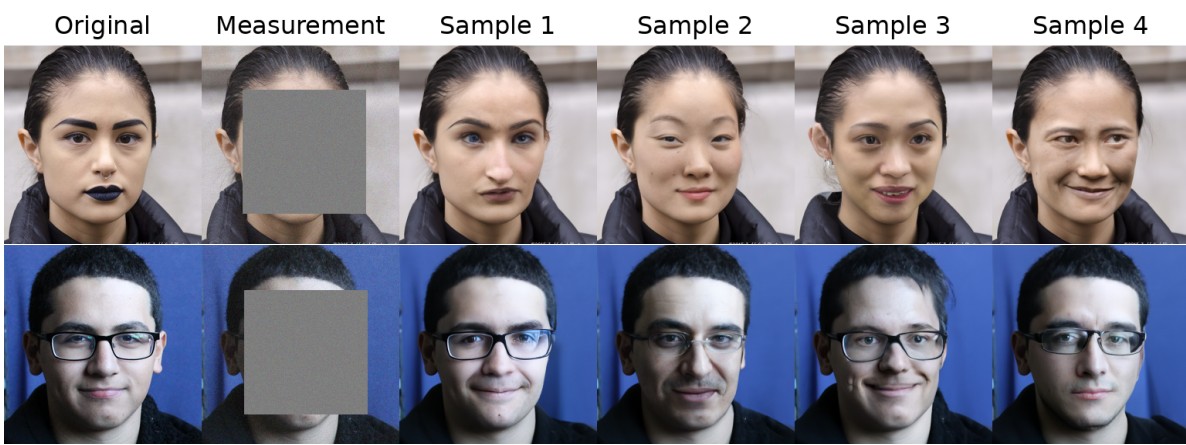

(b) Box inpainting

Figure 15: **Qualitative diversity examples across Gaussian deblur and box inpainting.** Each subfigure shows $K = 4$ posterior samples for two different test images. (a) Gaussian deblur: samples differ in facial expressions and accessories (i.e., earrings in first test image). (b) Box inpainting: Inputs were obstructed with $160 \times 160$ masks. Samples exhibit perceptually distinct facial features (i.e., expressions, eye gaze, facial structure). Across both tasks, EquiReg produces diverse plausible reconstructions rather than collapsing to a single mode.

### D.3 Conclusion

Finally, we highlight that EquiReg improves both fidelity and diversity on 2 of the 3 considered tasks, an encouraging outcome that is uncommon given the general behavior of classical regularizers. Hand-crafted regularizers such as TV and $\ell_1$ may suppress diversity by shrinking solutions toward simple structures. By contrast, EquiReg leverages data-dependent regularization that captures the richness and structural complexity of the underlying data manifold, enabling it to preserve manifold-consistent variability while suppressing implausible samples.

High diversity without fidelity is not meaningful for posterior sampling. A method that samples the entire solution space, including low-probability and artifacted regions, may score well on diversity but fail to provide useful reconstructions. Equi-DPS avoids this failure mode: it maintains meaningful diversity while reducing artifacts and improving perceptual quality. In the experiments conducted during the rebuttal, our goal was to demonstrate clearly that EquiReg preserves meaningful diversity, reflecting the posterior uncertainty, rather than unstructured or unconstrained variability.

# E   Implementation Details for Image Restoration Tasks

**Experimental Setup.**   We evaluate EquiReg on a variety of linear and nonlinear restoration tasks for natural images. We fix sets of 100 images from FFHQ and ImageNet as our validation sets. All images are normalized from $[0, 1]$. For the majority of experiments, we use noise level $\sigma_{\boldsymbol{y}} = 0.05$ (we indicate $\sigma_{\boldsymbol{y}}$ in our tables). For linear inverse problems, we consider (1) box inpainting, (2) random inpainting, (3) Gaussian deblur, (4) motion deblur, and (5) super-resolution. We apply a random $128 \times 128$ pixel box for box inpainting, and a 70% random mask for random inpainting. For Gaussian and motion deblur, we use kernels of size $61 \times 61$, with standard deviations of 3.0 and 0.5, respectively. For super-resolution, we downscale images by a factor of 4 using a bicubic resizer. For nonlinear inverse problems, we consider (1) phase retrieval, (2) nonlinear deblur, and (3) high dynamic range (HDR). We use an oversampling rate of 2.0 for phase retrieval, and due to instability of the task, we generate four independent reconstructions and take the best result (as also done in DPS (Chung et al., 2023), DAPS (Zhang et al., 2025a), and DiffStateGrad (Zirvi et al., 2025)). We use the default setting from (Tran et al., 2021) for nonlinear deblur, and a scale factor of 2 for HDR.

**Hyperparameters.**   Our method introduces a single hyperparameter $\lambda_t$ that controls the amount of regularization applied. Below we include a table detailing the use of this hyperparameter in the main experiments (Table 10). For all experiments, we keep $\lambda_t$ constant throughout iterations. We note that for the DPS method, the effective EquiReg regularization is given by $\zeta_t \lambda_t$. Therefore, scheduling the measurement guidance to decrease over time proportionally reduces the effective equivariance regularization as well.

Table 10: Equivariance regularization weight $\lambda_t$ used in main experiments.

| Method | Box Inpainting | Random Inpainting | Gaussian Deblur | Motion Deblur | Super-resolution ($\times 4$) |
|---|---|---|---|---|---|
| *FFHQ* $256 \times 256$ | | | | | |
| Equi-PSLD | 0.05 | 0.05 | 0.03 | 0.03 | 0.02 |
| EquiCon-PSLD | 0.01 | 0.01 | 0.01 | 0.01 | 0.01 |
| Equi-ReSample | 0.03 | 0.05 | 0.02 | 0.02 | 0.05 |
| EquiCon-ReSample | 0.001 | 0.001 | 0.001 | 0.001 | 0.001 |
| Equi-DPS | 0.0001 | 0.001 | 0.001 | 0.001 | 0.1 |
| *ImageNet* $256 \times 256$ | | | | | |
| EquiCon-PSLD | 0.0015 | 0.05 | 0.06 | 0.07 | 0.001 |

---

**Algorithm 2** Equi-PSLD for Image Restoration Tasks

**Require:** $T, \boldsymbol{y}, \{\eta_t\}_{t=1}^T, \{\gamma_t\}_{t=1}^T, \{\tilde{\sigma}_t\}_{t=1}^T$
**Require:** $\mathcal{E}, \mathcal{D}, \mathcal{A}\boldsymbol{x}_0^*, \mathcal{A}, \boldsymbol{s}_\theta, T_g$ and $S_g, \{\lambda_t\}_{t=1}^T$
 1:  $\boldsymbol{z}_T \sim \mathcal{N}(\boldsymbol{0}, \boldsymbol{I})$
 2:  **for** $t = T - 1$ **to** 0 **do**
 3:      $\hat{\boldsymbol{s}} \leftarrow \boldsymbol{s}_\theta(\boldsymbol{z}_t, t)$
 4:      $\boldsymbol{z}_{0|t} \leftarrow \frac{1}{\sqrt{\bar{\alpha}_t}}(\boldsymbol{z}_t + (1 - \bar{\alpha}_t)\hat{\boldsymbol{s}})$
 5:      $\boldsymbol{\epsilon} \sim \mathcal{N}(\boldsymbol{0}, \boldsymbol{I})$
 6:      $\boldsymbol{z}'_{t-1} \leftarrow \frac{\sqrt{\alpha_t}(1-\bar{\alpha}_{t-1})}{1-\bar{\alpha}_t}\boldsymbol{z}_t + \frac{\sqrt{\bar{\alpha}_{t-1}}\beta_t}{1-\bar{\alpha}_t}\boldsymbol{z}_{0|t} + \tilde{\sigma}_t\boldsymbol{\epsilon}$
 7:      $\boldsymbol{z}''_{t-1} \leftarrow \boldsymbol{z}'_{t-1} - \eta_t \nabla_{\boldsymbol{z}_t}\|\boldsymbol{y} - \mathcal{A}(\mathcal{D}(\boldsymbol{z}_{0|t}))\|_2^2$
 8:      $\boldsymbol{z}_{t-1} \leftarrow \boldsymbol{z}''_{t-1} - \gamma_t \nabla_{\boldsymbol{z}_t}\|\boldsymbol{z}_{0|t} - \mathcal{E}(\mathcal{A}^T\mathcal{A}\boldsymbol{x}_0^* + (\boldsymbol{I} - \mathcal{A}^T\mathcal{A})\mathcal{D}(\boldsymbol{z}_{0|t}))\|_2^2$
 9:      $\boldsymbol{z}_{t-1} \leftarrow \boldsymbol{z}_{t-1} - \lambda_t \nabla_{\boldsymbol{z}_t}\|S_g(\mathcal{D}(\boldsymbol{z}_{0|t})) - \mathcal{D}(T_g(\boldsymbol{z}_{0|t}))\|_2^2$
10:  **end for**
11:  **return** $\mathcal{D}(\boldsymbol{z}_{0|t})$

---

**PSLD.** We integrate EquiReg into PSLD by simply adding an additional gradient update step using our regularization term (Algorithms 2 and 3).

In our experiments, we use the official PSLD implementation from Rout et al. (2023), running with its default settings to reproduce the baseline results. We note that in our code, we do not square the norm when computing the gradient, aligning with PSLD's implementation.

---

**Algorithm 3** EquiCon-PSLD for Image Restoration Tasks

---

**Require:** $T, \boldsymbol{y}, \{\eta_t\}_{t=1}^T, \{\gamma_t\}_{t=1}^T, \{\tilde{\sigma}_t\}_{t=1}^T$
**Require:** $\mathcal{E}, \mathcal{D}, \mathcal{A}\boldsymbol{x}_0^*, \mathcal{A}, \boldsymbol{s}_\theta, T_g \text{ and } S_g, \{\lambda_t\}_{t=1}^T$
1: $\boldsymbol{z}_T \sim \mathcal{N}(\boldsymbol{0}, \boldsymbol{I})$
2: **for** $t = T - 1$ **to** $0$ **do**
3:  $\hat{\boldsymbol{s}} \leftarrow \boldsymbol{s}_\theta(\boldsymbol{z}_t, t)$
4:  $\boldsymbol{z}_{0|t} \leftarrow \frac{1}{\sqrt{\bar{\alpha}_t}}(\boldsymbol{z}_t + (1 - \bar{\alpha}_t)\hat{\boldsymbol{s}})$
5:  $\boldsymbol{\epsilon} \sim \mathcal{N}(\boldsymbol{0}, \boldsymbol{I})$
6:  $\boldsymbol{z}_{t-1}' \leftarrow \frac{\sqrt{\alpha_t}(1 - \bar{\alpha}_{t-1})}{1 - \bar{\alpha}_t}\boldsymbol{z}_t + \frac{\sqrt{\bar{\alpha}_{t-1}}\beta_t}{1 - \bar{\alpha}_t}\boldsymbol{z}_{0|t} + \tilde{\sigma}_t\boldsymbol{\epsilon}$
7:  $\boldsymbol{z}_{t-1}'' \leftarrow \boldsymbol{z}_{t-1}' - \eta_t \nabla_{\boldsymbol{z}_t}\|\boldsymbol{y} - \mathcal{A}(\mathcal{D}(\boldsymbol{z}_{0|t}))\|_2^2$
8:  $\boldsymbol{z}_{t-1} \leftarrow \boldsymbol{z}_{t-1}'' - \gamma_t \nabla_{\boldsymbol{z}_t}\|\boldsymbol{z}_{0|t} - \mathcal{E}(\mathcal{A}^T \mathcal{A}\boldsymbol{x}_0^* + (\boldsymbol{I} - \mathcal{A}^T\mathcal{A})\mathcal{D}(\boldsymbol{z}_{0|t}))\|_2^2$
9:  $\boldsymbol{z}_{t-1} \leftarrow \boldsymbol{z}_{t-1} - \lambda_t \nabla_{\boldsymbol{z}_t}\|\boldsymbol{z}_{0|t} - \mathcal{E}(S_g^{-1}(\mathcal{D}(T_g(\boldsymbol{z}_{0|t}))))\|_2^2$
10: **end for**
11: **return** $\mathcal{D}(\boldsymbol{z}_{0|t})$

---

**ReSample.** We integrate EquiReg into ReSample by adding our regularization term into the hard data consistency step (Algorithms 4 and 5). We note that the ReSample algorithm employs a two-stage approach; initially, it performs pixel-space optimization, and later it performs latent-space optimization. We apply EquiReg in the latent-space optimization stage.

In our experiments, we use the official ReSample implementation from Song et al. (2023a), running with its default settings to reproduce the baseline results.

---

**Algorithm 4** Equi-ReSample for Image Restoration Tasks

---

**Require:** Measurements $\boldsymbol{y}$, $\mathcal{A}(\cdot)$, Encoder $\mathcal{E}(\cdot)$, Decoder $\mathcal{D}(\cdot)$, Score function $\boldsymbol{s}_\theta(\cdot, t)$, Pretrained LDM Parameters $\beta_t$, $\bar{\alpha}_t$, $\eta$, $\delta$, Hyperparameter $\gamma$ to control $\sigma_t^2$, Time steps to perform resample $C$, $T_g \text{ and } S_g, \{\lambda_t\}_{t=1}^T$
1: $\boldsymbol{z}_T \sim \mathcal{N}(\boldsymbol{0}, \boldsymbol{I})$                   ▷ Initial noise vector
2: **for** $t = T - 1, \ldots, 0$ **do**
3:  $\boldsymbol{\epsilon}_1 \sim \mathcal{N}(\boldsymbol{0}, \boldsymbol{I})$
4:  $\hat{\boldsymbol{\epsilon}}_{t+1} = \boldsymbol{s}_\theta(\boldsymbol{z}_{t+1}, t+1)$               ▷ Compute the score
5:  $\hat{\boldsymbol{z}}_0(\boldsymbol{z}_{t+1}) = \frac{1}{\sqrt{\bar{\alpha}_{t+1}}}(\boldsymbol{z}_{t+1} - \sqrt{1 - \bar{\alpha}_{t+1}}\hat{\boldsymbol{\epsilon}}_{t+1})$    ▷ Predict $\hat{\boldsymbol{z}}_0$ using Tweedie's formula
6:  $\boldsymbol{z}_t' = \sqrt{\bar{\alpha}_t}\hat{\boldsymbol{z}}_0(\boldsymbol{z}_{t+1}) + \sqrt{1 - \bar{\alpha}_t - \eta\delta^2}\hat{\boldsymbol{\epsilon}}_{t+1} + \eta\delta\boldsymbol{\epsilon}_1$    ▷ Unconditional DDIM step
7:  **if** $t \in C$ **then**                   ▷ ReSample time step
8:    Initialize $\hat{\boldsymbol{z}}_0(\boldsymbol{y})$ with $\hat{\boldsymbol{z}}_0(\boldsymbol{z}_{t+1})$
9:    **for** each step in gradient descent **do**
10:     $\boldsymbol{g} \leftarrow \nabla_{\hat{\boldsymbol{z}}_0(\boldsymbol{y})}\frac{1}{2}\|\boldsymbol{y} - \mathcal{A}(\mathcal{D}(\hat{\boldsymbol{z}}_0(\boldsymbol{y})))\|_2^2 + \lambda_t \nabla_{\hat{\boldsymbol{z}}_0(\boldsymbol{y})}\|S_g(\mathcal{D}(\hat{\boldsymbol{z}}_0(\boldsymbol{y}))) - \mathcal{D}(T_g(\hat{\boldsymbol{z}}_0(\boldsymbol{y})))\|_2^2$
11:     Update $\hat{\boldsymbol{z}}_0(\boldsymbol{y})$ using gradient $\boldsymbol{g}$
12:    **end for**
13:   $\boldsymbol{z}_t = \text{StochasticResample}(\hat{\boldsymbol{z}}_0(\boldsymbol{y}), \boldsymbol{z}_t', \gamma)$        ▷ Map back to $t$
14:  **else**
15:   $\boldsymbol{z}_t = \boldsymbol{z}_t'$             ▷ Unconditional sampling if not resampling
16:  **end if**
17: **end for**
18: $\boldsymbol{x}_0 = \mathcal{D}(\boldsymbol{z}_0)$                 ▷ Output reconstructed image
19: **return** $\boldsymbol{x}_0$

---

---

**Algorithm 5** EquiCon-ReSample for Image Restoration Tasks

---

**Require:** Measurements $\boldsymbol{y}$, $\mathcal{A}(\cdot)$, Encoder $\mathcal{E}(\cdot)$, Decoder $\mathcal{D}(\cdot)$, Score function $\boldsymbol{s}_\theta(\cdot, t)$, Pretrained LDM Parameters $\beta_t$,
$\bar{\alpha}_t$, $\eta$, $\delta$, Hyperparameter $\gamma$ to control $\sigma_t^2$, Time steps to perform resample $C$, $T_g$ and $S_g$, $\{\lambda_t\}_{t=1}^T$
1: $\boldsymbol{z}_T \sim \mathcal{N}(\boldsymbol{0}, \boldsymbol{I})$ ▷ Initial noise vector
2: **for** $t = T - 1, \ldots, 0$ **do**
3: $\quad \boldsymbol{\epsilon}_1 \sim \mathcal{N}(\boldsymbol{0}, \boldsymbol{I})$
4: $\quad \hat{\boldsymbol{\epsilon}}_{t+1} = \boldsymbol{s}_\theta(\boldsymbol{z}_{t+1}, t+1)$ ▷ Compute the score
5: $\quad \hat{\boldsymbol{z}}_0(\boldsymbol{z}_{t+1}) = \frac{1}{\sqrt{\bar{\alpha}_{t+1}}}(\boldsymbol{z}_{t+1} - \sqrt{1 - \bar{\alpha}_{t+1}}\hat{\boldsymbol{\epsilon}}_{t+1})$ ▷ Predict $\hat{\boldsymbol{z}}_0$ using Tweedie's formula
6: $\quad \boldsymbol{z}'_t = \sqrt{\bar{\alpha}_t}\hat{\boldsymbol{z}}_0(\boldsymbol{z}_{t+1}) + \sqrt{1 - \bar{\alpha}_t - \eta\delta^2}\hat{\boldsymbol{\epsilon}}_{t+1} + \eta\delta\boldsymbol{\epsilon}_1$ ▷ Unconditional DDIM step
7: $\quad$ **if** $t \in C$ **then** ▷ ReSample time step
8: $\quad\quad$ Initialize $\hat{\boldsymbol{z}}_0(\boldsymbol{y})$ with $\hat{\boldsymbol{z}}_0(\boldsymbol{z}_{t+1})$
9: $\quad\quad$ **for** each step in gradient descent **do**
10: $\quad\quad\quad \boldsymbol{g} \leftarrow \nabla_{\hat{\boldsymbol{z}}_0(\boldsymbol{y})}\frac{1}{2}\|\boldsymbol{y} - \mathcal{A}(\mathcal{D}(\hat{\boldsymbol{z}}_0(\boldsymbol{y})))\|_2^2 + \lambda_t\nabla_{\hat{\boldsymbol{z}}_0(\boldsymbol{y})}\|\hat{\boldsymbol{z}}_0(\boldsymbol{y}) - \mathcal{E}(S_g^{-1}(\mathcal{D}(T_g(\hat{\boldsymbol{z}}_0(\boldsymbol{y})))))\|_2^2$
11: $\quad\quad\quad$ Update $\hat{\boldsymbol{z}}_0(\boldsymbol{y})$ using gradient $\boldsymbol{g}$
12: $\quad\quad$ **end for**
13: $\quad\quad \boldsymbol{z}_t = \text{StochasticResample}(\hat{\boldsymbol{z}}_0(\boldsymbol{y}), \boldsymbol{z}'_t, \gamma)$ ▷ Map back to $t$
14: $\quad$ **else**
15: $\quad\quad \boldsymbol{z}_t = \boldsymbol{z}'_t$ ▷ Unconditional sampling if not resampling
16: $\quad$ **end if**
17: **end for**
18: $\boldsymbol{x}_0 = \mathcal{D}(\boldsymbol{z}_0)$ ▷ Output reconstructed image
19: **return** $\boldsymbol{x}_0$

---

**DPS.** Similar to PSLD, we integrate EquiReg into DPS by simply adding an additional gradient update step using our regularization term (Algorithm 6).

In our experiments, we use the official DPS implementation from Chung et al. (2023), running with its default settings to reproduce the baseline results.

---

**Algorithm 6** Equi-DPS for Image Restoration Tasks

---

**Require:** $T, \boldsymbol{y}, \{\zeta_t\}_{t=1}^T, \{\tilde{\sigma}_t\}_{t=1}^T, \boldsymbol{s}_\theta, \mathcal{E}, T_g$ and $S_g$, $\{\lambda_t\}_{t=1}^T$
1: $\boldsymbol{x}_T \sim \mathcal{N}(\boldsymbol{0}, \boldsymbol{I})$
2: **for** $t = T - 1$ **to** $0$ **do**
3: $\quad \hat{\boldsymbol{s}} \leftarrow \boldsymbol{s}_\theta(\boldsymbol{x}_t, t)$
4: $\quad \boldsymbol{x}_{0|t} \leftarrow \frac{1}{\sqrt{\bar{\alpha}_t}}(\boldsymbol{x}_t + (1 - \bar{\alpha}_t)\hat{\boldsymbol{s}})$
5: $\quad \boldsymbol{\epsilon} \sim \mathcal{N}(\boldsymbol{0}, \boldsymbol{I})$
6: $\quad \boldsymbol{x}'_{t-1} \leftarrow \frac{\sqrt{\alpha_t}(1 - \bar{\alpha}_{t-1})}{1 - \bar{\alpha}_t}\boldsymbol{x}_t + \frac{\sqrt{\bar{\alpha}_{t-1}}\beta_t}{1 - \bar{\alpha}_t}\boldsymbol{x}_{0|t} + \tilde{\sigma}_t\boldsymbol{\epsilon}$
7: $\quad \boldsymbol{x}_{t-1} \leftarrow \boldsymbol{x}'_{t-1} - \zeta_t\nabla_{\boldsymbol{x}_t}\|\boldsymbol{y} - \mathcal{A}(\boldsymbol{x}_{0|t})\|_2^2$
8: $\quad \boldsymbol{x}_{t-1} \leftarrow \boldsymbol{x}_{t-1} - \lambda_t\nabla_{\boldsymbol{x}_t}\|S_g(\mathcal{E}(\boldsymbol{x}_{0|t})) - \mathcal{E}(T_g(\boldsymbol{x}_{0|t}))\|_2^2$
9: **end for**
10: **return** $\boldsymbol{x}_0$

---

**SITCOM.** We augment the original SITCOM algorithm by introducing an additional equivariant refinement stage at each reverse diffusion step. After completing the standard measurement and backward-consistency gradient updates, we perform a second optimization over the equivariance loss, enforcing consistency between $\mathcal{E}(T_g(v))$ and $T_g(\mathcal{E}(v))$ (Algorithm 7).

In our experiments, we use the official SITCOM implementation from Alkhouri et al. (2025), running with its default settings to reproduce the baseline results.

---

**Algorithm 7** Equi-SITCOM for Image Restoration Tasks

---

**Require:** Measurements $\mathbf{y}$, forward operator $\mathcal{A}(\cdot)$, pre-trained DM $\epsilon_\theta(\cdot, \cdot)$, diffusion steps $N$, schedule $\bar{\alpha}_i$, measurement gradient steps $K$, equivariant gradient steps $K_{\text{equi}}$, stop $\delta$, lr $\gamma$, reg. $\lambda$.

**Ensure:** Restored image $\hat{\mathbf{x}}$.

1: **Initialize** $\mathbf{x}_N \sim \mathcal{N}(\mathbf{0}, \mathbf{I})$, $\Delta t = \lfloor \frac{T}{N} \rfloor$.
2: **for** $i = N, N-1, \ldots, 1$ **do**                    ▷ Reducing diffusion sampling steps
3:     $\mathbf{v}_i^{(0)} \leftarrow \mathbf{x}_i$                    ▷ Init for closeness (C3)
4:     **for** $k = 1, \ldots, K$ **do**                    ▷ Adam on measurement/backward consistency (C1, C2)
5:         $\mathbf{v}_i^{(k)} \leftarrow \mathbf{v}_i^{(k-1)} - \gamma \nabla_{\mathbf{v}_i} \left[ \left\| \mathcal{A}\left( \frac{1}{\sqrt{\bar{\alpha}_i}} \left( \mathbf{v}_i - \sqrt{1 - \bar{\alpha}_i}\, \epsilon_\theta(\mathbf{v}_i, i\Delta t) \right) \right) - \mathbf{y} \right\|_2^2 + \lambda \|\mathbf{x}_i - \mathbf{v}_i\|_2^2 \right] \Big|_{\mathbf{v}_i = \mathbf{v}_i^{(k-1)}}$
6:         **if** $\left\| \mathcal{A}\left( \frac{1}{\sqrt{\bar{\alpha}_i}} \left( \mathbf{v}_i^{(k)} - \sqrt{1 - \bar{\alpha}_i}\, \epsilon_\theta(\mathbf{v}_i^{(k)}, i\Delta t) \right) \right) - \mathbf{y} \right\|_2^2 < \delta^2$ **then**
7:             **break**                    ▷ Prevent noise overfitting
8:         **end if**
9:     **end for**
10:    $\mathbf{v}_i^{(0)} \leftarrow \mathbf{v}_i^{(k)}$                    ▷ Initialize to optimized $\mathbf{v}_i$
11:    **for** $k = 1, \ldots, K_{\text{equi}}$ **do**
12:        $\mathbf{v}_i^{(k)} \leftarrow \mathbf{v}_i^{(k-1)} - \gamma \nabla_{\mathbf{v}_i} \left[ \left\| \mathcal{E}\left( T_g(\mathbf{v}_i^{(k)}) \right) - T_g\left( \mathcal{E}(\mathbf{v}_i^{(k)}) \right) \right\|_2^2 \right] \Big|_{\mathbf{v}_i = \mathbf{v}_i^{(k-1)}}$
13:        **if** $\left\| \mathcal{E}\left( T_g(\mathbf{v}_i^{(k)}) \right) - T_g\left( \mathcal{E}(\mathbf{v}_i^{(k)}) \right) \right\|_2^2 < \delta^2$ **then**
14:            **break**
15:        **end if**
16:    **end for**
17:    $\hat{\mathbf{v}}_i \leftarrow \mathbf{v}_i^{(k)}$                    ▷ Backward diffusion consistency (C2)
18:    $\hat{\mathbf{x}}_0' \leftarrow \frac{1}{\sqrt{\bar{\alpha}_i}} \left[ \hat{\mathbf{v}}_i - \sqrt{1 - \bar{\alpha}_i}\, \epsilon_\theta(\hat{\mathbf{v}}_i, i\Delta t) \right]$                    ▷ Backward consistency (C2)
19:    $\mathbf{x}_{i-1} \leftarrow \sqrt{\bar{\alpha}_{i-1}}\, \hat{\mathbf{x}}_0' + \sqrt{1 - \bar{\alpha}_{i-1}}\, \boldsymbol{\eta}_i, \quad \boldsymbol{\eta}_i \sim \mathcal{N}(\mathbf{0}, \mathbf{I})$                    ▷ Forward consistency (C3)
20: **end for**
21: **return** $\hat{\mathbf{x}} = \mathbf{x}_0$

---

## F  Experiment Setup for PDE Reconstructions

**Helmholtz equation.**  The Helmholtz equation represents wave propagation in heterogeneous media:

$$\nabla^2 u(x) + k^2 u(x) = a(x), \quad x \in (0,1)^2, \tag{6}$$

where $\nabla^2$ denotes the Laplacian operator, with $k = 1$ and $u|_{\partial\Omega} = 0$. Coefficient fields $a(x)$ are generated according to $a \sim \mathcal{N}(0, (-\Delta + 9\mathbf{I})^2)$. The forward problem is to predict the solution $u$ from sparse observations of the coefficient field $a$; the inverse problem is to predict $a$ from sparse observations of $u$. We note that this system has reflection equivariance along $x_1 = \frac{1}{2}, x_2 = \frac{1}{2}, x_1 = x_2$ and rotation equivariance by $\frac{\pi}{2}, \pi, \frac{3\pi}{2}$.

**Navier-Stokes equations.**  Following the methodology of (Li et al., 2020), we model the time evolution of a vorticity field, $u(x, t)$, governed by:

$$\partial_t u(x, t) + \boldsymbol{w}(x, t) \cdot \nabla u(x, t) = \nu \Delta u(x, t) + f(x), \quad x \in (0,1)^2, \, t \in (0, T], \tag{7}$$

$$\nabla \cdot \boldsymbol{w}(x, t) = 0, \quad x \in (0,1)^2, \, t \in [0, T], \tag{8}$$

$$u(x, 0) = a(x), \quad x \in (0,1)^2, \tag{9}$$

where $\boldsymbol{w}$ is the velocity field; $\nu = \frac{1}{1000}$, viscosity; and $f$, a fixed forcing term. The initial condition $a(x)$ is drawn from $\mathcal{N}(0, 7^{3/2}(-\Delta + 49\mathbf{I})^{-5/2})$ under periodic boundary conditions. The forcing term is $f(x) = 0.1\left(\sin(2\pi(x_1 + x_2)) + \cos(2\pi(x_1 + x_2))\right)$. We borrow the dataset from (Huang et al., 2024). The forward problem is to predict the final-time vorticity field $u(\cdot, T)$ from sparse observations of the initial condition $a$; the inverse problem is to predict $a$ from sparse observations of $u(\cdot, T)$. We note that this system has a reflection symmetry along the $x_1 = x_2$ axis.

**Implementation details.**  EquiReg, as a regularizer for diffusion posterior sampling, can be adapted to many inverse solvers in a plug-in manner. For PDE experiments, we use the same model weights and

configurations as FunDPS (Yao et al., 2025). Error rates are calculated using the $L^2$ relative error between the predicted and true solutions, averaged on 100 randomly selected test samples. We provide the information on the EquiReg scaling weights in Table 11.

Table 11: **EquiReg loss used in PDE experiments.**

|  | Helmholtz | | Navier-Stokes | |
| --- | --- | --- | --- | --- |
|  | Forward | Inverse | Forward | Inverse |
| EquiReg Norm Type | MSE | L2 | MSE | L2 |
| EquiReg Weight $\lambda$ | 100 | 100 | 100 | 1000 |

## G  A Wasserstein-flow perspective on posterior sampling

### G.1  Summary

We offer a Wasserstein-gradient-flow perspective on conditional diffusion sampling that provides intuition for why a manifold-preference regularizer such as EquiReg is a reasonable design. Proposition G.1 below identifies the specific functional that the ideal reverse-conditional dynamics minimize, and Proposition G.2 shows that EquiReg's practical regularized dynamics approximate the Wasserstein gradient flow of a reweighted version of this functional. We adopt this perspective as motivation rather than a rigorous derivation. Whether diffusion models follow exact Wasserstein dynamics remains an open problem (Zheng et al., 2025).

**Proposition G.1.** *Let $\rho(\boldsymbol{x}, t)$ be the distribution of $\boldsymbol{x}_{T-t}$ driven by the ideal reverse dynamics (eq. (3)). Then, the evolution of $\rho$ follows the Wasserstein-2 gradient flow associated with minimizing functional $\Phi(\rho, t)$ defined as $\beta_{T-t} \int [\rho \phi(\boldsymbol{x}, t) + \frac{1}{2}\rho \log \rho] d\boldsymbol{x}$, where $\phi(\boldsymbol{x}, t) = -(\log p_{T-t}(\boldsymbol{x}|\boldsymbol{y}) + \frac{1}{4}\|\boldsymbol{x}\|^2)$.*

The dynamics of $\rho$ remain the same if we replace $\phi(\boldsymbol{x}, t)$ with $\phi_C(\boldsymbol{x}, t) := \phi(\boldsymbol{x}, t) - C(t)$ for arbitrary temporal function $C(t)$, so we can without loss of generality assume $\phi_C(\boldsymbol{x}, t) < 0$ for all $\boldsymbol{x}$ and $t$ by choosing $C(t) > \sup_{\boldsymbol{x}} \phi(\boldsymbol{x}, t)$ on the support of $\rho(\cdot, t)$, provided $\phi(\cdot, t)$ is bounded above on this support. In practice, the density function $p_{T-t}$ is not available and thus $\phi_C(\boldsymbol{x}, t)$ is approximated as $\hat{\phi}$ with $p_{T-t}(\boldsymbol{x}_{T-t}|\boldsymbol{y}) \approx \tilde{C} p_{T-t}(\boldsymbol{x}_{T-t}) p(\boldsymbol{y}|\mathbb{E}[\boldsymbol{x}_0|\boldsymbol{x}_{T-t}])$ where $\tilde{C}$ only depends on $\boldsymbol{y}$.

Because the conditional expectation $\mathbb{E}[\boldsymbol{x}_0|\boldsymbol{x}_{T-t}]$ is a linear combination of all candidate $\boldsymbol{x}_0$, the approximation remains relatively accurate when $T-t$ is small (i.e., $\boldsymbol{x}_{T-t}$ stays close to the data manifold under low noise) but may incur high error for larger $T-t$, as shown in Figure 1b. To mitigate this, we reweight the contributions to the first term of $\Phi$, down-weighting unreliable estimates, and amplifying the reliable ones. The resulting reweighted functional is

$$\tilde{\Phi}(\rho, t) = \beta_{T-t}\left[\int \rho(\boldsymbol{x})\hat{\phi}_c(\boldsymbol{x}, t)e^{\frac{\mathcal{R}(\boldsymbol{x})}{\hat{\phi}_c(\boldsymbol{x}, t)}}d\boldsymbol{x} + \frac{1}{2}\int \rho(\boldsymbol{x})\log \rho(\boldsymbol{x})d\boldsymbol{x}\right], \tag{10}$$

where $\mathcal{R}(\boldsymbol{x})$ is a positive regularization that is nearly zero near the data manifold and much larger elsewhere. Intuitively, since $\hat{\phi}_C < 0$, the weight is nearly one for $\boldsymbol{x}$ near the data manifold and much smaller elsewhere.

**Proposition G.2.** *(Informal) The evolution of $\rho$, the probability distribution of $\boldsymbol{x}_{T-t}$ driven by the practical and regularized reverse dynamics (eq. (11)), is an approximation of the Wasserstein-2 gradient flow associated with minimizing $\tilde{\Phi}$.*

$$d\boldsymbol{x} = [-\frac{\beta_t}{2}\boldsymbol{x}dt - \beta_t \nabla_{\boldsymbol{x}_t}(\log p_t(\boldsymbol{x}_t) + \log \int p(\boldsymbol{y}|\boldsymbol{x}_0)\tilde{p}_t(\boldsymbol{x}_0|\boldsymbol{x}_t)d\boldsymbol{x}_0 - \mathcal{R}(\boldsymbol{x}_t))]dt + \sqrt{\beta_t}d\bar{\boldsymbol{w}} \tag{11}$$

**Connection to EquiReg**  Proposition G.2 identifies a role for the regularizer $\mathcal{R}(\boldsymbol{x})$, which reweights the gradient-flow functional $\Phi$ to down-weight regions where the Tweedie approximation is unreliable. EquiReg instantiates $\mathcal{R}(\boldsymbol{x})$ as a manifold-preferential equivariance loss, a nonnegative scalar that is small for inputs on the data manifold and grows as inputs drift off it.

### G.2 Preliminary and Notations

We first remind the readers of gradient flow under the Wasserstein-2 metric and introduce the notations related to the diffusion model.

**Wasserstein Gradient Flow**   Let $\mathcal{F}: \mathcal{P}_2(\mathbb{R}^d) \to \mathbb{R} \cup \{+\infty\}$ be a functional of probability distributions. The Wasserstein gradient flow of $\mathcal{F}$ is characterized by the minimizing movement scheme (also known as JKO scheme) introduced by (Jordan et al., 1998). For a fixed time step $\tau > 0$, the sequence $(\rho_k)_{k \in \mathbb{N}}$ of probability densities is defined recursively by:

$$\rho_{k+1} \in \arg \min_{\rho \in \mathcal{P}_2(\mathbb{R}^d)} \left\{ \frac{1}{2\tau} W_2^2(\rho, \rho_k) + \mathcal{F}(\rho) \right\},$$

where $W_2$ denotes the 2-Wasserstein distance, and each $\rho_k$ is a probability density representing the distribution at time $t = k\tau$. In the limit $\tau \to 0$, this discrete-time scheme recovers the continuous-time gradient flow of $\mathcal{F}$ under the $W_2$ metric.

**Diffusion Model**   A diffusion model defines a forward stochastic process $(\boldsymbol{x}_t)_{t \in [0,T]}$ governed by the Itô SDE:

$$\mathrm{d}\boldsymbol{x}_t = f(\boldsymbol{x}_t, t) \, \mathrm{d}t + \sqrt{\beta_t} \, \mathrm{d}\boldsymbol{w}_t, \tag{12}$$

where $\boldsymbol{w}_t$ is standard Brownian motion, $\beta_t > 0$ is a time-dependent variance schedule, and $f(\boldsymbol{x}, t)$ is a drift term. For instance, $f \equiv 0$ for a variance-exploding SDE and $f(\boldsymbol{x}, t) = -\frac{\beta_t}{2}\boldsymbol{x}$ for a variance-preserving SDE defined in (Song et al., 2021). In this work, we carry out our analysis under a more general setting.

**Assumption G.1.** *The drift term is a gradient field, $f(\boldsymbol{x}, t) = \nabla h(\boldsymbol{x}, t)$ for a scalar function $h$.*

This process progressively transforms an initial data distribution $\boldsymbol{x}_0 \sim p_0$ into a tractable reference distribution (e.g., approximately a Gaussian $\mathcal{N}(0, I)$) at time $T$.

Sampling is performed by simulating the *reverse-time SDE*:

$$\mathrm{d}\boldsymbol{x}_t = [f(\boldsymbol{x}_t, t) - \beta_t \nabla_{\boldsymbol{x}} \log p_t(\boldsymbol{x}_t)] \, \mathrm{d}t + \sqrt{\beta_t} \, \mathrm{d}\bar{\boldsymbol{w}}_t, \tag{13}$$

where $p_t$ is the marginal density of $\boldsymbol{x}_t$, and $\bar{\boldsymbol{w}}_t$ is a standard Brownian motion in reverse time.

In practice, the score function $\nabla_{\boldsymbol{x}} \log p_t(\boldsymbol{x})$ is approximated by a neural network $s_\theta(\boldsymbol{x}, t)$ trained to estimate the score of the forward process. For *conditional sampling*, where we sample $\boldsymbol{x}_0$ given some observed variable $y$, the score is replaced by $\nabla_{\boldsymbol{x}} \log p_t(\boldsymbol{x}|\boldsymbol{y})$ and decomposed as

$$\nabla_{\boldsymbol{x}} \log p_t(\boldsymbol{x}|\boldsymbol{y}) = \nabla_{\boldsymbol{x}} \log p_t(\boldsymbol{x}) + \nabla_{\boldsymbol{x}} \log p_t(\boldsymbol{y}|\boldsymbol{x}), \tag{14}$$

based on Bayes' rule.

To simplify notation in the sequel, we perform a time reparameterization $t = T - t'$, so that the reverse process is written as a forward SDE over $t \in [0, T]$:

$$\mathrm{d}\boldsymbol{x}_t = -[f(\boldsymbol{x}_t, T - t) - \beta_{T-t}[\nabla_{\boldsymbol{x}} \log p_{T-t}(\boldsymbol{x}_t) + \nabla_{\boldsymbol{x}} \log p_t(y|\boldsymbol{x}_t)]] \, \mathrm{d}t + \sqrt{\beta_{T-t}} \, \mathrm{d}\boldsymbol{w}_t, \tag{15}$$

This form describes the generative process as evolving forward from $t = 0$ to $t = T$, matching the usual direction of analysis in gradient flow frameworks.

### G.3 Proof of Proposition G.1

In this work, we consider Wasserstein gradient flow under the setting where the functional $\mathcal{F}$ depends on time.

**Lemma G.1.** *Consider a time-dependent functional $\mathcal{F}(\rho, t) = \int \rho(\boldsymbol{x}) V(\boldsymbol{x}, t) \mathrm{d}x + \int \alpha(t) \rho \log \rho \, \mathrm{d}x$. Then the particle description of Wasserstein-2 gradient flow associated with this functional derived by JKO scheme is*

$$\mathrm{d}\boldsymbol{x}_t = -\nabla V(\boldsymbol{x}_t, t) \mathrm{d}t + \sqrt{2\alpha(t)} \mathrm{d}\boldsymbol{w}_t. \tag{16}$$

*Proof.* Consider the following optimization

$$\min_{\rho'} \mathcal{F}(\rho', t + \Delta t) - \mathcal{F}(\rho, t) + \frac{1}{2\Delta t} W_2^2(\rho, \rho'), \tag{17}$$

where the change of density is restricted to the Liouville equation

$$\partial_t \rho = -\nabla \cdot (\rho v(\boldsymbol{x}, t)), \text{ and } \rho'(x) = \rho(x) - \Delta t \nabla \cdot (\rho(\boldsymbol{x}) v(\boldsymbol{x})) + o(\Delta t). \tag{18}$$

Using the static formulation of $W_2$ distance, we have

$$W_2^2(\rho, \rho') = \int \rho(\boldsymbol{x}) \|\boldsymbol{x} - T^*(\boldsymbol{x})\|^2 \, \mathrm{d}\boldsymbol{x} = \Delta t^2 \int \rho(\boldsymbol{x}) \|v^*(\boldsymbol{x})\|^2 \, \mathrm{d}\boldsymbol{x}, \tag{19}$$

where $T^*(\boldsymbol{x})$ is the optimal transport map, and $v^*(\boldsymbol{x})$ is the associated optimal velocity field.

Thus, we can rewrite the eq. (17) as

$$\inf_v \ \mathcal{F}(\rho, t) - \Delta t \int \nabla \cdot (\rho(\boldsymbol{x}) v(\boldsymbol{x})) \frac{\delta \mathcal{F}(\rho, t)}{\delta \rho}(\boldsymbol{x}) \, \mathrm{d}\boldsymbol{x} + \Delta t \int \left[ \rho(\boldsymbol{x}) \partial_t V(\boldsymbol{x}, t) + \dot{\alpha}(t) \rho \log \rho \right] \mathrm{d}\boldsymbol{x} \tag{20}$$

$$- \mathcal{F}(\rho, t) + \frac{\Delta t}{2} \int \rho(\boldsymbol{x}) \|v(\boldsymbol{x})\|^2 \, \mathrm{d}\boldsymbol{x}, \tag{21}$$

which simplifies to

$$\min_v \ \int \rho(\boldsymbol{x}) \left\langle v(\boldsymbol{x}), \nabla \frac{\delta \mathcal{F}(\rho, t)}{\delta \rho}(\boldsymbol{x}) \right\rangle \mathrm{d}\boldsymbol{x} + \frac{1}{2} \int \rho(\boldsymbol{x}) \|v(\boldsymbol{x})\|^2 \, \mathrm{d}\boldsymbol{x}, \tag{22}$$

since the last term in the first line of (20) does not depend on $v$. and further to

$$\min_v \ \int \rho(\boldsymbol{x}) \left\| v(\boldsymbol{x}) + \nabla \frac{\delta \mathcal{F}(\rho, t)}{\delta \rho}(\boldsymbol{x}) \right\|^2 \mathrm{d}\boldsymbol{x}. \tag{23}$$

From the optimality condition of the above problem, we obtain

$$v(\boldsymbol{x}, t) = -\nabla \frac{\delta \mathcal{F}(\rho, t)}{\delta \rho}(\boldsymbol{x}) = -(\nabla V(\boldsymbol{x}, t) + \alpha(t) \nabla \log \rho(\boldsymbol{x}, t)). \tag{24}$$

We note that By Hörmander's theorem, a smooth density $\rho(\boldsymbol{x}, t)$ exists for $t > 0$, ensuring that the above $v$ is well-defined. The corresponding evolution of probability density is

$$\partial_t \rho(\boldsymbol{x}, t) = -\nabla \cdot (\rho(\boldsymbol{x}, t) v(\boldsymbol{x}, t)) \tag{25}$$

$$= \nabla \cdot (\rho(\boldsymbol{x}, t)(\nabla V(\boldsymbol{x}, t) + \alpha(t) \frac{\nabla \rho(\boldsymbol{x}, t)}{\rho})) \tag{26}$$

$$= -\nabla \cdot (\rho(\boldsymbol{x}, t)(-\nabla V(\boldsymbol{x}, t)) + \alpha(t) \Delta \rho(\boldsymbol{x}, t)), \tag{27}$$

which is exactly the Fokker-Planck equation describing the evolution of the probability density describing the particles following

$$\mathrm{d}\boldsymbol{x}_t = -\nabla V(\boldsymbol{x}_t, t) \mathrm{d}t + \sqrt{2\alpha(t)} \mathrm{d}\boldsymbol{w}_t. \tag{28}$$

$$\square$$

Now we come back to Proposition G.1. From eq. (15) we know that choosing

$$V(\boldsymbol{x}, t) = h(\boldsymbol{x}, T - t) - \beta_{T-t}[\log p_{T-t}(\boldsymbol{x}) + \log p_{T-t}(\boldsymbol{y}|\boldsymbol{x})] \text{ and } \alpha(t) = \frac{\beta_{T-t}}{2} \tag{29}$$

in Lemma G.1 completes the proof, where $h$ is defined in Assumption G.1.

### G.4 Detailed version of Proposition G.2

In practice, one does not have access to $\log p_t(\boldsymbol{y}|\boldsymbol{x}_t)$ which appears in the reverse SDE. The most popular approach is do the following approximation,

$$p_t(\boldsymbol{y}|\boldsymbol{x}_t) = \int p(\boldsymbol{y}|\boldsymbol{x}_0)p(\boldsymbol{x}_0|\boldsymbol{x}_t)\mathrm{d}\boldsymbol{x}_0 = \mathbb{E}_{\boldsymbol{x}_0 \sim p(\boldsymbol{x}_0|\boldsymbol{x}_t)}[p(\boldsymbol{y}|\boldsymbol{x}_0)] \approx p(\boldsymbol{y}|\ \mathbb{E}[\boldsymbol{x}_0|\boldsymbol{x}_t]), \tag{30}$$

which can be interpreted as exchanging two operations, the conditional expectation and the measurement $p(\boldsymbol{y}|\cdot)$.

As discussed in the main text, since the conditional expectation is a linear combination over all possible values of $\boldsymbol{x}_0$, it may fall outside the data manifold, resulting in physically invalid samples. One of the central challenges in diffusion-based inverse sampling is guiding the sampling trajectory, generated by the reverse SDE dynamics, toward the data manifold. A common strategy is to incorporate regularization into the reverse SDE to encourage manifold adherence. In this work, building on the perspective of Wasserstein gradient flow as outlined above, we provide a novel interpretation of the role played by such regularization terms.

We show that the regularizer serves to reweight the contribution of different regions in the calculation of the underlying functional being minimized, $\Phi(\rho, t)$ defined in Proposition G.1. Specifically, it amplifies the influence of regions where the density estimate is reliable (typically near the data manifold), while down-weighting regions with poor approximation quality of based on eq. (30), often corresponding to off-manifold samples.

Following from what we have shown in the main text, $\Phi(\rho, t)$ has the form of $\beta_{T-t}\int[\rho\phi(\boldsymbol{x},t) + \frac{1}{2}\rho\log\rho]d\boldsymbol{x}$ for a function $\phi(\boldsymbol{x},t)$, which can be derived by (29). The $\log p_t(\boldsymbol{y}|\boldsymbol{x})$ term in (29) or $\nabla \log p_t(\boldsymbol{y}|\boldsymbol{x})$ term in (28), equivalently, is computed based on approximation (30). We denote the corresponding approximation of $\phi(\boldsymbol{x},t)$ as $\hat{\phi}(\boldsymbol{x},t)$. As discussed in the main text, we can assume without loss of generality that $\phi(\boldsymbol{x},t) < 0$ and $\hat{\phi}(\boldsymbol{x},t) < 0$. This is achieved by choosing the temporal shift $C(t)$ in $\phi_C(\boldsymbol{x},t) := \phi(\boldsymbol{x},t) - C(t)$ to satisfy $C(t) > \sup_{\boldsymbol{x}\in\mathrm{supp}(\rho(\cdot,t))}\phi(\boldsymbol{x},t)$ at each $t$, which leaves the Wasserstein gradient flow unchanged since the flow depends only on derivatives of $\phi$ with respect to $\boldsymbol{x}$; the same construction applies to $\hat{\phi}_C$. We assume $\phi(\cdot,t)$ and $\hat{\phi}(\cdot,t)$ are bounded above on the support of $\rho(\cdot,t)$ at each $t$, which is the standard regime for the analysis. We have

$$\hat{\Phi}(\rho, t) = \beta_{T-t}\Big[\int_{\boldsymbol{x}\in N(\mathcal{M})} \rho(\boldsymbol{x})\hat{\phi}(\boldsymbol{x},t)\mathrm{d}\boldsymbol{x} + \int_{\boldsymbol{x}\notin N(\mathcal{M})} \rho(\boldsymbol{x})\hat{\phi}(\boldsymbol{x},t)\mathrm{d}\boldsymbol{x} + \frac{1}{2}\int \rho\log\rho\,\mathrm{d}\boldsymbol{x}\Big], \tag{31}$$

where $N(\mathcal{M})$ denotes a neighborhood of the data manifold $\mathcal{M}$. Intuitively, we aim to focus on the contribution from regions near $\mathcal{M}$, which corresponds to the first term, while down-weighting the influence of points farther away, where the approximation tends to be unreliable. For instance, we can introduce two positive weights $A \gg B$ and adopt the modified functional

$$\tilde{\Phi}(\rho, t) = \beta_{T-t}\Big[A\int_{\boldsymbol{x}\in N(\mathcal{M})} \rho(\boldsymbol{x})\hat{\phi}(\boldsymbol{x},t)\mathrm{d}\boldsymbol{x} + B\int_{\boldsymbol{x}\notin N(\mathcal{M})} \rho(\boldsymbol{x})\hat{\phi}(\boldsymbol{x},t)\mathrm{d}\boldsymbol{x} + \frac{1}{2}\int \rho\log\rho\,\mathrm{d}\boldsymbol{x}\Big]. \tag{32}$$

In this work, we further generalize this idea and consider a continuous weight function,

$$\tilde{\Phi}(\rho, t) = \beta_{T-t}\Big[\int \rho(\boldsymbol{x})\hat{\phi}(\boldsymbol{x},t)\lambda(\boldsymbol{x})\mathrm{d}\boldsymbol{x} + \frac{1}{2}\int \rho\log\rho\,\mathrm{d}\boldsymbol{x}\Big], \tag{33}$$

where the non-negative weight $\lambda(\boldsymbol{x})$ is large for $\boldsymbol{x} \in N(\mathcal{M})$ and small elsewhere.

In practice, a nonnegative regularization function $\mathcal{R}(\boldsymbol{x})$ is introduced, ideally being nearly zero for $\boldsymbol{x}$ near the data manifold and much larger elsewhere. We consider the following modified functional with weight function $\lambda(\boldsymbol{x},t) := e^{\frac{\mathcal{R}(\boldsymbol{x})}{\hat{\phi}(\boldsymbol{x},t)}}$,

$$\tilde{\Phi}(\rho, t) = \beta_{T-t}\Big[\int \rho(\boldsymbol{x})\hat{\phi}(\boldsymbol{x},t)e^{\frac{\mathcal{R}(\boldsymbol{x})}{\hat{\phi}(\boldsymbol{x},t)}}\,\mathrm{d}\boldsymbol{x} + \frac{1}{2}\int \rho(\boldsymbol{x})\log\rho(\boldsymbol{x})d\boldsymbol{x}\Big]. \tag{34}$$

Note that $\hat{\phi} < 0$, we have that

$$\mathcal{R}(\boldsymbol{x}) \approx \begin{cases} 0, & \boldsymbol{x} \in N(\mathcal{M}) \\ \gg 1, & \boldsymbol{x} \text{ far away from } N(\mathcal{M}) \end{cases} \quad \Rightarrow \quad \lambda(\boldsymbol{x}, t) \approx \begin{cases} 1, & \boldsymbol{x} \in N(\mathcal{M}) \\ 0, & \boldsymbol{x} \text{ far away from } N(\mathcal{M}) \end{cases}.$$

Next, we consider practical algorithms based on this reweighted functional. In practice, we only have the score function instead of the function value of $\log p_{T-t}(\boldsymbol{x})$. Thus, the Wasserstein gradient flow associated with (34) is intractable since we cannot evaluate the weight function. We consider the following approximation based on $e^{\delta} \approx 1 + \delta$ when $\delta$ is sufficiently small. This is a first-order Taylor expansion in $\delta(\boldsymbol{x}, t) := \mathcal{R}(\boldsymbol{x})/\hat{\phi}(\boldsymbol{x}, t)$, accurate in the regime where $|\delta|$ is small. By construction, $\mathcal{R}(\boldsymbol{x})$ is small for $\boldsymbol{x}$ near the data manifold, and the regularizer in the resulting practical dynamics drives trajectories toward this region, so the typical $\delta$ encountered during sampling stays small.

$$\tilde{\Phi}(\rho, t) \approx \beta_{T-t} \left[ \int \rho(\boldsymbol{x}) \hat{\phi}(\boldsymbol{x}, t) \left( 1 + \frac{\mathcal{R}(\boldsymbol{x})}{\hat{\phi}(\boldsymbol{x}, t)} \right) \mathrm{d}\boldsymbol{x} + \tfrac{1}{2} \int \rho(\boldsymbol{x}) \log \rho(\boldsymbol{x}) d\boldsymbol{x} \right] \tag{35}$$

$$= \beta_{T-t} \left[ \int \rho(\boldsymbol{x}) \left( \hat{\phi}(\boldsymbol{x}, t) + \mathcal{R}(\boldsymbol{x}) \right) \mathrm{d}\boldsymbol{x} + \tfrac{1}{2} \int \rho(\boldsymbol{x}) \log \rho(\boldsymbol{x}) d\boldsymbol{x} \right]. \tag{36}$$

By Lemma G.1, the dynamics of $\boldsymbol{x}$ driven by the Wasserstein gradient flow associated with the approximated functional above is

$$\mathrm{d}\boldsymbol{x} = [-f(\boldsymbol{x}, T - t) - \beta_{T-t} \nabla_{\boldsymbol{x}} \left( \log p_{T-t}(\boldsymbol{x}) + \log \hat{p}_{T-t}(\boldsymbol{y}|\boldsymbol{x}) + \mathcal{R}(\boldsymbol{x}) \right)] \mathrm{d}t + \sqrt{\beta_{T-t}} \mathrm{d}\bar{\boldsymbol{w}}. \tag{37}$$

This completes the proof.

**Remark 1.** *Since $\hat{\phi} < 0$, and $e^A \geq 1 + A$ for any $A \in \mathbb{R}$, the dynamics derived by the approximated functional in (36) is evolving to minimize an upper bound of the reweighted functional $\tilde{\Phi}$.*

## H   Additional Background Information

**Solving inverse problems with deep learning prior to diffusion models.**   Earlier works (Metzler et al., 2016; Romano et al., 2017; Zhang et al., 2017; Metzler et al., 2017) used deep neural networks as denoisers to solve inverse problems. Furthermore, deep generative models such as variational autoencoders (VAEs) (Kingma, 2013), and generative adversarial networks (GANs) (Goodfellow et al., 2014) were employed. Notable applications include compressed sensing (Bora et al., 2017) and MRI (Jalal et al., 2021).

**Applications on diffusion models to solve inverse problems.**   Most popular applications include image restoration (Chung et al., 2023; 2022b; Kawar et al., 2022; Lugmayr et al., 2022; Saharia et al., 2022; Song et al., 2023a; Rout et al., 2023; Zhu et al., 2023; Zhang et al., 2025a; Zirvi et al., 2025), medical imaging (Song et al., 2022; Chung & Ye, 2022; Chung et al., 2022a; Hung et al., 2023; Dorjsembe et al., 2024; Li et al., 2024; Kazerouni et al., 2023; Bian et al., 2024), and solving partial differential equations (PDEs) (Isakov, 2006; Huang et al., 2024; Shysheya et al., 2024; Liu et al., 2023; Li et al., 2025; Baldassari et al., 2023; Mammadov et al., 2024a; Yao et al., 2025). On the methodology side, there has been numerous advancements (Chung et al., 2023; 2022b; Kawar et al., 2022; Lugmayr et al., 2022; Saharia et al., 2022; Song et al., 2023a; Rout et al., 2023; Zhu et al., 2023; Zhang et al., 2025a; Zirvi et al., 2025; Song et al., 2022; Chung & Ye, 2022; Chung et al., 2022a; Hung et al., 2023; Dorjsembe et al., 2024; Li et al., 2024; Kazerouni et al., 2023; Bian et al., 2024; Huang et al., 2024; Shysheya et al., 2024; Mammadov et al., 2024b; Cardoso et al., 2024).

**Resources for Definition H.2 on vanishing-error autoencoders.**   Manifold constrained distribution-dependent equivariance error uses the notion of *vanishing-error autoencoders* (Shao et al., 2018; Anders et al., 2020; He et al., 2024) (Definition H.1), also known as an asymptotically-trained autoencoder (Anders et al., 2020) or a perfect autoencoder (He et al., 2024). Vanishing-error autoencoders have previously been employed by diffusion-based inverse solvers to preserve the diffusion process on the manifold (He et al., 2024).

**Definition H.1** (Vanishing-Error Autoencoder)**.** *A vanishing-error autoencoder under the manifold $\mathcal{M}$ with encoder $\mathcal{E}: \mathcal{X} \to \mathcal{Z}$ and decoder $\mathcal{D}: \mathcal{Z} \to \mathcal{X}$ with $\mathcal{Z} = \mathbb{R}^k$ where $k < d$, has zero reconstruction error under the support of the data distribution $\mathcal{X}$, i.e., $\forall \boldsymbol{x} \in \mathcal{X} \subset \mathcal{M}, \boldsymbol{x} = \mathcal{D}(\mathcal{E}(\boldsymbol{x}))$. It follows that the decoder is surjective on the data manifold, $\mathcal{D}: \mathcal{Z} \to \mathcal{M}$ (He et al., 2024), and the encoder-decoder composition forms an identity map, i.e., $\forall \boldsymbol{z} \in \mathcal{M}, \boldsymbol{z} = \mathcal{E}(\mathcal{D}(\boldsymbol{z}))$.*

Using the vanishing-error autoencoder, we define a manifold-constrained variant of the distribution-dependent equivariance error of Definition 3.1.

**Definition H.2** (Manifold-Constrained Distribution-Dependent Equivariant Functions). *Let $G$ act on $\mathcal{Z}$ via $T_g : \mathcal{Z} \to \mathcal{Z}$ and on $\mathcal{X}$ via $S_g : \mathcal{X} \to \mathcal{X}$. The manifold-constrained equivariance error of the function $f : \mathcal{Z} \to \mathcal{X}$ under the data distribution $p$ is $\sup_g \mathbb{E}_{\boldsymbol{z} \sim p}[\|\boldsymbol{z} - h(S_g^{-1}(f(T_g(\boldsymbol{z}))))\|]$ where $h : \mathcal{X} \to \mathcal{Z}$, and the pair $(f, h)$ forms a vanishing-error autoencoder (Definition H.1).*

This is the equivariance error used by the EquiCon family of algorithms in our experiments. The qualifier *manifold-constrained* reflects that, when $(f, h)$ is a vanishing-error autoencoder, the composition $h \circ S_g^{-1} \circ f \circ T_g$ remains on the manifold, and the loss measures how far this constrained reconstruction lies from the original $\boldsymbol{z}$ under the group action. The vanishing-error property is essential for this interpretation; without it, the composition is not guaranteed to return to the manifold and the loss loses its meaning as a manifold-distance proxy.

**Equivariance.** Let $\boldsymbol{z} \in \mathbb{R}^d$ and $\boldsymbol{x} = f(\boldsymbol{z}) \in \mathbb{R}^d$. For rotation and reflection equivariance, the transformations $T_g$ and $S_g$ can be defined by a rotation matrix $\boldsymbol{R} \in \mathbb{R}^{d \times d}$; then, a function $f$ with the rotation equivariant property would satisfy $\boldsymbol{R}\boldsymbol{x} = f(\boldsymbol{R}\boldsymbol{z})$. For translation equivariance, the transformations would be $T_g(\boldsymbol{z}) = \boldsymbol{z} + g$ and $S_g(\boldsymbol{x}) = \boldsymbol{x} + g$, where $g \in \mathbb{R}^d$. Hence, for a translation equivariance function $f$, we would have $\boldsymbol{x} + g = f(\boldsymbol{z} + g)$. For the case where the output dimension is larger than the input, $f : \mathbb{R}^k \to \mathbb{R}^d$ with $d > k$, translation equivariance can be defined up to a discrete scale, i.e., $T_g(\boldsymbol{z}) = \boldsymbol{z} + g$ and $S_g(\boldsymbol{x}) = T_{sg}(\boldsymbol{z})$ where $s = d/k$. The equivariance properties of translation, rotation, and reflections, combined, are referred to as E(3) symmetries. Without reflections, the symmetries form a Euclidean group SE(3) (Thomas et al., 2018; Fuchs et al., 2020).

E(3), SE(3), and SO(3) are important symmetry groups in 3D Euclidean space, with well-established applications in physics and chemistry, computer vision, and reinforcement learning (Cohen & Welling, 2016; Thomas et al., 2018; Hoogeboom et al., 2022; Xu et al., 2024; Park et al., 2025). Finally, our contributions are complementary to, and can be combined with, the growing literature on meta-learning and automatic symmetry discovery to learn symmetry groups and their actions directly from data (Zhou et al., 2021; Quessard et al., 2020; Dehmamy et al., 2021; Mohapatra et al., 2025).

**Data manifold hypothesis.** Let data $\boldsymbol{x} \in \mathcal{X} \subset \mathbb{R}^d$ be in an ambient space of dimension $d$ with support $\mathcal{X}$ distribution. We assume that data are sampled from a low-dimensional manifold $\mathcal{M}$ (Cayton et al., 2005; Ma & Fu, 2012) embedded in a high-dimensional space (Assumption H.1). This hypothesis is popular in machine learning (Bordt et al., 2023), and has been studied mathematically in the literature (Narayanan & Mitter, 2010; Bortoli, 2022). Moreover, empirical evidence in image processing supports the manifold hypothesis (Weinberger & Saul, 2006; Fefferman et al., 2016), and diffusion-based solvers assume this property (He et al., 2024; Chung et al., 2022b; 2023).

**Assumption H.1** (Manifold Hypothesis). *Let $\boldsymbol{x} \in \mathcal{X} \subset \mathbb{R}^d$ be a data sample. The support $\mathcal{X}$ of the data distribution lies on a $k$ dimensional manifold $\mathcal{M}$ within an ambient space $\mathbb{R}^d$ where $k \ll d$.*

Table 12: DPS superresolution with $\lambda = 0.01$ using different MPE functions.

(a) FFHQ 256.

| MPE function | PSNR | SSIM | LPIPS |
|---|---|---|---|
| None | 26.521 (1.863) | 0.757 (0.046) | 0.133 (0.035) |
| LDM Encoder (FFHQ) | 26.581 (2.457) | 0.773 (0.044) | 0.120 (0.030) |
| CNN Autoencoder (FFHQ) | 26.866 (1.943) | 0.771 (0.044) | 0.116 (0.029) |
| Pretrained ResNet50 | 26.873 (1.941) | 0.771 (0.044) | 0.116 (0.029) |
| Pretrained CLIP | 26.860 (1.942) | 0.771 (0.044) | 0.116 (0.029) |

(b) ImageNet.

| MPE function | PSNR | SSIM | LPIPS |
|---|---|---|---|
| None | 21.573 (4.149) | 0.612 (0.115) | 0.308 (0.107) |
| LDM Encoder (ImageNet) | 22.200 (4.295) | 0.568 (0.146) | 0.384 (0.130) |
| CNN Autoencoder (ImageNet) | 22.178 (4.294) | 0.568 (0.148) | 0.375 (0.125) |
| Pretrained ResNet50 | 22.176 (4.290) | 0.568 (0.148) | 0.375 (0.125) |
| Pretrained CLIP | 22.177 (4.293) | 0.568 (0.148) | 0.376 (0.125) |

## I    Additional figures/tables

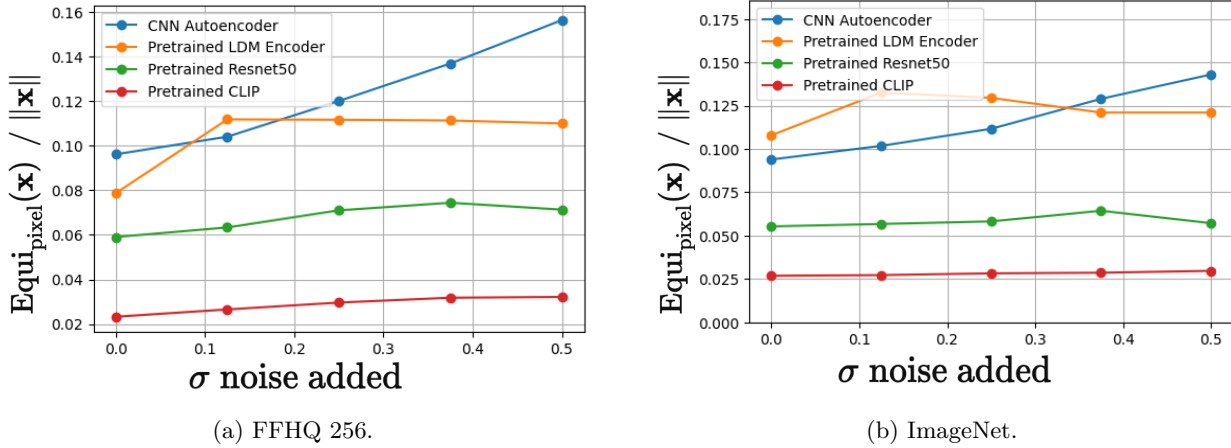

(a) FFHQ 256.

(b) ImageNet.

Figure 16: **Equivariance error vs. $\sigma$ noise added.** As more noise is added, equivariance error, computed with all MPE functions, increases.

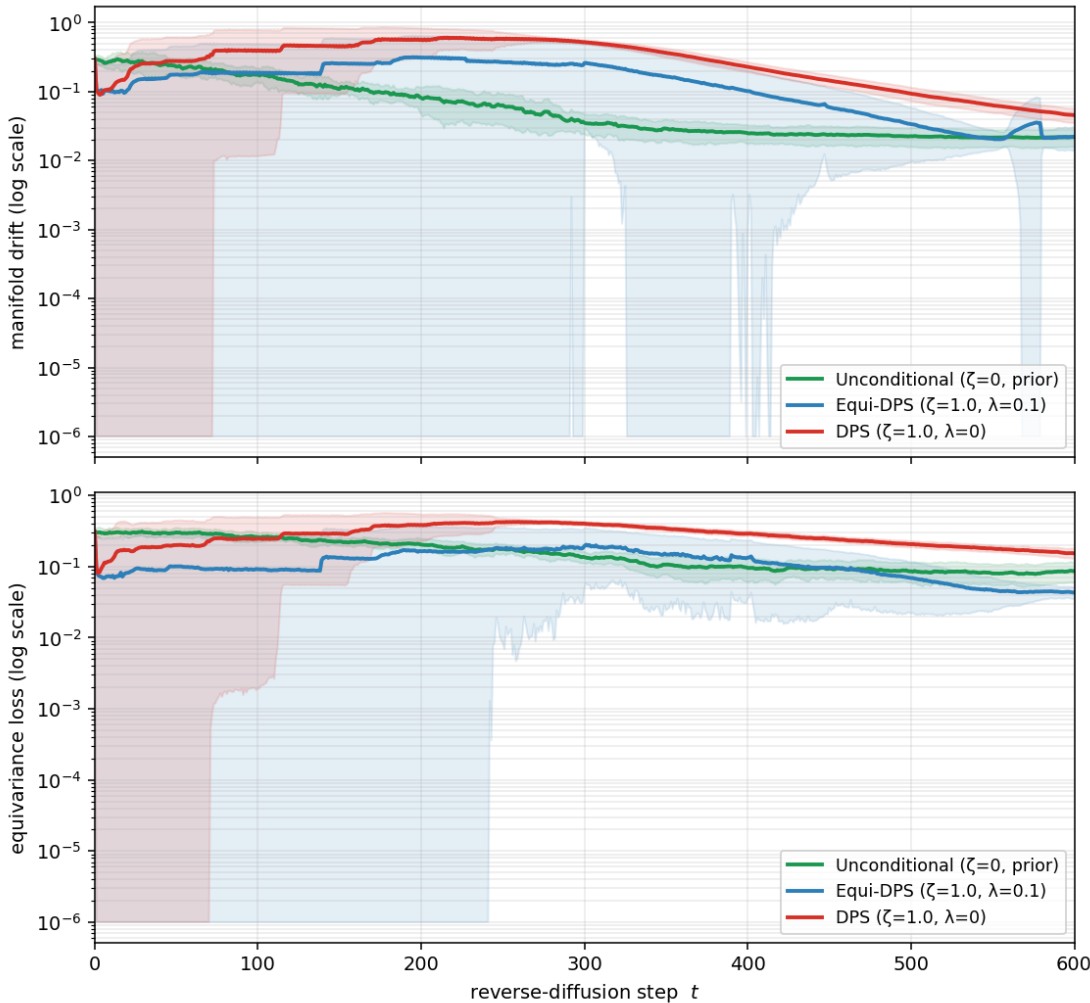

Figure 17: **Assessing manifold distance error and equivariance loss during unconditional and conditional generation.** Analysis are done for first 600 reverse conditional sampling out of 1000 iterations.

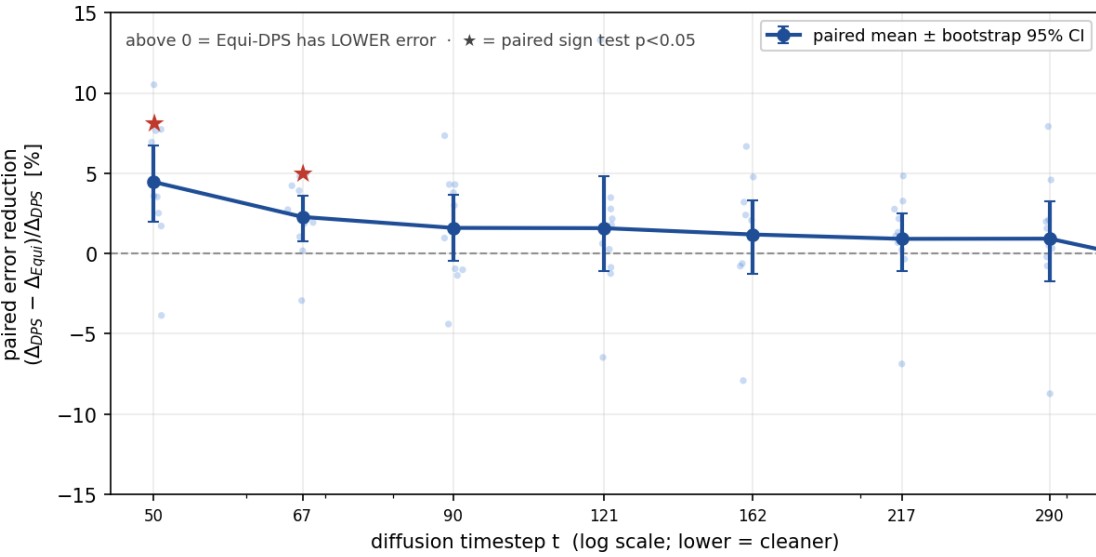

Figure 18: **Assessing likelihood error with and without EquiReg during conditional generation with DPS.**

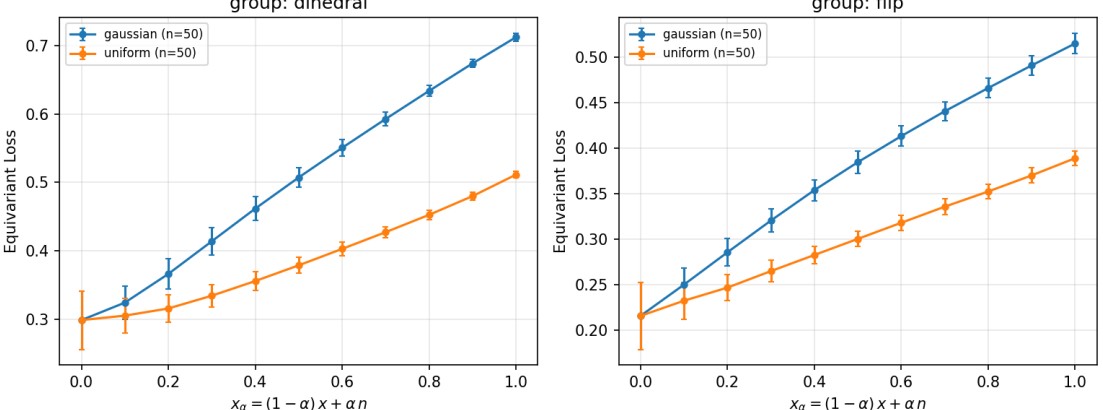

Figure 19: **White-noise experiment**.

Table 13: **EquiReq Characterization under change of EquiReg period, $\lambda_t$, and DDIM steps.**

| | | Super Resolution | | | | Gaussian Blur | | | |
|---|---|---|---|---|---|---|---|---|---|
| **Method** | **Period** | Runtime (s) | PSNR↑ | LPIPS↓ | FID↓ | Runtime (s) | PSNR↑ | LPIPS↓ | FID↓ |
| DPS | N/A | 46.20 | 26.52 (1.86) | 0.133 (0.04) | – | 46.50 | 25.87 (2.06) | 0.125 (0.03) | – |
| Equi-DPS | 1 | 51.10 | 26.73 (1.99) | 0.12 (0.03) | 87.97 | 52.20 | 26.08 (2.25) | 0.12 (0.03) | 87.11 |
| Equi-DPS | 2 | 48.90 | 26.73 (1.99) | 0.12 (0.03) | 87.98 | 49.10 | 26.06 (2.24) | 0.12 (0.03) | 87.19 |
| Equi-DPS | 5 | 47.10 | 26.73 (1.99) | 0.12 (0.03) | 87.98 | 47.30 | 26.06 (2.24) | 0.12 (0.03) | 87.32 |
| Equi-DPS | 10 | 46.90 | 26.73 (1.99) | 0.12 (0.03) | 87.99 | 47.00 | 26.05 (2.24) | 0.12 (0.03) | 87.04 |

(a) Robustness and computational efficiency of applying EquiReg under various periods during sampling. EquiReg maintains performance when applied every $\{1, 2, 5, 10\}$ DDIM steps while incurring minimal computational overhead.

| | PSLD | | |
|---|---|---|---|
| $\lambda_t^{\mathrm{PSLD}}$ | PSNR↑ | SSIM↑ | LPIPS↓ |
| 0.0 | 23.83 (2.61) | 0.63 (0.12) | 0.315 (0.07) |
| 0.01 | 25.35 (2.24) | 0.70 (0.09) | 0.280 (0.07) |
| 0.1 | 26.63 (1.68) | 0.74 (0.08) | 0.337 (0.06) |
| 0.25 | 26.22 (1.57) | 0.72 (0.08) | 0.366 (0.05) |
| 1.0 | 24.74 (1.28) | 0.66 (0.07) | 0.438 (0.05) |

(b) Robustness of EquiReg to the choice of $\lambda_t$. Sensitivity analysis for latent diffusion PSLD.

Figure 20: **Characterization of EquiReg on DPS as a function of regularization parameter $\lambda$ and the measurement step size $\zeta$.** For small values of $\zeta$, the inverse-solver control signal is weak, resulting in only minor deviations from the data manifold. Consequently, off-manifold trajectories are limited and EquiReg provides little benefit. However, in this regime, DPS often exhibits poor measurement consistency due to the weak guidance from the measurement term. As $\zeta$ increases, the measurement control signal becomes stronger, improving measurement consistency but also increasing the tendency of the sampling trajectory to move off the data manifold. In this setting, EquiReg becomes important, regularizing the trajectory toward the manifold while preserving the stronger measurement guidance. Therefore, the benefits of EquiReg become more pronounced as $\zeta$ increases.

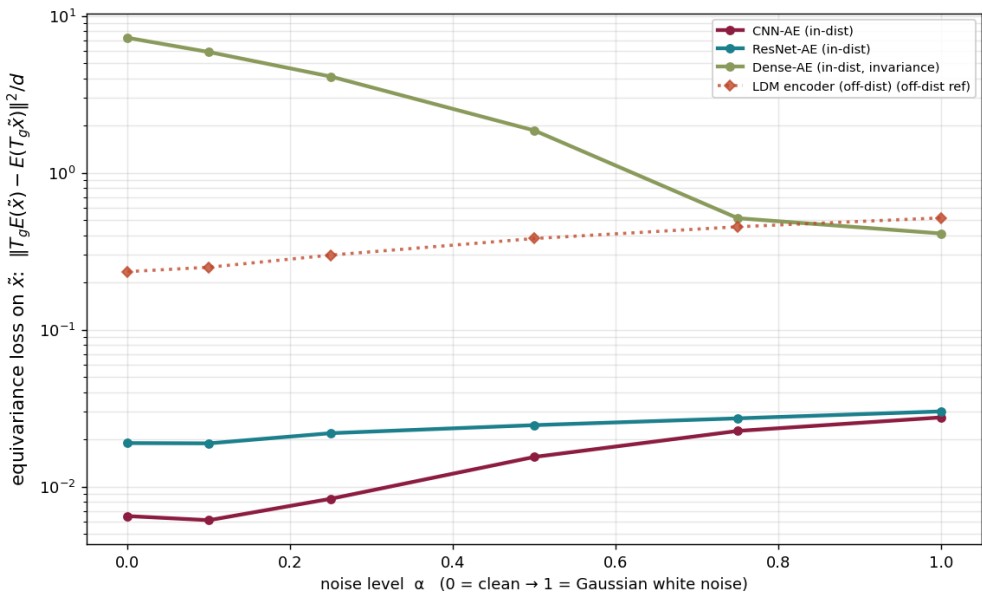

Figure 21: **Role of architecture on MPE strength under same data distribution.** MPE strength can be governed by architectural inductive biases. These analyses are on FFHQ datasets under flip transformation for only characterization. Hence, the resulting models are not used as general purpose pre-trained MPE functions.

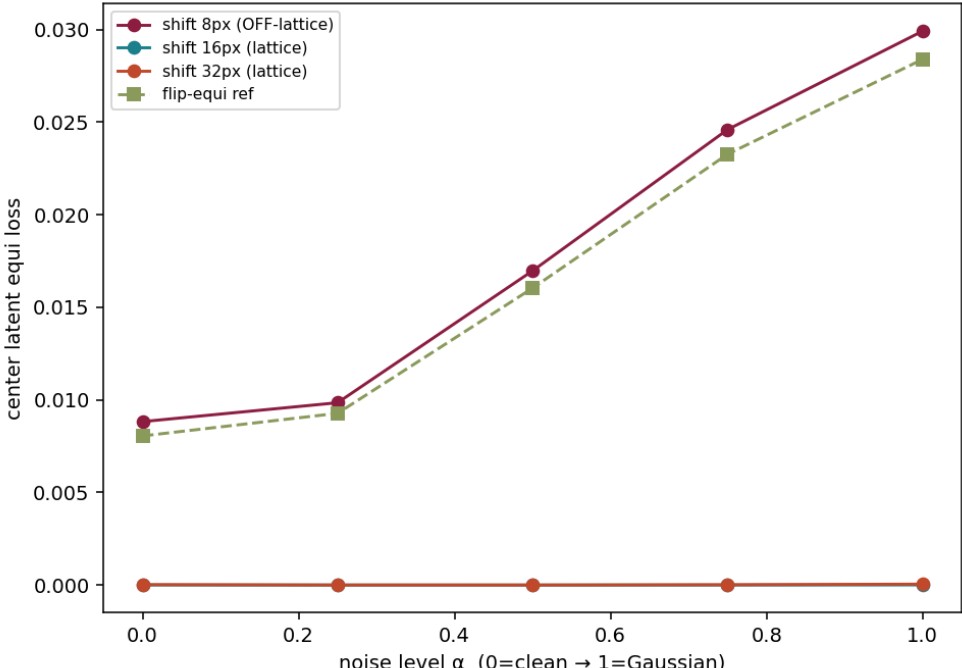

Figure 22: **An equivariant-by-construction function cannot be an MPE function for that group.** A CNN example. The function does not show the MPE property for on-lattice (stride) translation as the group transformation as the function would be equivariant to it globally. On the other hand, the function shows the MPE property for off-lattice translation as the group transformation.

## J  Computing Resources

We conduct experiments on two NVIDIA GeForce RTX 4090 GPUs with 24 GB of VRAM. We note that we use pre-trained models and perform inference, so not much compute is required.

## K  Assets

We use the publicly available code from PSLD (https://github.com/LituRout/PSLD), Re-Sample (https://github.com/soominkwon/resample), DPS (https://github.com/DPS2022/diffusion-posterior-sampling), and SITCOM (https://github.com/sjames40/SITCOM).

## L  Responsible Release

Our approach uses only publicly available datasets and standard pre-trained diffusion models, introducing no novel dual-use or privacy risks. Consequently, no additional safeguards are required.

