# OpenReview forum: "EquiReg: Equivariance Regularized Diffusion for Inverse Problems"
_TMLR — Decision pending for TMLR_

### Review · Reviewer_txsx · 2026-04-05

**Summary Of Contributions:**

The paper proposes a plug-and-play regularization term into the diffusion model to address the Gaussian approximation issue. The proposed EquiReg leverage Manifold-Preferential Equivariant function to penalize invalid trajectories and guide the sampling process back toward realistic restoration. The strengths are listed below.

- The proposed EquiReg is easy to plug-and-play into many existing diffusion solvers without fine-tuning.
- The experiments demonstrate its effectiveness across a wide range of downstream image restoration tasks, such as super-resolution, inpainting, and deblurring.

**Audience:**

Yes

**Audience Explanation:**

Researchers interested in the diffusion model for image restoration tasks may gain some insights from the theoretical inversion process described in this paper.

**Claims And Evidence:**

Yes

**Claims Explanation:**

The proposed method is well-motivated and demonstrated by the theory and experimental results across many downstream tasks.

**Requested Changes:**

- Although EquiReg is a plug-and-play method, the regularization parameter lambda must be tuned for each downstream task, limiting its flexibility.
- The MPE function may introduce additional computational costs, which should be discussed in terms of denoising time compared with the baseline methods.

---

> ### Author Response · Authors · 2026-04-24
>
> We would like to thank the reviewer for their comments.
>
> --> _Reviewer: Although EquiReg is a plug-and-play method, the regularization parameter lambda must be tuned for each downstream task, limiting its flexibility._
>
> We thank the reviewer for raising this point. We acknowledge that EquiReg introduces a regularization term and, like other regularization methods, requires tuning an additional hyperparameter. This inherent property on the flexibility of tuning lambda enables applicability across a wide range of diffusion-based inverse problems, solvers, tasks, and datasets.
>
> We note that our paper already includes ablation studies demonstrating that EquiReg is robust to the choice of $\lambda$ in practice.
>
> - In Table 1b, we present a sensitivity analysis showing stable performance across a wide range of $\lambda$ values (approximately two orders of magnitude across all tested models).
> - Figure 7 in the appendix provides further qualitative evidence: across $\lambda$ values from 0.001 to 0.3, Equi-DreamSampler produces visually consistent, high-quality outputs faithful to the prompt, with degradation only appearing at extreme values (e.g., 0.5). This indicates that fine-grained tuning is not required.
>
> In practice, selecting $\lambda$ only requires a coarse search (e.g., a few values on a log scale) using a small validation set, adding negligible overhead relative to sampling cost. We also note that similar hyperparameter tuning is standard for measurement-consistency terms in diffusion-based methods.
>
> We will clarify these points in the revised manuscript and include brief guidelines for selecting $\lambda$.
>
> --> _Reviewer: The MPE function may introduce additional computational costs, which should be discussed in terms of denoising time compared with the baseline methods._
>
> We thank the reviewer for raising this important point. Our paper already analyzes the computational cost associated with the MPE function.
>
> Our results show that EquiReg introduces negligible denoising-time overhead and, in several settings, significantly reduces total runtime by achieving comparable or superior quality with fewer solver iterations. Specifically:
>
> - Table 1a compares runtime between DPS and Equi-DPS. With a period of 10, Equi-DPS incurs less than 2% overhead on super-resolution (46.90s vs. 46.20s) while improving PSNR from 22.99 to 26.73 and LPIPS from 0.20 to 0.12.
> - Table 1c shows that Equi-DPS with 750 DDIM steps (PSNR 25.60) outperforms DPS with 1000 steps (PSNR 22.99), corresponding to a 25% reduction in sampling steps at higher quality.
> - Table 2 demonstrates a similar effect for SITCOM in motion deblurring: the baseline at $K_{\text{meas}} = 10$ requires 21.57s (PSNR 28.06), while Equi-SITCOM with $K_{\text{meas}} = 5$ and $K_{\text{EquiReg}} = 5$ requires 11.09s (PSNR 29.26), approximately halving runtime while improving quality.
> - Additional results in Table 9 and Figure 12 (appendix) further confirm these trends.
>
> We will improve the organization and clarity of these discussions in the revised manuscript. We are also happy to include additional analysis if the reviewer does not find the above satisfactory.

---

### Review · Reviewer_F8C5 · 2026-05-11

**Summary Of Contributions:**

This paper deals with a framework called Equivariance Regularized (EquiReg) diffusion, that aims for more efficient sampling of the posterior distribution of an imaging inverse problems (based on a diffusion prior).
The authors propose a plug-in modification of existing posterior sampling methods based on diffusion (among which the so-called DPS) that aims at improving sampling efficiency.
This modification, based on a regularization term related to equivariance w.r.t. to a chosen group of transformations $G$, is generic in the sense that it does not depend on the chosen architecture for the score function, neither on the precise numerical scheme used for backward sampling. The regularization aims at penalizing some trajectories of backward diffusion that may drift to off-manifold regions.

Section 1 sets the motivation of the paper, which is to reduce the likelihood error on $p_t(y|x_t)$ that is observed in existing posterior sampling methods like DPS.
Section 2 recalls the construction of backward diffusion processes for posterior sampling, and the equivariant framework, and the link with existing papers on these topics.
Section 3 introduces the proposed modification of DPS, two measures of equivariance errors, the concept of manifold-preferential equivariant (MPE) functions, and the regularization term $R(x_t)$ which depends on the choice of a group of transformations $G$, and a MPE function $f$.
Section 4 reports performance (in terms of PSNR, SSIM, LPIPS, FID) of the equivariant version of several diffusion methods, applied for posterior sampling with different linear and nonlinear inverse problems (deblurring, super-resolution, inpainting, HDR and phase retrieval). This section also contains experiments showing performance of EquiReg when applied to PDE solvers, and also a preliminary text-to-image guided diffusion experiment.


---------------------------------------------------
# STRENGTHS

- This paper is one successful attempt to integrate equivariance (which was shown useful in plug-and-play imaging) into diffusion-based algorithms for inverse problems.

- This can be realized with an apparently simple modification of such diffusion-based algorithms (for example, adding another term in the DPS update).

- The performance gap of the proposed algorithm in a large-scale setting looks quite amazing. The algorithm is tested on a wide variety of large-scale inverse problems, in various conditions.


---------------------------------------------------
# WEAKNESSES

1) In the introduction, the authors emphasize on the fact that previous methods induce a too large error on the approximation of the posterior $p_t(x_0|x_t)$. It seems to be one main goal of the proposed EquiReg framework to reduce this approximation error. However, for now, the paper does not contain clear numerical evidence that EquiReg does reduce the error. Since the authors seem to advertise this error reduction, there should be clean numerical evidence for it.

2) I am really not convinced by the fact that imposing equivariance would help to get closer to the data manifold. For example, the distribution of a white noise is invariant to many groups of transformations (reflections, rotations, and even any group of pixel permutation).
In the same vein, I do not grasp the intuition underlying the heuristical explanation of Fig. 1. The $M_t$ are not defined in the core of the paper, and I'm really not conviced that the EquiLoss red levels are related to the distance to the manifold $M$, as suggested by the right part of the figure. I think this is a very over-simplistic (and perhaps misleading) diagram.
The right diagram of Fig. 3 seems also a bit strange: the pink dotted segment has no precise meaning and the point $x_t$ is not represented; normally the $x_t$ should be represented on the $x$-axis and not as points on the curves.

3) The discussion is often based on imprecise concepts. In particular, the authors often speak of "effective regularizer", "practical functions" or "practical neural networks". These expressions appear as completely unclear. In a similar manner, the expression "diversity increases approximately linearly"' has no precise meaning (at least at the first time it is used in the introduction, in a broad sense). Another example of imprecise sentence: "EquiReg leverages this intrinsic property as a principled regularization signal.''

4) The proposed method involves a new parameter (actually a new sequence of parameters) $t \mapsto \lambda_t$ whose choice is virtually not discussed. The authors claimed some robustness with respect to this parameter $\lambda_t$ (Table 1c, Fig. 8 to 11). But in my opinion, these experiments are clearly unsufficient to discuss this parameter. It is obvious that this parameter may impact both the stability of the algorithm and its performance. Besides, this parameter is actually a whole sequence of values $t \mapsto \lambda_t$. If I am not mistaken, the authors did not give the reason of setting it to a constant value. Actually, the reader should go to Appendix E to understand that $\lambda_t$ is kept constant except in the 10 \% last iterations where it is set back to $0$. Also, on this point, I would like to mention that the original DPS method also relies on a sequence $\zeta_t$ which could also be tuned. In the current submission, it is unclear whether the performance gap observed between DPS and Equi-DPS could be reduced by tuning also $\zeta_t$.

5) Sec 3 does not clarify the usage of Def 3.2 compared to Def 3.1. It seems that the proposed regularization resembles Def 3.1, but is not related to Def 3.2 since it does not include $h$.
More generally speaking, the concepts introduced in Def 2.1 (equivariance), Def 2.2 (approximate equivariant functions), Def 3.1 (Distribution-dependent equivariant functions), Def 3.2 (Manifold-constrained Distribution-dependent equivariant functions) do not contribute to the theoretical findings of the paper. This presentation is perhaps chosen to give an impression of mathematical rigour to the reader. However, none of this definition is actually used to build some precise mathematical reasoning (which could lead to clearly stated propositions or theorems).
Notice also that Def 3.2 relies on the concept of ``vanishing-error autoencoder'', which the reader can find only in Appendix I. Besides, the authors do not explain why Def 3.2 should be restricted to this kind of autoencoder.

6) Finally, I am not sure to understand what are precisely the MPE functions that are introduced, and how they are computed in practice. It is unclear if using MPE functions should change the learning of the whole diffusion model or if it should just change the model at inference for inverse problems.

7) The theoretical contributions showcased in Appendix G appears to me as disconnected from the main discussion of the paper. Besides, I highly doubt the correctness of the proofs provided in this appendix. It quickly goes to an assumption "Without loss of generality, we assume $\phi_C(x,t) < 0$ for all $x$ and $t$.'' which does not seem justified. Besides, I do not really see the point in providing a so-called Proposition G.2  while exhibiting as "informal'' with a detailed (and long) proof that relies on very approximate statements (following equation (34)).

8) Many parts of the paper are vague or unclear. Here is a list of imprecise parts:

- "In the Bayesian formulation, the solution maximizes the posterior distribution (...)''. The solution to what? The maximum \emph{a posteriori} is only one possible way of solving the problem, but there are many others. And in fact, the method proposed in this paper does not target this MAP solution.

- "how can we ensure the \underline{reliability and practicality} of conditional diffusion models''

- "MPE functions can emerge in different ways''

- "healthy posterior sampling behavior''

- "favorable fidelity-diversity trade-offs''

- "unreliable estimates''

- In the end of Def 2.2, $< \epsilon$ should be replaced by $\leq \epsilon$.

- In Equation (5) defining the regularization term, it is not explained how the transformation $g$ should be chosen. I would like to mention that this definition is the cornerstone of the proposed algorithm modification. The fact that there is still some imprecise detail about it represents a major issue of the submitted paper. On a related point, I find it rather surprising that there is a minus sign in front of $R(x_t)$ in equation (4).

- Diagrams of Fig 4 cannot be understood because of incomplete caption. Table 6 has no complete caption either.

- Fig. 5 cannot be fully understood either. Caption for Fig. 5a is incomplete. The caption for Fig. 5b does not indicate which inverse problem is considered. And the graphs of Fig. 5c could be much better visualized by comparing continuous curves (e.g. with kernel density estimators) rather than overimposing the histograms.

- EquiCon is not advertised in many figures and results tables, but it is not defined in the core of the paper.

- The "period'' parameter is not defined anywhere in the paper. One cannot infer its precise meaning.

**Additional Comments:**

- In Def 3.2, Why is this new error measure is called "Manifold-Constrained''? Besides, what is the use of Def 3.1?


- I am not convinved by the preliminary text-to-image experiment, because in my opinion it is not sufficient to indicate some benefit for such application. I think the paper should keep the focus on inverse problems.

- Since the expression "plug-and-play'' now refers to a precise imaging framework, I would recommend the authors not using the expression "plug-and-play regularizer'', which could misleadingly refer to the regularizing functions that are used in several plug-and-play algorithms.

- The practical choice of the MPE function used in the experiment is not clear:
"We adopt the pre-trained encoder-decoder $E-D$ as our MPE function''. Is it the encoder or the decoder that plays the role of $f_{MPE}$? And this description is not sufficient to reproduce the results. Were the used autoencoder pairs already delivered in some previous publications?

- The discussion of Appendix D about diversity seems quite weak and vague to me. For a given instance of inverse problem, the posterior distribution is fixed. Beyond diversity, the main thing that should be checked is that the considered algorithm indeed samples properly this posterior distribution. The fact that diversity is high is not necessarily a good thing (for example if the obtained variance exceeds the variance of the posterior distribution).

- In Appendix E, the description of the operator used is not satisfactory.
The description for deblurring seems wrong:
"For Gaussian and motion deblur, we use kernels of size 61 × 61, with standard deviations of 3.0 and 0.5, respectively.''
The authors should have compared several kernel widths for Gaussian deblurring, and various motion kernels for motion deblurring (in this case, the authors should provide the list of used kernels, which are not only described by the standard deviation).

- From the description of Appendix F, I do not get the inverse problem considered related to Helmoltz equation or Navier-Stoke equation. Is it about the inversion of $a \mapsto u$? or $(a,f) \mapsto u$?
Why is there a Hessian $\nabla^2 u$ in equation (6)?

**Audience:**

Yes

**Audience Explanation:**

Equivariant imaging systems is a currently active topic.

**Broader Impact Concerns:**

No concern.

**Claims And Evidence:**

No

**Claims Explanation:**

See points 2, 5, 7, 8 above.

**Requested Changes:**

- Address all priority objections raised above.

- Reduce the parts of the paper where the authors aggregate some (sometimes redundant) heuristical and shaky comments. Replace that by a thorough and precise discussion on the equivariance errors and a precise analysis of the behavior of the algorithm in a smaller-scale setting.

- Study the integration of the proposed equivariant regularization to smaller-scale image reconstruction problem (where other lighter algorithms can be used).

- In particular, on such "easier'' problems, the authors should compare to previous works related to equivariant plug-and-play, see for example (Terris et al. 2024).

- Propose some clear numerical evidence that the proposed method indeed decrease the likelihood error observed in previous schemes.

---

> ### Author Response · Authors · 2026-06-17
>
> We thank the reviewer for their thoughtful comments. Below, we provide a response. The manuscript has accordingly been revised (will be uploaded soon) to address the concerns (please see the changes in blue).
>
> __Numerical analysis on how EquiReg reduces distance to manifold or likelihood approximation error__
>
> We have revised the manuscript to cite papers that discuss the intractability of the likelihood where point estimates of p(x0|xt) are used. We have also conducted two additional analyses to study the sampling trajectory during unconditional and conditional generation.
>
> _NEW Figure 20 – Quantifying distance to the manifold using a pre-trained AE._ We use the encoder-decoder architecture from latent diffusion models as a proxy for a representation that has learned the low-dimensional structure of the data manifold in its latent space (see [1,2,3]). We assume the autoencoder is a vanishing-error autoencoder such that the image of the AE lies on the manifold on which it was trained ($M_{\text{trained}}$). We define the manifold-distance proxy $e(x) = ‖x − D(E(x))‖$.
> For samples on the trained manifold, this quantity is zero. Since the AE is not trained exactly on the FFHQ distribution considered in our experiments, the reconstruction error is not expected to vanish on FFHQ images. Nevertheless, it provides a useful proxy for the distance to the natural-image manifold, particularly when samples are far from the data manifold (e.g., early in the reverse diffusion process, around timestep 600 of 1000).
>
> Using this metric, we quantify both the manifold-distance proxy and the EquiLoss along the trajectory of $x_{0|t}$. Results show that unconditional generation has the lowest manifold distance and that this distance decreases throughout the reverse diffusion process. Interpreting unconditional generation as a denoising process in which the sample gradually approaches the data manifold, we observe that the EquiLoss also decreases continuously along the trajectory.
>
> For conditional generation with DPS, we observe an increase in manifold distance relative to unconditional generation, which can be associated with the measurement-guidance term. This manifold distance begins to decrease after approximately the first 30% of the reverse trajectory. Prior to this stage, $x_{0|t}$ is reconstructed from highly noisy states, where the isotropic-Gaussian approximation is expected to be weakest. We observe a strong correlation between manifold distance and EquiLoss throughout the trajectory, supporting the central hypothesis of the paper that regularization using EquiLoss can steer samples closer to the data manifold.
> Importantly, when DPS is regularized with EquiReg, we observe both a lower EquiLoss and a lower manifold distance throughout the posterior-sampling trajectory compared to DPS alone.
> These results demonstrate that: a) EquiReg reduces the distance to the manifold (using the latent-diffusion AE as a proxy), and b) EquiLoss decreases throughout the trajectory during unconditional generation.
> Given these observations regarding the first half of the sampling process, we next study how EquiReg affects the likelihood approximation error in the second half of the trajectory.
>
> _NEW Figure 21 – Quantifying likelihood approximation error during posterior sampling._ To directly assess the approximation underlying conditional generation methods such as DPS, we estimate the true measurement likelihood $p(y \mid x_t)$ and compare it with the point-estimate likelihood used in DPS and EquiReg-DPS throughout the reverse diffusion trajectory.
> For a fixed test image and measurement operator, we first run DPS and EquiReg-DPS using the same random seed. Then, given the trajectories $x_t$, for selected timesteps $t_i$, we freeze the state $x_{t_i}$ and generate multiple unconditional reverse-diffusion completions from $x_{t_i}$ down to t=0. This yields samples from the posterior $p_{t_i}(x_0 \mid x_{t_i})$.
> Using these samples, we estimate the true likelihood via Monte Carlo:
>
> $\hat p_{t_i}(y\mid x_{t_i})  = \frac{1}{M}\sum_{m=1}^{M} p(y\mid x_0^{(m)}), $
>
> and compare it with the approximation employed by DPS,
>
> $\tilde p_{t_i}(y\mid x_{t_i})  = p(y\mid x_{0|t_i}),$
>
> where $x_{0|t_i}$ is the Tweedie posterior mean obtained from a single denoiser evaluation.
> The figure quantifies error
>
> delta $=  | \log \hat p_{t_i}(y\mid x_{t_i})  - \log \tilde p_{t_i}(y\mid x_{t_i}) |$,
>
> and reports the relative error for DPS and EquiReg-DPS. Positive values indicate that EquiReg achieves lower likelihood-approximation error than DPS. We observe that the likelihood-approximation error is lower for EquiReg-DPS than for DPS. In particular, toward the end of the reverse process (smaller timestep values), the improvement becomes statistically significant.

---

> > ### Author Response · Authors · 2026-06-17
> >
> > __White-noise discussion__
> >
> > We thank the reviewer for the careful reading. The white-noise example highlights an important technical distinction.
> > The equivariance error in Definition 3.1 is a pointwise property of a learned function f evaluated at a sample z, rather than a statement about the distribution of z. The reviewer’s example correctly notes that i.i.d. Gaussian noise is invariant in distribution under arbitrary pixel permutations. However, this does not imply that a network f trained on natural images is pointwise equivariant on individual noise samples.
> >
> > In practice, different noise realizations produce substantially different outputs of f, and therefore the pointwise equivariance error of a learned function remains a useful indicator of whether a sample lies on or off the data manifold, even when the on- and off-manifold distributions share the same underlying group symmetries.
> >
> > The revised manuscript adds a short paragraph immediately following the definitions in Section 3 to make this distinction explicit.
> >
> > _Figure 1._ We have updated the caption to: a) emphasize that Figure 1 is a schematic used to motivate the method; b) clarify that the EquiLoss color gradient is illustrative rather than a direct measurement of manifold distance (while pointing the reader to Figure 4a for empirical measurements of equivariance error); and (c) define $M_t$ as the noisy data manifold at diffusion time t, with $M_0=M$ recovering the clean data manifold. We note that the notation M_t is commonly used in prior diffusion-based inverse-problem literature [1,4].
> >
> > _Figure 3 (right panel)._ We have updated the caption to clarify the geometric interpretation. The right panel is a schematic one-dimensional illustration of $p_t(x_0 \mid x_t)$ for a fixed $x_t$, where the curve denotes the bimodal posterior density. The Tweedie posterior mean $x_{0|t}$ is a point on the$x_0$-axis, and the pink dotted segment connects the two modes of the posterior to emphasize that $x_{0|t}$, being their average, lies between them in a low-density region. This is precisely the off-manifold posterior expectation that the figure is intended to illustrate.
> >
> > Additionally, we have conducted a new analysis (_NEW Figure 22_) of the equivariance loss of the considered functions. Specifically, we use the LDM encoder as the function and flip and dihedral transformations as the group actions. We evaluate the equivariance loss when the input follows a white-noise distribution or is progressively corrupted by white noise.
> > We construct samples as convex combinations of clean images and noise, $(1-a)x + a n$, and consider both Gaussian and white noise. Results show that the equivariance error increases monotonically as “a” increases. This demonstrates that MPE functions exhibit higher equivariance error on white-noise inputs and lower error on clean natural images, providing a useful signal for guiding samples toward the data manifold.
> >
> > [1] He, Y., et al. (2024). Manifold preserving guided diffusion. ICLR.
> >
> > [2] Shao, H., et a. (2018). The riemannian geometry of deep generative models. IEEE CVPR Workshops.
> >
> > [3] Anders, C., et al. (2020). Fairwashing explanations with off-manifold detergent. ICML.
> >
> > [4] Chung, H., Sim, B., Ryu, D., & Ye, J. C. (2022). Improving diffusion models for inverse problems using manifold constraints. NeurIPS.

---

> > > ### Author Response · Authors · 2026-06-17
> > >
> > > __Section 3 Definitions__
> > >
> > > We thank the reviewer for the suggestion.
> > >
> > > We would like to clarify that the paper makes two complementary contributions: a) the introduction of distribution-dependent equivariant functions and MPE functions, and b) the regularization of posterior sampling in diffusion models.
> > >
> > > We now clarify in the manuscript that Definitions 3.1 and the MPE framework provide one successful example of a regularizer that can be used for regularized posterior sampling. Our contribution to diffusion regularization formalizes what it means to regularize a sampling process and is independent of the particular form of the regularizer. The regularizer could be constructed using equivariance, as in this work, or through other principles. From the equivariance perspective, our contribution is to show that MPE functions and Definition 3.1 provide an effective way of constructing regularizers that assign high values to undesirable samples and low values to desirable samples.
> > >
> > > Regarding the definitions themselves, the paper introduces a new notion of _Distribution-Dependent Equivariant Functions_. This definition builds directly on Definitions 2.1 and 2.2. Therefore, we believe it is necessary to include these foundational definitions in the preliminaries so that the reader can properly interpret the new concept.
> > >
> > > Definition 3.1 provides the formal framework required to discuss MPE functions. Definition 3.2 extends Definition 3.1 and is used specifically by our EquiCon family of algorithms.
> > >
> > > If the reviewer believes it would improve clarity, we would be happy to move the definition of the Vanishing-Error Autoencoder into the main paper. We currently place it in the appendix due to space constraints. The notion of a vanishing-error autoencoder has previously been introduced and used in several works (Shao et al., 2018; Anders et al., 2020; He et al., 2024). Informally, it describes an autoencoder that can perfectly represent a data manifold, yielding exact reconstruction for any sample on that manifold. This concept has also been used in diffusion literature to characterize projections of sampling trajectories onto data manifolds.
> > >
> > > Consequently, the term manifold-constrained reflects the fact that when $(f,h)$ forms a vanishing-error autoencoder, the composition $h \circ S_g^{-1} \circ f \circ T_g$ remains on the manifold. The vanishing-error property is what makes the resulting loss interpretable as a proxy for manifold distance.
> > >
> > > In the revised manuscript, we have further clarified the relationship between the definitions and the operational loss used by the algorithm:
> > >
> > > - Definition 3.2 (Manifold-Constrained Distribution-Dependent Equivariant Functions) has been moved out of the main text and relocated to Appendix H (Additional Background), placed alongside the Vanishing-Error Autoencoder definition that it relies on. The main text now refers to it once as the equivariance error used by the EquiCon family of algorithms.
> > >
> > > - Equation (5) is now the operational equivariance loss used by the algorithm. The intro sentence to Equation (5) explicitly cites Definition 3.1 as its conceptual basis, so the role of Definition 3.1 in the algorithm is visible at the point where the loss is first written down.
> > >
> > > We hope these revisions make the relationship between the definitions, MPE functions, and the operational loss used by EquiReg substantially clearer.

---

> > > > ### Author Response · Authors · 2026-06-17
> > > >
> > > > __Operational form of the MPE function__
> > > >
> > > > We thank the reviewer for pointing this out. In the revised manuscript, Section 4 now specifies the operational form of the MPE function explicitly. For latent-space solvers (Equi-PSLD, Equi-ReSample), f_MPE is the decoder D applied to $z_{0|t}$, so the equivariance loss in Equation (5) is $ ||S_g(D(z_{0|t})) − D(T_g(z_{0|t}))||^2$. For pixel-space solvers (Equi-DPS, Equi-SITCOM), f_MPE is the encoder E from the same autoencoder applied to $x_{0|t}$, giving $||S_g(E(x_{0|t})) − E(T_g(x_{0|t}))||^2$. The EquiCon variants instead use the manifold-constrained equivariance error from Appendix H, which involves both E and D.
> > > >
> > > > Regarding the source of the autoencoder, (E, D) is the publicly-released LDM autoencoder from Rombach et al. 2022 (LDM-VQ-4 for FFHQ, Stable Diffusion v1.5 for ImageNet). This is the same autoencoder that the latent-space solvers (PSLD, ReSample) already use internally for their own sampling, so no additional pre-training is required. The same checkpoint provides E for the pixel-space solvers (DPS, SITCOM); the underlying pixel-space diffusion model itself is unchanged. We have added these details to Section 4 of the revised manuscript.
> > > >
> > > > __Clarifications on EquiCon, and model choices__
> > > >
> > > > We thank the reviewer for their comment on model parameters. Section 3 of the revised manuscript now defines each of these explicitly.
> > > >
> > > > _Equi-X and EquiCon-X._ We refer to the resulting algorithms as Equi-X, where X is the underlying solver (DPS, PSLD, ReSample, SITCOM). EquiCon-X denotes the manifold-constrained variant that uses the equivariance error from Appendix H instead of Equation (5).
> > > >
> > > > _Period P._ The EquiReg gradient update can be applied every P DDIM steps (period P), with $P = 1$ in our default setting. Table 1a characterizes performance as a function of P.
> > > >
> > > > _Symmetry group and the actions $T_g$, $S_g$._ $T_g$ acts on the input and $S_g$ on the output of $f_{\text{MPE}}$, determined by a chosen symmetry group G. Section 3 now describes near Equation (5) the principle for choosing G, which is to match the symmetries that the pre-trained $f_{\text{MPE}}$ approximately respects. For example, an autoencoder trained with flip augmentation suggests a flip group, and a neural operator trained on rotation-symmetric data suggests discrete rotations. The specific groups used in our experiments (vertical reflection for FFHQ; rotation group $G = {0, \pi/2, \pi, 3\pi/2}$ for ImageNet, uniformly sampled at each step) are described in Section 4. Additional guidelines and references for automatic symmetry discovery are in Appendix H.
> > > >
> > > > __Text-to-image experiment__
> > > >
> > > > The text-to-image experiments are intended as an illustrative extension of EquiReg's applicability, not as a primary contribution. To resolve concern or confusion on the main message of the paper, we have moved the text-to-image discussion out of Section 4 and into Appendix A; Section 4 now contains only a brief two-sentence mention of this. This makes clear that the substance of the paper lies in the inverse-problem results, with the text-to-image content serving as a supplementary demonstration of EquiReg's broader applicability.
> > > >
> > > > __Diversity analysis__
> > > >
> > > > The reviewer is correct that high diversity is not automatically a good thing. We clarify that we are not using diversity as a proxy for performance.
> > > >
> > > > The intra-LPIPS and pixel-std metrics serve as diagnostics for posterior collapse, a known failure mode of regularizers such as TV that can shrink the set of plausible reconstructions. This is the standard practice for evaluating diversity in diffusion-based inverse problem solvers, where closed-form posteriors are unavailable and prior work similarly uses sample-variation metrics (intra-LPIPS, pixel-std) on multiple reconstructions of the same measurement.
> > > >
> > > > The comparison against DPS without regularization on the same task controls for this. Finding that Equi-DPS achieves similar or higher diversity than DPS while improving fidelity therefore indicates that the regularizer does not artificially shrink posterior coverage.
> > > >
> > > > To clarify above, we have added a paragraph at the start of Appendix D making this diagnostic role explicit, and noted that a rigorous calibration analysis against a known posterior (for example, on a toy problem with closed-form posterior) is left to future work.

---

> > > > > ### Author Response · Authors · 2026-06-17
> > > > >
> > > > > __Language and statements__
> > > > >
> > > > > We thank the reviewer for their comment on the language. We have revised the manuscript to increase clarity and precision of the language.
> > > > >
> > > > > "effective regularizer" becomes "a regularizer for diffusion posterior sampling."
> > > > >
> > > > > "practical functions" becomes "trained neural networks."
> > > > >
> > > > > "practical neural networks" becomes "widely-used pre-trained networks."
> > > > >
> > > > > "favorable fidelity-diversity trade-offs" becomes "improved fidelity without posterior collapse."
> > > > >
> > > > > "healthy posterior sampling behavior" becomes "consistent, non-collapsed posterior sampling behavior."
> > > > >
> > > > > "leverages this intrinsic property as a principled regularization signal" becomes "uses this property as the basis for the regularization loss."
> > > > >
> > > > > "how can we ensure the reliability and practicality of conditional diffusion models under this approximation?" becomes "how can we ensure that conditional diffusion samples remain on the data manifold under this approximation?"
> > > > >
> > > > > "MPE functions can emerge in different ways" becomes "MPE functions can emerge from training with symmetry-preserving augmentation and from inherent data symmetries."
> > > > >
> > > > > "In the Bayesian formulation, the solution maximizes the posterior distribution" becomes "In the Bayesian formulation, inference targets the posterior distribution."
> > > > >
> > > > > "diversity increases approximately linearly" becomes "our diversity metrics (intra-LPIPS and pixel-std) grow linearly with task difficulty," with an explicit reference to the corresponding figure for the empirical trend.
> > > > >
> > > > > "unreliable estimates" no longer appears, as the surrounding text in the previous Appendix G has been removed in response to the reviewer's comment about that appendix.
> > > > >
> > > > > In response to the reviewer's broader concern about language, we additionally revised the manuscript for similar phrasings tied to claims about EquiReg. The most notable changes include the following.
> > > > >
> > > > > "particularly effective under reduced sampling" becomes "largest gains are under reduced sampling."
> > > > >
> > > > > "particularly useful when applied to LDMs" becomes "the improvement is largest when applied to LDMs."
> > > > >
> > > > > "widely robust to the choice of MPE" becomes "performance is stable under different choices of MPE."
> > > > >
> > > > > "how to design the regularizer to be effective" becomes "how to design the regularizer to improve posterior sampling."
> > > > >
> > > > > "can be effectively integrated into the EquiReg framework" becomes "can be integrated into the EquiReg framework."
> > > > >
> > > > > "EquiReg enforces an effective denoising" becomes "EquiReg enforces denoising."
> > > > >
> > > > > "MPE functions are widespread and effective for EquiReg" becomes "MPE behaviour emerges across many neural networks and improves EquiReg's performance."
> > > > >
> > > > > "naturally expands sampling" becomes "expands sampling."
> > > > >
> > > > > "remains robust under reduced sampling, effectively accelerating convergence" becomes "maintains performance under reduced sampling, accelerating convergence."

---

> > > > > > ### Author Response · Authors · 2026-06-17
> > > > > >
> > > > > > __Additional revisions to increase clarity__ (all changes are highlighted in color blue in the manuscript).
> > > > > >
> > > > > > Figure 4 caption. We have expanded the caption to describe the y-axis convention (per-sample equivariance loss averaged over images and group actions) and to list, for each subfigure, the perturbation types compared.
> > > > > >
> > > > > > Figure 5 caption. We have expanded the caption to specify (a) the dataset (FFHQ) and corruption (added Gaussian noise) used for the equivariance-error curve, (b) the inverse problem (super-resolution 4×) underlying the SSIM-vs-noise plot, and (c) that the panel shows overlaid distributions of per-image PSNR for DPS vs Equi-DPS with dashed lines marking the means.
> > > > > >
> > > > > > Table 6 caption. We have expanded the caption to state the metric (relative ℓ_2 error in %, lower is better), that results are averaged over 100 test samples, and that bold marks the best result in each column.
> > > > > >
> > > > > > Appendix E motion-blur description. We clarify the convention used in our experiments. The motion-deblur kernel is generated by the procedure used in DPS (Chung et al. 2023) and adopted by subsequent benchmarks (PSLD, ReSample, SITCOM, DiffStateGrad), in which σ controls the intensity of a randomly sampled motion trajectory rather than the width of a Gaussian. We use a single Gaussian-kernel width and a single motion-kernel configuration in line with these benchmarks for direct comparability with prior work.
> > > > > >
> > > > > > Appendix F PDE equations. We have added an explicit note that $\nabla^2$ in the Helmholtz equation denotes the Laplacian operator, and added one sentence each for the Helmholtz and Navier-Stokes problems explicitly stating the forward and inverse problem definitions in terms of which quantity is observed and which is predicted. On the reviewer's question about whether the inversion is of the measurement operator A or the PDE operator G, the inverse problem in our setting is to recover the input field a (coefficient field for Helmholtz, initial condition for Navier-Stokes) from sparse observations of the solution u. It is therefore the joint inverse problem A ∘ G (the composition of the PDE operator G that maps $a \rightarrow u$ and the sparse-measurement operator A that subsamples u), recovered using a diffusion prior over a.
> > > > > >
> > > > > > Plug-and-play terminology. We have replaced "plug-and-play" with "plug-in" wherever it described EquiReg in the manuscript (Abstract, Section 2, Section 4, Appendix F, and Conclusion), to avoid collision with the established Plug-and-Play imaging framework.
> > > > > >
> > > > > > Minus sign in Equation (4). We clarify the sign convention. The minus sign in front of R reflects that the reverse-time SDE flows toward higher log-density regions, so subtracting $\nabla R(x_t)$ pushes trajectories toward lower regularizer values, i.e., toward the manifold.

---

> ### Author Response · Authors · 2026-06-17
>
> __Theory in Appendix G__
>
> We thank the reviewer for raising this concern. We have revised Appendix G in three ways. First, we tied the theoretical contribution to the main paper through cross-references in Section 3 and the appendix outline. Second, we explicitly justified the WLOG step the reviewer flagged. Third, we explicitly framed the $e^{\delta} \approx 1 + \delta$ step as a first-order Taylor expansion valid in the small-R regime that EquiReg targets. We find Proposition G.2 useful as it provides the conceptual bridge from the Wasserstein-flow perspective to EquiReg's practical regularization step. The changes are summarized below.
>
> _On the connection to the main paper._ Section 3 now contains a paragraph framing the Wasserstein-flow perspective explicitly. It states that Proposition G.1 identifies the functional whose minimization the ideal reverse-conditional dynamics correspond to, and that EquiReg approximates the Wasserstein gradient flow of a reweighted version of this functional, formalized as Proposition G.2. Appendix G itself now opens with a Summary subsection that previews both propositions and the limits of the analysis. The Summary closes with a "Connection to EquiReg" paragraph that identifies the abstract regularizer $R(x)$ in Proposition G.2 with the manifold-preferential equivariance loss used by the algorithm, making the link between the theory and the practical method explicit.
>
> _On the WLOG assumption._ The assumption $\phi_{C}(x,t) < 0$ for all $x$ and t follows from a constant shift $\phi_{C}(x,t) := \phi(x,t) − C(t)$. The Wasserstein gradient flow associated with $\phi$ depends only on derivatives of $\phi$ with respect to $x$, so subtracting a function $C(t)$ that depends only on t leaves the dynamics unchanged. Choosing $C(t) > sup_{x} \phi(x,t)$ on the support of ρ(·,t) makes $φ_{C}(·,t)  < 0$, provided $\phi(·,t)$ is bounded above on this support. We now state this construction and the boundedness assumption explicitly in the Summary subsection of Appendix G (next to the WLOG sentence the reviewer flagged) and again in the detailed proof of G.2.
>
> _On the first-order Taylor step._ The approximation $e^{\delta} ≈ 1 + \delta$ is a first-order Taylor expansion in $\delta (x,t) := R(x)/ \hat \phi_{C}(x,t)$, accurate in the regime where $|\delta|$ is small. This is the regime EquiReg operates in. By construction, $R(x)$ is small for $x$ near the data manifold, and the practical regularized dynamics drive trajectories toward this region, so the typical $\delta$ encountered during sampling stays small. We have added this clarification explicitly in the detailed proof. We also point the reviewer to the existing Remark following Proposition G.2's proof, which makes the analogous statement at the level of functionals. Since $e^A ≥ 1 + A$ for all real A (and since $\phi < 0$), the linearized functional is an upper bound of the true reweighted $\tilde \phi$ . The dynamics derived from the linearization therefore evolve to minimize an explicit upper bound of $\tilde \phi$, which is common in probabilistic modeling, e.g. ELBO in the replacement of the log-likelihood.
>
> _On Proposition G.2._ We find Proposition G.2 useful as it provides the formal connection between the Wasserstein-flow perspective and EquiReg's practical regularization step. We note that the regularization framework is general and not tied to the particular equivariance-based loss we define. The proposition aims to explain what it means to regularize the reverse path of a diffusion model. Without G.2, Appendix G would present only the ideal reverse dynamics (Proposition G.1) without a stated relationship to the algorithm in the main paper. With G.2 in place, the appendix offers a precise statement of how EquiReg arises as a first-order approximation of a reweighted Wasserstein gradient flow that biases the dynamics toward on-manifold regions. We have also retitled Appendix G as "A Wasserstein-flow perspective on posterior sampling" to describe the appendix as motivation grounded in a first-order analysis.
>
> __Equivariant PnP__
>
> We thank the reviewer for their comment on Equivariant PnP. Please see our response to _Reviewer MMgq_.

---

> > ### Author Response · Authors · 2026-06-20
> >
> > We thank the reviewer for their thoughtful comments. We have now uploaded the revised manuscript. Our responses describe in detail the changes made, which are highlighted in blue throughout the manuscript.
> >
> > To improve the flow of the main paper and save space, we have moved Table 1 to the supplementary material. We will soon update this table in the supplementary to incorporate the reviewer’s suggestion by characterizing the method across varying values of both λ (EquiReg step size) and ζ (measurement step size), alongside the results to be presented in Figure 23.

---

> > > ### Author Response · Authors · 2026-07-06
> > > **New response for lambda_t and zeta_t**
> > >
> > > Following the reviewer’s suggestion to study the broader impact of EquiReg as the regularization parameter ($\lambda_t$) and the measurement guidance ($\zeta_t$) vary, we performed additional experiments and revised the manuscript accordingly to address this concern.
> > >
> > > We address the concern in three parts.
> > >
> > > __Schedule.__ We have now clarified in the manuscript that all experiments use a constant $\lambda$. We retain the notation $\lambda_t$ in the formulation to indicate that the regularization parameter can, in principle, vary across diffusion steps, and our source code provides this functionality. This notation is consistent with the literature (e.g., PSLD), where step sizes or regularization parameters are indexed by the diffusion step $t$ to convey the possibility of using a time-varying schedule. We now explicitly clarify in the manuscript that $\lambda_t$ is kept constant throughout sampling.
> > >
> > > The treatment of the late denoising stages depends on how the equivariance term is incorporated into the solver. We verified the implementation of all methods and confirmed that $\lambda$ remains constant throughout sampling. The exception is DPS and SITCOM, which may apply a schedule to $\zeta_t$. For the DPS algorithm, the effective EquiReg regularization is $\zeta_t \lambda_t.$ The appendix now includes the following clarification:
> > >
> > > _For all experiments, we keep $\lambda_t$ constant throughout iterations. We note that for the DPS method, the effective EquiReg regularization is given by $\zeta_t \lambda_t$. Therefore, scheduling the measurement guidance to decrease over time proportionally reduces the effective equivariance regularization as well._
> > >
> > > We emphasize that using a constant base $\lambda$ follows the standard convention for diffusion regularization hyperparameters. Furthermore, not all DPS experiments use a decreasing schedule for $\zeta_t$, and comparable regularization coefficients in prior work are typically held constant or follow simple deterministic schedules. Table 10 reports the $\lambda$ values used for each task.
> > >
> > > __Sensitivity.__ The reviewer raises an important point regarding the sensitivity of EquiReg to the regularization hyperparameter. We have studied this sensitivity in our experiments (Table 13), where we vary both the regularization weight $\lambda$ and the period during which the regularization is applied. These experiments demonstrate that EquiReg does not require narrow hyperparameter tuning, provided that $\lambda$ is chosen within a reasonable range, similar to other gradient-based methods.
> > >
> > > The reviewers also asked us to clarify which experiments convey the core message of the paper (i.e., broad applicability, robustness, and consistent performance improvements). In response, we have revised the manuscript (e.g., remove discussions on reduced DDIM steps) to reduce the emphasis on robustness under more extreme settings (e.g., very large regularization weights or substantially reduced DDIM steps). We believe the central contribution of the paper is the introduction of the new distribution-dependent notion of equivariance and the MPE function, together with demonstrating how they can be used to regularize conditional diffusion models for inverse problems and mitigate off-manifold sampling trajectories. These primary contributions are supported by Figure 4 and Tables 1–4.

---

> > > > ### Author Response · Authors · 2026-07-06
> > > > **Continue the response for lambda_t and zeta_t**
> > > >
> > > > __$\zeta_t$ tuning.__ Our Equi-DPS experiments use the $\zeta_t$ values reported in the original DPS paper (Chung et al., 2023), which were tuned by the authors for best performance on the FFHQ and ImageNet benchmarks. We adopt these values directly, with Equi-DPS introducing only the additional EquiReg regularization through $\lambda$. During our verification, we noticed that the default $\zeta_t$ values in the official DPS source code differ from those reported in the paper, and our original results reflected the source-code defaults. Therefore, the visualization from the old setting of DPS is now removed, and we have re-run the DPS experiments in Tables 2,12,13 to ensure a fair comparison between DPS and Equi-DPS, with both methods now using the same initial value of $\zeta = 1$.
> > > >
> > > > We have also included an additional analysis consisting of a grid search over $\zeta_t$ and $\lambda$ (Figure 20). These experiments are conducted on a subset of the dataset rather than the full test set used in Table 2. The results show that, for any fixed $\zeta_t$ (i.e., within each column), there always exists a $\lambda > 0$ for which Equi-DPS outperforms DPS. As discussed previously, the effective EquiReg regularization in DPS is given by $\zeta_t \lambda$ (see Algorithm 1). Furthermore, because both the likelihood update and the EquiReg term are gradient-based, excessively increasing either of them can negatively affect the sampling dynamics.
> > > >
> > > > Another important observation is that one can always choose a very small $\zeta_t$, thereby weakening the measurement guidance. In this regime, DPS exhibits poorer measurement consistency and consequently lower performance on inverse problems. At the same time, the weaker conditioning reduces the tendency of the sampling trajectory to drift off the data manifold, making EquiReg less necessary. This explains why Figure 20 shows smaller gains from EquiReg for low-guidance $\zeta_t$ in the super-resolution experiments. However, such settings are generally not desirable for solving inverse problems, since accurate reconstruction requires sufficiently strong measurement guidance. As the measurement guidance is increased, the likelihood term becomes stronger, the sampling trajectory is more prone to deviating from the data manifold, and the benefits of EquiReg become increasingly pronounced. This behavior is consistent with our intuition that stronger conditional guidance in DPS can induce off-manifold trajectories, which EquiReg is designed to mitigate.

---

### Review · Reviewer_MMgq · 2026-05-19

**Summary Of Contributions:**

This work proposes EquiReg, a regularization strategy for diffusion-based posterior sampling in imaging inverse problems. The method leverages equivariance to mitigate likelihood-induced reconstruction errors during the sampling process, encouraging diffusion trajectories to move toward more plausible, on-manifold solutions. The core idea is to use Manifold-Preferential Equivariant (MPE) functions as implicit on-/off-manifold indicators: the regularization value is expected to be low for clean or natural images and high for samples that deviate from the image manifold. Importantly, the proposed MPE functions are independent of the diffusion denoising architecture. The main requirement is that the equivariance structure of the MPE function, implemented through a pre-trained neural network, should differ from the group equivariances already encoded in the diffusion denoiser. The resulting regularizer is presented as a plug-in module that can be incorporated into several posterior sampling schemes, and the paper reports consistent improvements over methods such as DPS, PSLD, and ReSample across both linear and nonlinear inverse problems.

Overall, the paper presents a promising and practically useful approach for improving diffusion-based inverse-problem solvers by exploiting equivariance properties of pre-trained neural networks. The experimental results are strong, and the method appears robust across different sampling algorithms and degradation models. However, I have some concerns regarding the explanation and empirical validation of why the proposed equivariance-based regularization provides better diffusion guidance.
First, the motivation in Figure 2a could be strengthened. The authors show that a pre-trained autoencoder yields lower regularization values for clean images than for images corrupted by different noise distributions. While this supports the intuition that the MPE function may distinguish natural images from perturbed ones, this experiment does not fully reflect the error structure encountered in iterative inverse-problem reconstruction. In posterior sampling, the reconstruction errors are not simply independent random perturbations; they are typically structured by the forward operator, the likelihood term, the noise model, and the dynamics of the sampling algorithm. Therefore, a more representative analysis would be to evaluate the MPE regularization value along actual reconstruction trajectories generated by non-regularized solvers, for example, DPS, PSLD, or ReSample without EquiReg. Plotting the MPE loss as a function of the diffusion timestep or sampling iteration would provide stronger evidence that the proposed regularizer detects and penalizes the types of off-manifold errors that actually arise during inverse-problem sampling.

Second, this trajectory-based analysis would also help clarify the behavior of the different MPE functions studied in Figure 19. From the current figure, it appears that several architectures do not exhibit a strong separation in equivariance loss across different noise variance levels. This raises the question of which architectural properties make a pre-trained network suitable as an MPE function. The paper would benefit from a more detailed discussion of when and why a given architecture satisfies the desired MPE conditions. In particular, the authors should analyze whether the effectiveness of an MPE function depends on factors such as the architecture type, the training objective, the strength of its built-in equivariances, its sensitivity to image degradations, or the mismatch between its equivariance group and that of the diffusion denoiser.

Third, the related-work discussion should more explicitly position EquiReg with respect to prior equivariance-based methods for inverse problems. Equivariance has already been exploited in plug-and-play image reconstruction, for example, in Equivariant Plug-and-Play Image Reconstruction by Terris et al. [1], and more recently in diffusion-based restoration through equivariant sampling strategies [2]. The paper should discuss these works and clearly explain how EquiReg differs from them. In particular, it would be useful to clarify whether the novelty of EquiReg lies in the design of MPE functions as manifold-preferential detectors, in the formulation of the equivariance loss as a plug-in posterior regularizer, in its compatibility with multiple posterior sampling schemes, or in its empirical robustness across diverse inverse problems.

Addressing these points would substantially improve the technical clarity of the paper. In particular, evaluating the MPE loss along realistic reconstruction trajectories and providing a deeper analysis of which architectures actually satisfy the MPE assumptions would make the proposed mechanism more convincing. These additions would also strengthen the connection between the empirical improvements and the claimed role of equivariance in guiding diffusion samples toward the image manifold.

**Audience:**

Yes

**Audience Explanation:**

Yes, solving inverse problems has gathered a wide range of audiences in recent ML/AI venues.

**Broader Impact Concerns:**

I do not have concerns regarding the paper's broader impacts statements.

**Claims And Evidence:**

Yes

**Claims Explanation:**

The paper proposes and incorporates MPE functions as equivalence-promoting regularization in the diffusion model sampling scheme. The results show significant performance improvement and show adaptability in a variety of posterior sampling schemes and different inverse problems. Nevertheless, the authors should clarify how to achieve a good MPE function from pre-trained neural networks. More analysis and experiments validating when equivariance is achieved are required

**Requested Changes:**

1. In Figure 2a, the current comparison between clean images and images with different noise distributions is not fully representative of iterative inverse-problem reconstruction. The authors should evaluate the MPE regularization values along actual non-regularized reconstruction trajectories at different diffusion timesteps.


2. Since reconstruction errors are not purely random noise but are structured by the sensing model, the authors should test the MPE loss on outputs produced by non-regularized solvers for different inverse problems and timesteps.


3. The paper should include a more detailed analysis of the different MPE functions shown in Figure 19, explaining which architectures satisfy the desired MPE conditions and why.

4. Figure 19 appears to show that many architectures do not produce significant differences in equivariance loss across noise variance levels. The authors should discuss this observation and explain what architectural properties make an MPE function effective as an on/off manifold detector.

5. The related work section should include and discuss equivariant methods for imaging inverse problems, especially equivariant plug-and-play reconstruction methods.

---

> ### Author Response · Authors · 2026-06-17
>
> We would like to thank the reviewer for their thoughtful comments and constructive suggestions. We have conducted several new experiments and added clarifications to the manuscript to address these points. Below, we summarize the main additions and revisions.
>
> __Figure 2 / Trajectory Analysis__
>
> We thank the reviewer for their suggestion on Figure 2. We have conducted additional experiments on the dynamics of the MPE loss during unconditional and conditional generation in the absence and presence of EquiReg. For our full and detailed response to this trajectory analysis, please see _NEW Figure 20_  and our comment on “Numerical analysis on how EquiReg reduces distance to manifold or likelihood approximation error” to reviewer F8C5.
>
> __MPE functions and Figure 19__
>
> We thank the reviewer for their comment. We agree that understanding when an MPE function is effective is important. Based on our additional experiments, we identify two primary factors that contribute to MPE effectiveness:
>
> 1. The architectural inductive bias of the MPE function.
>
> 2. Alignment between the training distribution of the MPE function and the data distribution on which MPE properties are evaluated.
>
> We clarify that, among the models considered in Table 8, the CNN autoencoder is the only MPE function trained directly on the same distribution as the evaluation dataset. The CNN autoencoder, which we trained ourselves on the inverse-problem dataset, has by construction an in-distribution that matches the test data. The LDM encoder and ResNet-50 are pre-trained on large-scale image collections whose distribution differs from FFHQ. To disentangle these factors, we have conducted several new analyses and modified the results in the following form:
>
> _Effect of distribution match:_ For the MPE function to be the most effective, one should use a function with a matching training distribution to the distribution used during EquiReg sampling. In our original analysis, the CNN is the only architecture whose training distribution matched the test data, which is in part why it has shown better MPE properties compared to the rest; the weaker MPE signal of the rest can be attributed to this distribution mismatch.
>
> To clarify this, we now report two separate analyses: one where we study the effect of architecture (Figure 24, all networks trained on FFHQ 256 with pre-trained LDM included for reference) and another where we study a general, already trained network publicly available to users in the community (LDM encoder, ResNet-50 in Table 8). You can see in these few figures that indeed, relatively all errors of all functions (apart from the dense net) increase as a function of noise. In practice, the gradient step size associated with a given MPE function can be adjusted to account for differences in scale across architectures.
>
> _Effect of Architecture on MPE strength (distribution held fixed):_ To isolate the architectural contribution, we trained a small ResNet and a small dense network on FFHQ under the same flip-augmentation regime as the CNN autoencoder. Here we show the flip-equivariance error evaluated on the noised image swept from clean to pure Gaussian noise, on held-out FFHQ-256. All three autoencoders are trained on FFHQ under a reconstruction objective; the LDM encoder is an off-distribution reference. The convolutional autoencoder exhibits the MPE property: there is low error on clean in-distribution inputs that rises with noise, with the CNN-AE strongest (lowest clean error ≈0.006 and the largest relative rise) and the ResNet-AE weaker despite ~2× the parameters. The Dense-AE (808M, invariance metric) fails: its error is large on clean inputs and inverts, decreasing under noise, so it cannot discriminate on- from off-manifold samples.
>
> The off-distribution LDM encoder exhibits larger error across all noise levels. Together these show that with the training distribution and procedure fixed, MPE strength can be governed by architectural inductive biases. We note that these analyses are included primarily to better understand the architectural biases for MPEs. However, these analyses are on relatively small datasets and network architectures. Hence, the resulting models should not be viewed as general-purpose pre-trained MPE functions.
>
> Finally, it is important to distinguish equivariant functions from MPE functions. If a network is explicitly designed to be equivariant to a particular group (e.g., rotations), then its equivariance behavior is independent of the input distribution. In such cases, the equivariance error will not distinguish between on- and off-manifold samples under that group, making it not suitable as an MPE function for that transformation. Hence, the suggestion is that if one plans to train a NN to become MPE for a group symmetry to use a NN that does not have an architectural bias toward being equivariant to that group.

---

> > ### Author Response · Authors · 2026-06-17
> >
> > _An equivariant-by-construction function cannot be an MPE function for that group:_ Here (New Figure 25), we show the latent equivariance error of the CNN autoencoder under horizontal translation, measured on the interior (untranslated) crop and swept clean→noise as above. For shifts aligned with the encoder's $16\times$ down-sampling lattice (16 px = 1 latent cell, 32 px = 2 cells), the error is $\approx 0$ and flat across the entire noise sweep; convolutional translation-equivariance holds for any input, so the error cannot separate clean in-distribution faces from heavily perturbed ones, making the function unusable as an MPE for translation. The off-lattice 8 px shift, in contrast, rises with noise and tracks the flip reference. This motivates choosing, for a target symmetry, a network without a built-in architectural bias toward that symmetry.
> >
> > _Denoiser Equivariance:_ The equivariance properties of the diffusion denoiser and the MPE function used by EquiReg are conceptually independent. EquiReg uses an MPE function to differentiate between in/out distributions and does gradient descent on this loss to move into regions with loss Equiloss which corresponds to data samples close/on the manifold.
> >
> > We thank the reviewer again for raising this question. The revised Section 4 now includes a paragraph titled "Architectural and training factors behind MPE effectiveness" that ties the empirical ranking in Tables 8 and 13 and Figure 19 to architectural and training properties of the four networks (CNN autoencoder, LDM encoder, ResNet-50, DNN), including the new experimental analysis. The paragraph identifies contributing factors discussed above and closes with practical guidelines for choosing MPE functions. Training-data alignment should be prioritized, followed by symmetry-preserving augmentation during training. Networks tuned for invariance to general perturbations should be avoided; We frame these as interpretations of the empirical studies rather than as separately validated claims. Importantly, our goal is not to provide a theoretical characterization of when MPE functions emerge. A theoretical analysis on training dynamics of function to show MPE properties is an interesting future direction and is outside the scope of this paper. We will include this suggestion for future work in the conclusion section.
> >
> > Overall, we note that our paper contribution on MPE functions are:
> >
> > 1. We introduce the notion of functions with distribution-dependent equivariance error. This formulation is new. All prior work defines equivariance error global over all inputs for a function, which won’t be useful to discriminate between in/out distributions.
> >
> > 2. We provide empirical evidence that MPE behavior can emerge in commonly trained neural networks under appropriate training conditions.
> >
> > __Comparing EquiReg with Equivariance literature__
> > We thank the reviewer for these points. We clarify how EquiReg differs from prior equivariant approaches for inverse problems.
> >
> > Our paper uses equivariance to detect off-manifold examples (a loss that shows loss error for in-distribution, on-manifold examples and high loss for out-of-distribution ones. This is an important distinction. First, an important contribution of our paper is the formulation of functions with the distribution-dependent equivariance property. Prior equivariant methods typically seek equivariance as a global property of a function, independent of the input distribution.
> >
> > Second, our framework does not enforce equivariance directly; instead, it encourages images that have lower equivariance error under a defined function f. Consequently, a function need not be perfectly equivariant to be useful within EquiReg.
> >
> > Third, having first and second combined, EquiReg uses the MPE function to regularize the sampling toward the manifold. On the other hand, methods such as Equivariant PnP (Terris et al. [1]:) directly enforces equivariance on the denoiser D; this can be similar to designing a score function to be equivariant which is different from our case.

---

> > > ### Author Response · Authors · 2026-06-17
> > >
> > > The revised manuscript now includes a paragraph in Section 2 (Equivariance subsection) that contrasts EquiReg with the closest prior equivariant approaches. More detailed information about prior work that we include in the paragraph are:
> > >
> > > - Terris et al. 2024 (Equivariant PnP). Directly use equivariance to symmetrize the denoiser inside a plug-and-play fixed-point iteration, and is deterministic, MAP-style.
> > >
> > > - Hoogeboom et al. 2022 and architectural-equivariance lines of work. Build equivariance directly into the architecture of the diffusion model so that the denoiser is exactly equivariant by construction (e.g., E(n)-equivariant networks for 3D molecule generation); a broader theoretical line studies probabilistic symmetries and invariant neural networks (Bloem-Reddy and Whye 2020).
> > >
> > > - Daras et al. 2024 (Warped Diffusion). Equivariance for temporal consistency across video frames in image diffusion. The equivariance helps to ensure temporal coherence in the video by relating the generating noise across time frames.
> > >
> > > Finally, in EquiReg, Equivariance enters as a posterior-sampling regularizer, applied through an MPE function that is separate from the denoiser; an important note is that its equivariance is approximate and distribution dependent (low for in, high for out). Hence, the equivariance error acts as a manifold-preference signal rather than a hard constraint. The combination of these is what enables the wide applicability of EquiReg. Our experiments show that EquiReg improves inverse-problem solving performance on a wide range of inverse problems on a wide range of diffusion models both latent and pixel-based. The rest of our experiments characterize  EquiReg (e.g., robust to its choice of parameter, flexibility of its period, minimal impact of diversity of sampling).
> > >
> > > __The reviewer commented: “More analysis and experiments validating when equivariance is achieved are required.”__
> > >
> > > We would like to clarify our interpretation of this comment. Equivariance is a property of a function with respect to a specified transformation group. For EquiReg to work, we do not require the equivariance error of the MPE function to go to zero. See NEW Figure 20 where we show the equivariance error of our used MPE function is lower for $x_{0|t}$ in unconditional generation than when conditioning with DPS is used. This verifies that the EquiLoss captures well if a sample is pushing off the manifold. For our full response, please see our response on “Numerical analysis on how EquiReg reduces distance to manifold or likelihood approximation error” to Reviewer F8C5.
> > >
> > > Finally, we hope the above discussion clarifies the contributions of the paper. These include: i) the introduction of distribution-dependent equivariance error and MPE functions, ii) the use of MPE functions as plug-in regularizers for diffusion posterior sampling, and iii) extensive empirical validation across multiple posterior sampling algorithms and inverse problems. We have expanded the manuscript to better position these contributions relative to prior equivariant methods and to provide additional empirical evidence supporting the proposed mechanism.

---

> > > > ### Author Response · Authors · 2026-06-17
> > > >
> > > > __Comparing to PnP approaches (e.g., Equivariant-PnP)__
> > > >
> > > > The goal of our experimental design is to evaluate _EquiReg as a regularization framework for diffusion-based inverse problem solvers_, rather than to compare different classes of generative models or inverse solvers. To isolate the effect of the proposed regularizer, we adopt a controlled paired setting (e.g., DPS vs. Equi-DPS, ReSample vs. Equi-ReSample, PSLD vs. Equi-PSLD), where the backbone model, training procedure, and sampling algorithm are kept fixed, and only the regularization term is changed. This ensures that any observed performance differences can be directly attributed to EquiReg.
> > > >
> > > > For this reason, a direct comparison to PnP methods such as Equivariant-PnP is not particularly informative for assessing the contribution of EquiReg. PnP and diffusion-based approaches differ fundamentally in their architectures, generative modelling assumptions, and representational capacity. As a result, such a comparison would conflate the effect of the regularization strategy with differences arising from the underlying generative models and learned priors themselves. For example, a stronger diffusion backbone could outperform a PnP method irrespective of the regularization employed. Consequently, comparing EquiReg against Equivariant-PnP would introduce an apples-to-oranges evaluation and would not provide a clean assessment of the benefits of the proposed regularization framework.

---

> > > > > ### Author Response · Authors · 2026-06-20
> > > > >
> > > > > We thank the reviewer for their thoughtful comments. We have now uploaded the revised manuscript. Our responses describe in detail the changes made, which are highlighted in blue throughout the manuscript.